# GEOMMOTIF: A BENCHMARK FOR ARBITRARY GEOMETRIC PRESERVATION IN PROTEIN GENERATION

**Pavel Strashnov**[1,*,†]       **Andrey Shevtsov**[1,*]       **Viacheslav Meshchaninov**[2,3]
**Olga Kardymon**[1]       **Dmitry Vetrov**[2]

[1]AXXX, Center for AI-driven Drug Design   [2]Constructor University   [3]HSE University

## ABSTRACT

Motif scaffolding in protein design involves generating complete protein structures while preserving the 3D geometry of designated structural fragments, analogous to image outpainting in computer vision. Current benchmarks focus on functional motifs, leaving general geometric preservation capabilities largely untested. We introduce GeomMotif, a systematic benchmark that evaluates arbitrary structural fragment preservation without requiring functional specificity. We construct 57 benchmark tasks, each containing one or two motifs with up to 7 continuous fragments, by sampling from the Protein Data Bank (PDB) to ensure a ground-truth, solvable conformation for every problem. The tasks are characterized by comprehensive structural and physicochemical properties: size, geometric context, secondary structure, hydrophobicity, charge, and degree of burial. These features enable detailed performance analysis beyond simple success rates, revealing model-specific strengths and limitations. We evaluate models using scRMSD and pLDDT for geometric fidelity and clustering for structural diversity and novelty. Our results show that sequence-based and structure-based approaches find different tasks challenging, and that geometric preservation varies significantly with structural and physicochemical context. GeomMotif provides insights complementary to function-focused benchmarks and establishes a foundation for improving protein generative models.

## 1 INTRODUCTION

Deep learning is revolutionizing protein design, enabling the creation of novel enzymes, vaccines, and protein-based therapeutics. A central task in this field is **motif scaffolding**, where a new protein is generated around a specific functional fragment, preserving its precise 3D geometry (Wang et al., 2022). This process is analogous to "image outpainting" (Saharia et al., 2022) and is a critical step in engineering new functional proteins.

However, a critical **diagnostic blind spot** exists in current evaluation methods. Benchmarks like MotifBench (Zheng et al., 2025) focus predominantly on known functional sites, such as enzyme active sites, which inherently biases evaluation toward specific structural and physicochemical contexts. More fundamentally, this approach conflates two distinct challenges: preserving 3D geometry and satisfying complex functional requirements. Beyond geometric scaffolding, functional success depends critically on residue identities, charge distributions, hydrophobic packing, side-chain conformations, etc. When a model fails on such tasks, it is impossible to diagnose whether the cause was a fundamental inability to preserve geometry or incompatibility with these physicochemical requirements.

This conflation creates a critical diagnostic problem because in protein engineering, geometric precision is a direct prerequisite for biological function. For computationally designed protein binders, for instance, experimental success rates can plummet from nearly 50% to zero as the backbone RMSD of the binding motif deviates from its target by just 1.0 Å (Cao et al., 2022). Consequently, geometric accuracy serves as the primary computational filter used to triage designs before costly wet-lab validation. Yet, current benchmarks entangle this foundational geometric challenge with complex functional requirements, making it impossible to distinguish between a model's failure stemming from a core inability to preserve structure or from more subtle functional incompatibilities.

---

[*]Core contributor, [†]Project lead

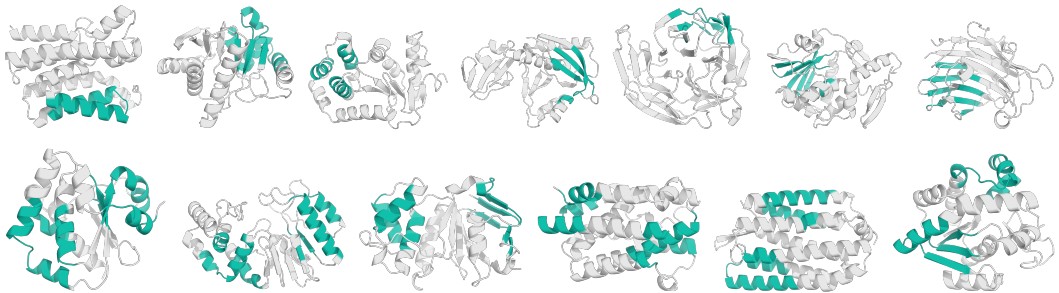

Figure 1: **Representative geometric preservation tasks from the GeomMotif benchmark.** The goal is to generate a scaffold to connect predefined motifs while preserving their 3D geometry.

To address this gap, we introduce GeomMotif, a benchmark designed to rigorously and exclusively evaluate the fundamental capability of **arbitrary geometric preservation**. By sampling structural fragments from across the Protein Data Bank (PDB) without a bias toward known functions, GeomMotif provides a comprehensive test of a model's ability to generalize across diverse structural contexts. This "protein outpainting" approach establishes a foundational test of generative capabilities: can a model maintain local and long-range geometric relationships between arbitrarily selected residues? Success on GeomMotif offers a direct measure of a model's core generative capacity, providing insights that are complementary to function-focused benchmarks. Critically, every task in GeomMotif is **guaranteed to be solvable** by design, is **modality-agnostic** to allow fair comparison between sequence-based and structure-based approaches, and is characterized by a rich set of physicochemical properties to enable fine-grained performance analysis.

Our main contributions are:

- We introduce GeomMotif, a modality-agnostic benchmark of 57 diverse protein scaffolding tasks that isolates geometric preservation from functional constraints.
- We adapt established evaluation protocols and the **SUN (Successful, Unique, Novel)** to provide a single score reflecting geometric accuracy, structural diversity, and novelty.
- Our evaluation of seven models reveals a significant performance gap, with structure-based models like Genie2 (39.4% SUN) and RFdiffusion (37.8%) far outperforming sequence-based models (best at 3.5%).
- We uncover counterintuitive performance patterns, notably that ESM3's multimodal (sequence + structure) mode (1.4% SUN) consistently underperforms its sequence-only counterpart (3.5%), suggesting structure conditioning can introduce conflicting signals.
- Our analysis reveals clear architectural limitations, demonstrating that sequence-based models categorically fail on tasks with spatially separated **(paired) motifs**, while structure-based models exhibit complex, non-monotonic performance as fragment complexity increases.

We provide the benchmark data at HuggingFace and the complete task construction and evaluation code at GitHub.

## 2 RELATED WORK

Computational protein design has evolved from physics-based approaches to modern machine learning methods, with motif scaffolding emerging as an important benchmark for evaluating generative models.

**Classical Approaches.** Rosetta pioneered computational motif scaffolding through physics-based energy functions and extensive conformational sampling (Kuhlman et al., 2003). The enzyme design modules introduced by the Baker lab (Rothlisberger et al., 2008; Jiang et al., 2008) enabled systematic placement of catalytic residues within pre-existing scaffolds, establishing methodologies for rational enzyme design. Subsequent developments included RosettaDesign (Kuhlman et al., 2003), enzyme design modules (Richter et al., 2011), and the fold-from-loops approach for epitope grafting (Correia et al., 2014). While successful for specific applications, these methods face computational limitations that scale exponentially with design complexity, motivating the transition to machine learning approaches.

**Deep Learning Methods.** The paradigm shift toward neural approaches began with Wang et al. (2022), who first formalized motif scaffolding as a machine learning task. RFdiffusion (Watson et al., 2023) revolutionized this field by adapting diffusion models from computer vision to protein

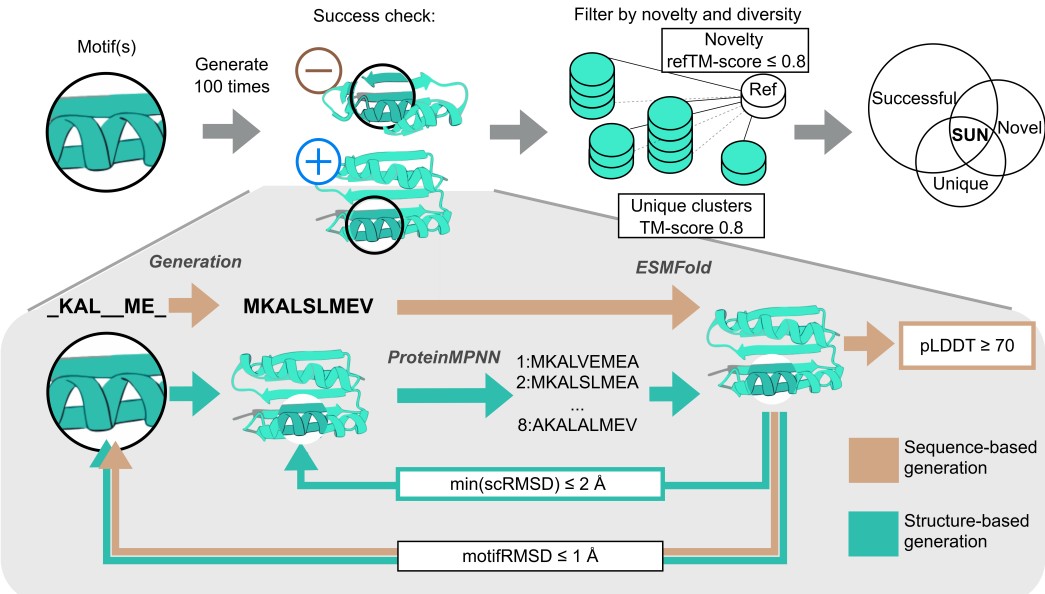

Figure 2: **GeomMotif**'s modality-agnostic evaluation pipeline. Generated proteins are assessed for geometric fidelity (**Success**), structural diversity (**Unique**), and **novelty**, yielding a final **SUN score**.

structure generation, introducing a benchmark of 25 functional motifs focused on enzymatic active sites and binding interfaces. This established the standard evaluation protocol: generating backbone structures, designing sequences with ProteinMPNN (Dauparas et al., 2022), and validating through structure prediction. Several models have adopted the RFdiffusion benchmark paradigm, including EvoDiff (Alamdari et al., 2024), DPLM (Wang et al., 2024b), DPLM-2 (Wang et al., 2024c), DiMA (Meshchaninov et al., 2025), and Genie 2 (Lin et al., 2024). FrameFlow (Yim et al., 2024) introduced SE(3) flow matching for protein backbone generation with reported success on motif scaffolding tasks. Multi-motif formulations demonstrated by Lin et al. (2024) and Liu et al. (2024) address scenarios with independently floating functional regions.

**MotifBench.** MotifBench (Zheng et al., 2025) is a standardized benchmark for functional motif scaffolding with 30 test cases, derived from the original RFdiffusion dataset. It focuses on enzymatic active sites and binding interfaces, using a fixed evaluation pipeline based on ProteinMPNN and ESMFold. While important for specific applications, this focus on **functional** sites makes it difficult to determine if a model's failure stems from an inability to preserve geometry or from not meeting complex functional requirements. Additionally, some tasks involving complex functional sites may be inherently unsolvable by current methods. GeomMotif is designed to be complementary to these efforts. It provides a benchmark that isolates the challenge of **general geometric preservation**, allowing for a direct assessment of the core structural capabilities required for successful protein design.

## 3 METHOD

We design GeomMotif to systematically evaluate the geometric preservation capabilities of protein generation models. Our approach addresses key limitations of existing benchmarks through principled task selection, comprehensive property characterization, and modality-agnostic evaluation.

### 3.1 DESIGN PRINCIPLES

GeomMotif is built on four core principles. First, we ensure systematic **coverage** of protein structure space through uniform sampling rather than focusing on functional regions. Second, we categorize tasks by **complexity** using both the number of motifs (1-2) and total fragments (1-7). Third, we characterize each task using eight structural and physicochemical **properties** to enable detailed, property-driven analysis. Fourth, we guarantee task **solvability** by extracting fragments from natural proteins, ensuring at least one valid solution exists.

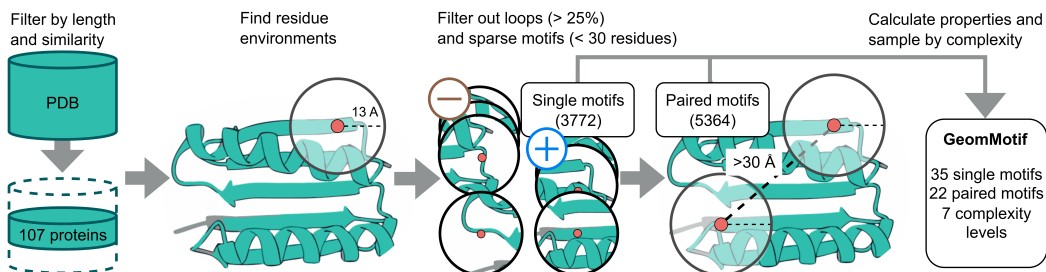

Figure 3: **Pipeline for GeomMotif benchmark construction.** Systematic workflow for constructing the benchmark tasks. The process includes filtering high-quality monomeric structures, clustering for non-redundancy, ensuring foldability, identifying residue neighborhoods, filtering motifs based on structural properties, and organizing tasks by complexity level. This construction procedure ensures comprehensive coverage of protein structural space and guarantees the task solvability.

## 3.2 TASK CONSTRUCTION

We construct benchmark tasks from monomeric protein structures to ensure structural stability and avoid conformational ambiguity. Proteins in complexes may adopt different structures in isolation, potentially leading to evaluation inconsistencies when comparing generated structures against ground truth conformations.

To assemble an initial high-quality dataset and ensure systematic coverage, we filtered the Protein Data Bank (PDB) according to three key criteria. First, we selected only structures determined by X-ray crystallography with a resolution of 2.5 Å or better to ensure atomic precision. Second, we included only biological monomers to avoid the conformational ambiguities that can arise when proteins are part of a larger complex. Finally, we imposed a length limit of 250 residues to prevent potential misfolding artifacts, as substructures from larger proteins may require their full context for proper folding. This rigorous filtering process resulted in an initial dataset of 24,001 structures.

To eliminate redundancy while preserving structural diversity, we apply a two-stage clustering protocol. First, we cluster sequences at 80% identity with 90% minimum coverage using MMseqs2 (Steinegger & Söding, 2017). Second, we perform structural clustering using complete linkage hierarchical clustering with TM-score threshold 0.5 and coverage 30%. The rationale behind the choice of particular threshold values is discussed in App. F.

To guarantee solvability, we generate ESMFold predictions for each cluster representative and retain only structures where the predicted fold aligns with the experimental structure at RMSD ≤ 1.0 Å. This filtering ensures all benchmark tasks are inherently solvable by the evaluation pipeline, addressing a critical limitation identified in previous benchmarks (Zheng et al., 2025). This procedure yields 107 unique protein structures from which we construct a set of candidate structural motifs. First, we iterate through **each residue** of every protein, treating it as a potential center. A motif is defined as the set of all residues whose $C\alpha$ atoms are within a 13Å radius of this central residue's $C\alpha$ atom. This step yields a large pool of overlapping structural neighborhoods. We then apply a series of filters to refine this set: (1) we **exclude small motifs** containing fewer than 30 residues, (2) **reduce local redundancy** by removing neighborhoods which share more than

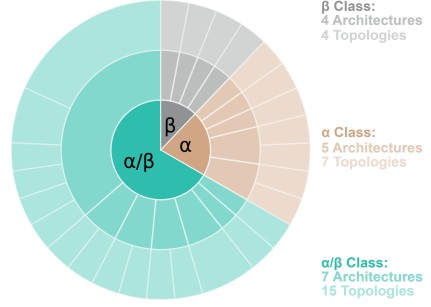

Figure 4: **GeomMotif covers diverse CATH structural folds.** Representation of fundamental protein architecture types across the benchmark shows balanced coverage of major structural classes: mainly-alpha ($\alpha$), mainly-beta ($\beta$) and alpha-beta ($\alpha/\beta$). For a detailed breakdown of the CATH architectures, see App. D

20% of their residues, and (3) **filter out motifs with >25% loop content** as determined by DSSP secondary structure assignment. This process produces 3,772 single-motif candidates. For paired-motif tasks, we identify motif pairs within the same structure separated by ≥30Å between their centers, yielding 5,364 paired motifs.

Although the residues forming a motif are spatially close, they are often non-contiguous in the amino acid sequence. The 'fragment complexity' of a task is therefore defined by the number of separate, continuous sequence segments that constitute the motif. To ensure the benchmark assesses model performance across a spectrum of geometric challenges, we stratify the final selection by fragment complexity. We curate the benchmark to include an **equal number of tasks for each fragment count** (1-7 for single motifs, 3-7 for paired), sampling up to five representatives for each category. This process yields our final benchmark of 57 tasks: 35 single-motif and 22 paired-motif problems. These tasks span diverse CATH fold classes, ensuring broad coverage of protein structural space (Fig. 4).

To minimize memorization and test true generative capabilities, we allow biologically plausible length variations for variable regions while preserving fixed motif geometry. This forces models to generate valid structures across diverse length contexts rather than reproducing memorized patterns. Detailed formulation is provided in App. A.1. The full specification of all the tasks is available in Tab. 3.

To enable a fine-grained analysis of model performance, each constructed task is further defined by a comprehensive set of structural and physicochemical properties, as detailed in Sec. 3.3.

## 3.3 PROPERTY CHARACTERIZATION

Each task in GeomMotif is characterized by eight structural and physicochemical properties that enable fine-grained analysis of model performance:

- **Secondary structure composition.** We characterize the motif's secondary structure using three distinct properties derived from DSSP assignments.
  - **Helical Content.** The proportion of residues in $\alpha$-helices, $3_{10}$-helices, $\pi$-helices.
  - **Extended Content.** The proportion of residues in $\beta$-strands and $\beta$-bridges.
  - **Loop Content.** The proportion of residues in loop or bend regions.
- **Motif size.** the number of residues in the motif, representing the extent of geometric constraints.
- **Mean hydrophobicity** of motif residues using the Eisenberg scale (Eisenberg et al., 1984), correlating with the tendency of residues to be buried versus exposed.
- **Burial ratio,** fraction of motif residues with relative solvent accessibility (RSA) $< 0.2$, indicating positions with stringent packing constraints.
- **Absolute charge density** calculated as absolute charge per residue based on standard assignments (Arg/Lys: +1, Asp/Glu: -1), affecting electrostatic compatibility.
- **Structural context** we evaluate as a ratio of internal contacts (between motif residues) to external contacts (between motif and non-motif residues) using a 4.5Å distance threshold.

These properties enable analysis beyond simple success rates, revealing which structural features contribute to task difficulty for different model architectures. The extended details on the calculation of the properties is provided in App. B. The detailed breakdown of all the properties for each task is provided in Tab. 2.

## 3.4 EVALUATION FRAMEWORK

Our evaluation framework extends established protocols from protein motif scaffolding benchmarks (Watson et al., 2023; Alamdari et al., 2024; Zheng et al., 2025), adapting them to assess general geometric preservation capabilities across both structure-based and sequence-based generative models. The framework captures three fundamental aspects of generative performance: geometric fidelity, structural diversity, and novelty relative to known proteins (Fig. 2).

The evaluation protocol differs between sequence and structure modalities to accommodate their distinct outputs. For structure-based models, we follow the established pipeline of generating backbone structures, designing 8 compatible sequences using ProteinMPNN (Dauparas et al., 2022), and validating through structure prediction. Sequence-based models bypass the ProteinMPNN step, as they directly generate sequences that we fold using ESMFold (Lin et al., 2023). The rationale for these pipeline choices is detailed in the App. E. In both cases, we assess geometric preservation by computing the backbone scRMSD between motif residues in the predicted structure and their corresponding positions in the ground truth.

A generated protein is considered **successful** when it meets two criteria: (1) the motif RMSD falls below 1.0Å, demonstrating faithful geometric preservation, and (2) the predicted structure exhibits

high confidence with average pLDDT $\geq 70$, ensuring the overall fold is well-formed. To quantify **diversity** among successful designs, we cluster scaffolds using TM-score based hierarchical clustering with threshold 0.8. The number of resulting clusters reflects the breadth of structural solutions each model generates. **Novelty** assessment involves computing the TM-score between successful scaffolds and the ground truth structures. Scaffolds with TM-score $< 0.8$ to the reference structures represent genuinely novel designs.

To capture the competing objectives in protein design—accuracy, diversity, and novelty—we adopt the SUN (Successful, Unique, Novel) score (Sriram et al., 2024). The SUN score represents the proportion of generated samples that simultaneously achieve all three criteria. The Uniqueness and Novelty components are assessed *within the set of successful designs*, ensuring that we only measure the diversity and originality of viable structures.

$$\text{SUN} = P(\text{Successful} \cap \text{Unique} \cap \text{Novel}) \tag{1}$$

This metric provides a stringent assessment of model capabilities, rewarding only those designs that combine geometric accuracy with structural diversity and originality. By requiring simultaneous achievement of all criteria, the SUN score naturally balances the competing objectives in protein design. A detailed discussion of this metric's interpretation and utility is provided in App. C.

Evaluation proceeds in both fixed-length and variable-length settings to distinguish memorization from true generalization. The fixed-length evaluation constrains proteins to match ground truth lengths exactly, establishing a controlled baseline. The variable-length setting permits biologically reasonable length variations, testing models' capacity to generate valid structures without strict dimensional constraints. This dual evaluation reveals whether models genuinely understand protein geometry or merely reconstruct memorized patterns.

## 4 EXPERIMENTS

### 4.1 EXPERIMENTAL SETUP

We evaluate ten protein generation models spanning different architectural paradigms: RFdiffusion (Watson et al., 2023), FrameFlow (Yim et al., 2024), Genie2 (Lin et al., 2024), La-Proteina (Geffner et al., 2025), Protpardelle-1c (Lu et al., 2025), and RFdiffusion2 (Ahern et al., 2025) (structure-based approaches), ESM3 1.4B (Hayes et al., 2025), DPLM-650M, DPLM-3B (Wang et al., 2024b) (sequence-based approaches), and a random baseline. For each of the 57 tasks, we generate 100 samples per model and evaluate them. To quantify the uncertainty in our SUN score measurements, we conduct bootstrap analysis: for each of the 57 tasks, we resample the 100 generated samples with replacement across 100 bootstrap iterations, calculate the SUN score for each bootstrap sample, and compute standard deviations across iterations.

To assess model capabilities beyond potential memorization of PDB structures, we conduct experiments in two settings. In the primary evaluation, task lengths vary from ground truth to test true generalization (Sec. 4.2). We also include a fixed-length evaluation (App. H), where protein lengths match ground truth exactly, serving as a controlled baseline to understand model behavior when length variation is removed as a factor.

### 4.2 OVERALL PERFORMANCE ACROSS TASK CATEGORIES

Figure 5 presents the performance of evaluated models on the GeomMotif benchmark using the SUN (Successful, Unique, Novel) metric. The SUN score represents the proportion of generated samples that simultaneously achieve geometric accuracy, structural diversity, and originality relative to known proteins.

Structure-based models demonstrate substantially higher SUN scores compared to sequence-based approaches. Genie2, La-Proteina, RFdiffusion, and Protpardelle-1c achieve comparable performance (39.4%, 38.8%, 37.8%, and 33.8%, respectively), while FrameFlow and RFdiffusion2 achieve only 23.3% and 17.9%. Interestingly, RFdiffusion2 shows notably weaker performance than the original RFdiffusion. This likely stems from RFdiffusion2 being specifically developed and tuned for enzyme design starting from precise atomic motifs of catalytic active sites, while the original RFdiffusion was trained to be as broad as possible. We hypothesize that this narrow focus results in high scores on the

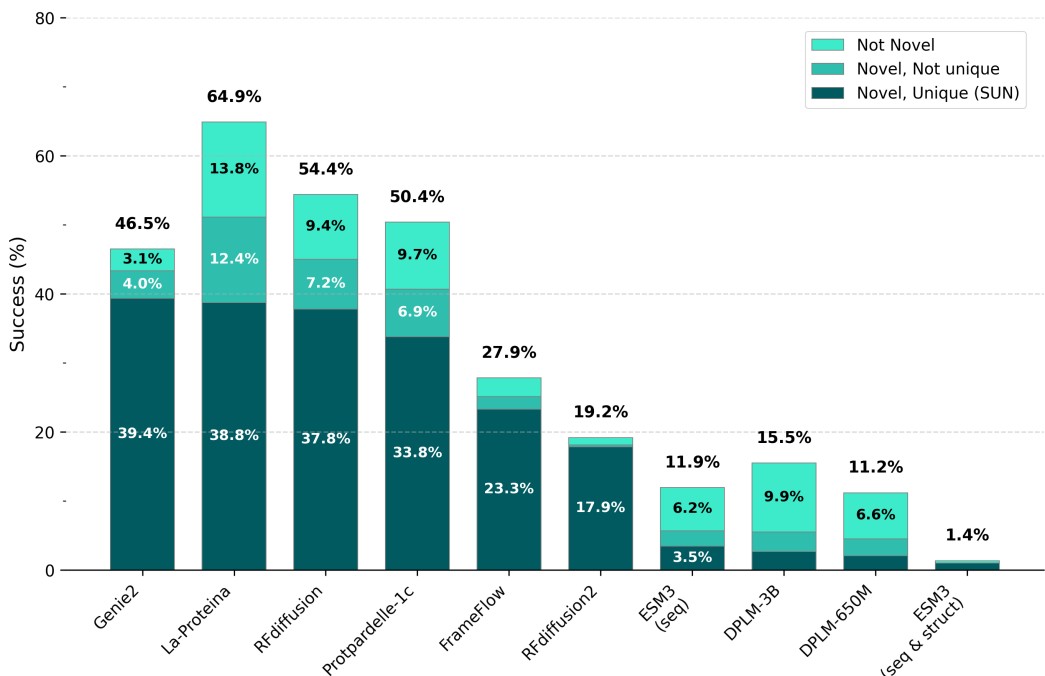

Figure 5: **Comparative performance of protein generation models on the GeomMotif benchmark.** Performance metrics for 10 protein generation models measured by the SUN score (Success, Uniqueness, Novelty). The percentage above each bar indicates overall success rate at preserving motif geometry. Each bar is segmented to show: proteins that are both novel and unique (dark teal, representing the SUN score), proteins that are novel but not structurally diverse (medium teal), and proteins that preserve geometry but are not novel (light teal). Structure-based models (left) substantially outperform sequence-based approaches (right).

Atomic Motif Enzyme Benchmark Ahern et al. (2025) (41/41 solved cases) but poor performance on GeomMotif (Success rate 19.2%, SUN 17.9%). In contrast, the original RFdiffusion was trained for a broad line of tasks and shows mediocre Atomic Motif Enzyme results (16/41 Ahern et al. (2025)) but much higher GeomMotif performance (Success rate 54.4%, SUN 37.8%). This pattern **highlights the complementary nature** of the Atomic Motif Enzyme Benchmark and GeomMotif in evaluating different aspects of scaffolding capability.

Among sequence-based models, ESM3 in sequence-only mode shows the highest performance at 3.5%, followed by DPLM-3B (2.7%) and DPLM-650M (2.1%). Notably, the larger DPLM-3B shows only marginal improvement over DPLM-650M, suggesting that simple parameter scaling may not address the underlying challenges of sequence-based motif scaffolding. Surprisingly, ESM3 variant using both sequence and structure modalities achieve significantly lower scores (1%) compared to

Table 1: **Component analysis of the SUN metric across model types and task categories.** Detailed breakdown of the SUN metric into its constituent components (Success, Novelty, Uniqueness) for single and paired motif tasks. Results demonstrate that while structure-based models maintain high performance across all metrics for single motifs, performance degrades significantly for paired motifs, with sequence-based models showing near-zero SUN scores on paired motifs. Bold values indicate best performance per metric category.

| Model | Successful, % ↑ | | Novel, % ↑ | | Unique, % ↑ | | SUN Score ↑ | |
| | Single | Paired | Single | Paired | Single | Paired | Single | Paired |
|---|---|---|---|---|---|---|---|---|
| Genie2 | 60.1 ± 1.0 | 32.9 ± 0.4 | 60.1 ± 1.0 | 26.6 ± 0.5 | 59.9 ± 1.0 | 22.5 ± 0.3 | 59.9 ± 1.0 | **18.8** ± 0.4 |
| La-Proteina | **67.1** ± 0.7 | **62.7** ± 0.5 | **67.1** ± 0.6 | **35.2** ± 0.9 | 61.3 ± 0.6 | **22.7** ± 0.4 | 61.3 ± 0.6 | 16.2 ± 0.5 |
| RFdiffusion | 65.1 ± 0.4 | 43.7 ± 1.0 | 65.1 ± 0.4 | 25.0 ± 0.5 | **62.4** ± 0.4 | 20.5 ± 0.7 | **62.4** ± 0.4 | 13.2 ± 0.4 |
| Protpardelle-1C | 56.2 ± 0.7 | 44.6 ± 0.5 | 56.2 ± 0.7 | 25.2 ± 0.4 | 53.5 ± 0.7 | 22.6 ± 0.3 | 53.5 ± 0.7 | 14.1 ± 0.4 |
| FrameFlow | 30.6 ± 0.4 | 25.1 ± 0.4 | 30.6 ± 0.4 | 19.7 ± 0.3 | 30.6 ± 0.4 | 20.2 ± 0.5 | 30.6 ± 0.4 | 16.0 ± 0.7 |
| RFdiffusion2 | 24.9 ± 0.6 | 13.5 ± 0.5 | 24.9 ± 0.6 | 11.4 ± 0.3 | 24.9 ± 0.6 | 12.7 ± 0.4 | 24.9 ± 0.6 | 10.8 ± 0.3 |
| ESM3 (seq) | 17.4 ± 0.3 | 6.5 ± 0.5 | 11.3 ± 0.5 | 0.1 ± 0.0 | 10.1 ± 0.2 | 0.1 ± 0.0 | 6.8 ± 0.3 | 0.1 ± 0.0 |
| DPLM-3B | 19.3 ± 0.7 | 11.0 ± 0.5 | 10.2 ± 0.5 | 0.9 ± 0.2 | 9.8 ± 0.5 | 0.6 ± 0.1 | 4.9 ± 0.4 | 0.5 ± 0.1 |
| DPLM-650M | 15.9 ± 0.4 | 6.5 ± 0.5 | 8.7 ± 0.2 | 0.4 ± 0.1 | 8.1 ± 0.3 | 0.3 ± 0.1 | 4.0 ± 0.2 | 0.2 ± 0.1 |
| ESM3 (seq & struct) | 2.0 ± 0.1 | 0.7 ± 0.2 | 2.0 ± 0.1 | 0.0 ± 0.0 | 2.0 ± 0.1 | 0.0 ± 0.0 | 2.0 ± 0.1 | 0.0 ± 0.0 |
| Random | 0.0 ± 0.0 | 0.0 ± 0.0 | 0.0 ± 0.0 | 0.0 ± 0.0 | 0.0 ± 0.0 | 0.0 ± 0.0 | 0.0 ± 0.0 | 0.0 ± 0.0 |

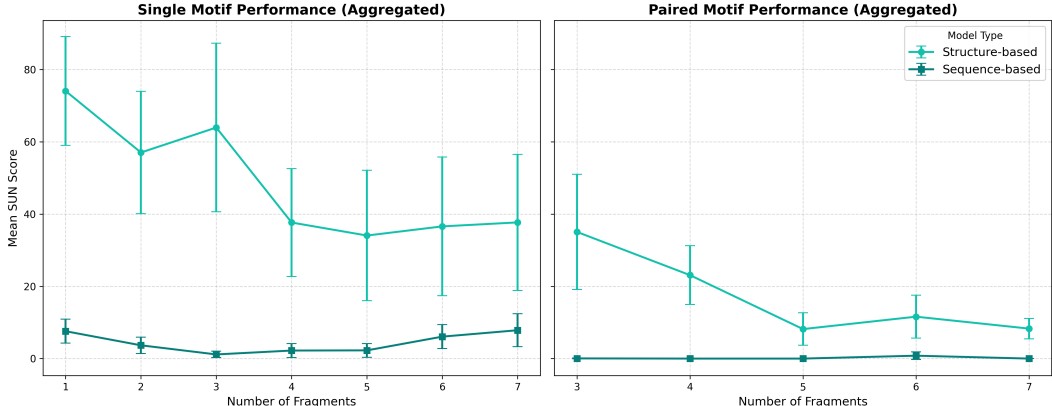

Figure 6: **Fragment complexity vs. model performance.** Structure-based models (cyan) show monotonic degradation with increasing fragment count, while sequence-based models (dark cyan) categorically fail on paired motifs. Error bars show standard deviations.

the sequence-only model. This counterintuitive result aligns with performance patterns observed on the RFdiffusion motif scaffolding benchmark reported in Wang et al. (2024c), where ESM3 in co-generation mode solved fewer tasks with lower average success rate than in sequence-only mode.

Table 1 decomposes the SUN metric into its constituent components and shows distinct performance patterns between single and paired motif tasks. For single motif tasks, RFdiffusion achieves the highest success rate (54.4%, next is Genie2 with 46.5%), but fall short against Genie2 by overall SUN score (37.8% vs 39.35%). The performance gap widens substantially for paired motif tasks, where Genie2 demonstrates the best performance (18.8% SUN), though all models show performance degradation compared to single motif scenarios.

The decomposition reveals that performance differences primarily stem from success rates rather than novelty or uniqueness metrics. For instance, RFdiffusion maintains comparable high novelty (45.05%) and uniqueness (41.4%) for successful single motif designs, indicating that when the model succeeds, it generates diverse and original solutions. However, the dramatic drop in success rates for paired motifs (43.7% for RFdiffusion) constrains the achievable SUN scores regardless of downstream novelty and diversity.

Sequence-based models demonstrate uniformly lower performance across all metrics, with paired motif tasks proving particularly challenging. ESM3 in sequence-only mode achieves the best sequence-based performance for single motifs (6.8 % SUN) but drops to near-zero for paired motifs (0.1% SUN). This categorical failure on paired motifs reveals a fundamental limitation in sequence-based approaches for maintaining geometric relationships between spatially separated regions.

These results show that current structure-based models possess significant advantages for geometric preservation tasks, and highlight the need for architectural innovations to enable sequence-based models to capture long-range spatial constraints. The performance patterns validate GeomMotif's design as a discriminative benchmark that reveals model-specific capabilities and limitations. These overall results, however, raise the question: what structural factors drive these performance differences? We first examine how geometric complexity affects model capabilities.

### 4.3 IMPACT OF FRAGMENT COMPLEXITY

To systematically evaluate how geometric complexity affects model performance, we aggregated results across all tasks by fragment count for each model type (Fig. 6). This aggregation reveals clear architectural differences in how models handle increasing structural constraints.

**Structure-based models demonstrate the expected monotonic relationship between fragment complexity and performance.** Table 7 and Figure 13 (Sec. K) provide the detailed breakdown of the performance of each model. For single motifs, mean SUN scores decline as fragment count increases. RFdiffusion achieves 91.9% on single fragments but drops to 38.7% at seven fragments, Genie2 shows a similar trend from 76.7% to 56.8%. This consistent degradation reflects the increasing difficulty of satisfying multiple geometric constraints simultaneously. While individual tasks within each complexity bin show performance variations due to their specific structural and

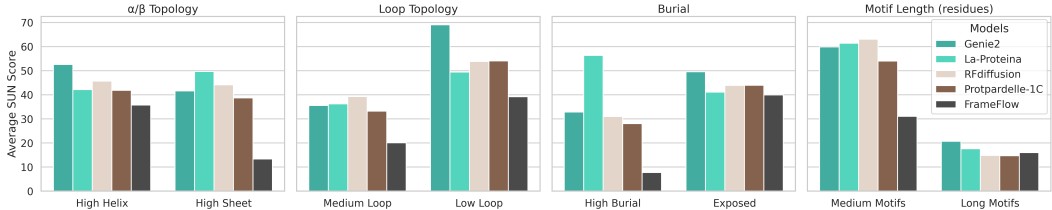

Figure 7: **Structure-based model performance across property categories.** Average SUN scores reveal shared sensitivities: helical over $\beta$-sheet topology, low over medium loop content, exposed over buried residues, and medium over long motifs. Sequence-based models omitted due to low overall success rates.

physicochemical properties, the aggregate trend clearly demonstrates that geometric complexity systematically challenges structure-based models.

For paired motifs, structure-based models face substantially greater challenges due to the spatial separation between motif regions. Performance across all fragment counts remains markedly lower than single-motif tasks, with mean SUN scores rarely exceeding 40%.

**Sequence-based models fail categorically on spatial separation.** All sequence-based models achieve near-zero performance on paired motifs, regardless of fragment complexity. This exposes a fundamental limitation: linear sequence models cannot encode spatial relationships between disconnected regions. For single motifs, these models show modest improvements at 7 fragments, possibly because highly fragmented motifs resemble the masked sequence recovery tasks used in their pre-training.

**The structure modality can impair rather than enhance performance.** ESM3's multimodal variant consistently underperforms its sequence-only counterpart, with the performance gap widening as fragment complexity increases. This counterintuitive result suggests that structural conditioning may introduce conflicting optimization signals, particularly when geometric constraints become more complex.

These patterns demonstrate that protein scaffolding difficulty scales with geometric complexity in expected ways when examined at aggregate level. Task-specific variations exist due to differences in physicochemical properties, but the fundamental relationship is clear: structure-based models struggle progressively with increasing constraints, while sequence-based models fundamentally cannot handle spatial separation. Understanding these limitations is essential for developing next-generation protein design models.

## 4.4 PROPERTY-BASED ANALYSIS OF GEOMETRIC PRESERVATION

Beyond fragment complexity, we investigated how specific structural and physicochemical properties influence model performance using our task characterization. The analysis shows that model success rates correlate with several key properties, revealing different performance patterns across various architectures (Figs. 7 and 8).

**Secondary Structure Composition Effects** Secondary structure composition emerges as a dominant factor affecting geometric preservation. For structure-based models, tasks with high $\alpha$-helical content consistently yield higher success rates. The SUN scores for Genie2, RFdiffusion and FrameFlow on single-motif tasks with >70% helical content are 84.6%, 83.4% and 54.4%, respectively, compared to 42.9%, 45.2% and 14.0% for tasks with <30% helical content. This disparity is particularly pronounced in small single-motif contexts, where purely helical motifs (>90% helix) achieve the highest performance (mean SUN score 94.6% for RFdiffusion). Conversely, $\beta$-sheet-rich motifs present

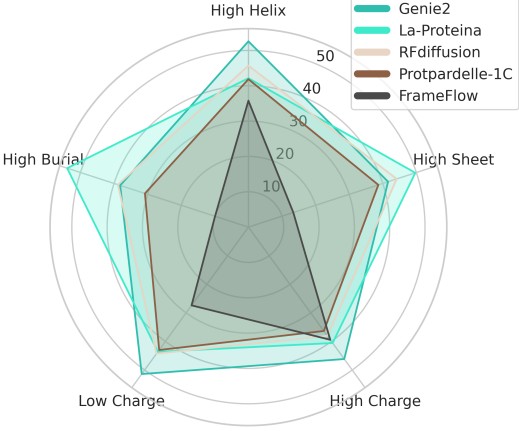

Figure 8: **Model sensitivity to structural properties.** Average SUN scores for structure-based models across five property categories show distinct sensitivity profiles.

significant challenges across all models. Since by design GeomMotif problems contain no more than 25% of loops, tasks with high sheet content match those with low helical content discussed above. This pattern aligns with the inherent complexity of $\beta$-sheets, which require precise long-range hydrogen bonding networks spanning residues distant in sequence.

The number of fragments and their spatial arrangement dramatically influence model performance. For paired motifs, structures with high helical content in both motifs yield significantly higher success rates than mixed secondary structure arrangements. When both motifs are purely helical (e.g. tasks 4_5XJ7, 9_6KFQ), Genie2 vastly outperforms other methods and achieves mean SUN scores of 46.5%, compared to 13.0% for paired motifs of mixed secondary structures.

**Burial and Contact Ratios**  The degree of residue burial and internal contact density provides further insight into model strengths and limitations (Fig. 9). Highly buried single motifs (>60% burial) with low internal-to-external contact ratios (<0.9) prove particularly challenging for all models. For Genie2 and RFdiffusion, the mean SUN score drops below 30% for such motifs, compared to 80.7% for exposed motifs with low burial (<50%) and higher contact ratios (>1.5).

**Model-Specific Property Sensitivities**  Different model architectures exhibit distinct sensitivities to property variations. RFdiffusion demonstrates superior performance on single-fragment tasks with high helical content (mean SUN score 92.0%) but struggles with multi-fragment $\beta$-sheet arrangements (mean SUN score 40.4% for 4+ fragments with >50% $\beta$-sheet content). FrameFlow shows particular strength on paired motifs with three fragments (average of 61.6% SUN), suggesting optimization for distributed spatial constraints. Meanwhile, Genie2 exhibits more balanced performance across property dimensions but shows pronounced weakness on tasks combining high fragment counts with high $\beta$-sheet content and low contact ratios.

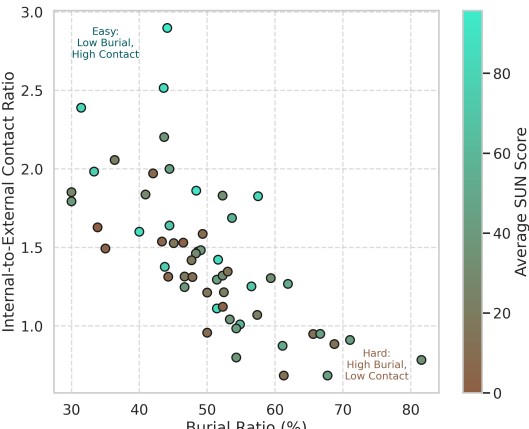

Figure 9: **Task performance correlates with burial and structural context.** GeomMotif tasks colored by average SUN score show that high burial with low internal contacts (lower right) yields harder scaffolding problems than low burial with high contacts (upper left).

These property-performance relationships provide detailed insights for benchmark design and model development. The consistent difficulty with $\beta$-sheet-rich motifs and complex packing environments across diverse architectures points to systemic, field-wide challenges. This approach, therefore, offers a diagnostic framework that complements existing benchmarks and can guide targeted improvements to future protein design models.

## 5 CONCLUSION

GeomMotif establishes the first systematic benchmark for geometric preservation in protein generation. It isolates this fundamental capability from the confounding effects of functional constraints. Our evaluation reveals a clear architectural divide. Structure-based models like Genie2, La-Proteina and RFdiffusion achieve strong geometric fidelity, however, they struggle as spatial complexity increases. Sequence-based approaches fail categorically on spatially separated motifs. This exposes their inability to encode long-range geometric relationships.

These findings provide actionable insights for model development. Helical motifs consistently outperform $\beta$-sheet arrangements. This pronounced sensitivity suggests specific training strategies could address these weaknesses. Our comprehensive property characterization reveals that burial patterns, fragment complexity, and structural context interact in non-obvious ways. These interactions offer concrete targets for architectural improvements.

GeomMotif demonstrates that geometric preservation varies systematically with structural features. This provides the diagnostic precision that functional benchmarks cannot. This complementary perspective is essential as the field advances toward more sophisticated protein design challenges. The benchmark establishes a foundation for understanding generative capabilities that underpin all successful protein engineering. This enables targeted improvements that will ultimately enhance both geometric accuracy and functional success.

## REPRODUCIBILITY STATEMENT

We have thoroughly documented our methodology to ensure the reproducibility of the GeomMotif benchmark's construction, evaluation, and experimental results. All necessary details are provided in the main text and expanded upon in the Appendix.

The GeomMotif construction pipeline is detailed in Sec. 3.2, App. A.1 and illustrated in Fig. 3. We specify our criteria for data filtering, structural quality, redundancy removal, and guaranteed task solvability. The process uses publicly available tools like MMseqs2 (Steinegger & Söding, 2017), TM-Align (Zhang, 2005) and ESMFold (Lin et al., 2023). Methods for calculating the eight characterizing properties for each task are also provided (Sec. 3.3 and App. B).

Our modality-agnostic evaluation framework (Sec. 3.4, Fig. 2) unambiguously defines our metrics for success (motifRMSD, pLDDT), diversity, and novelty. The adapted SUN score is explained in App. C. The evaluation pipeline uses public tools, including ProteinMPNN (Dauparas et al., 2022).

The baseline comparisons in Sec. 4 are reproducible, using publicly available models. Per-task performance data for each model is available in Sec. L.

Benchmark data, task construction scripts, and evaluation code are available at our GitHub, and on HuggingFace.

## ETHICS STATEMENT

This research was conducted in adherence with ethical scientific practices. The GeomMotif benchmark is derived entirely from the Protein Data Bank (PDB), a public and anonymized resource, and our work did not involve human or animal subjects.

We recognize the potential for dual-use applications in protein design. However, our research is foundational and focuses on creating a benchmark for geometric preservation, not on engineering proteins with specific biological functions. The intended purpose of GeomMotif is to advance protein science for beneficial outcomes, such as developing novel therapeutics and biomaterials.

By making our benchmark, code, and evaluation tools publicly available, we aim to foster transparency and support the responsible and collaborative development of protein generation models.

## ACKNOWLEDGEMENTS

This research was supported in part through computational resources of HPC facilities at HSE University. The work was supported by the grant for research centers in the field of AI provided by the Ministry of Economic Development of the Russian Federation in accordance with the agreement 000000C313925P4E0002 and the agreement with HSE University №139-15-2025-009.

## LLM STATEMENT

During the preparation of this manuscript, the authors utilized a Large Language Model (LLM) to assist with editing and refining the language in certain sections. All content was carefully reviewed, edited, and revised by the authors to ensure it accurately reflects our work and conclusions. The final responsibility for the content of this paper rests entirely with the authors.

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

# Appendix

# A    BENCHMARK DESIGN

## A.1    LENGTH RANGE DETERMINATION

To establish biologically plausible length ranges for the variable regions in GeomMotif benchmark, we developed a principled approach based on known structural properties of proteins. For a region of ground truth length $L$, we define the allowable length range as follows:

$$(L_{\min}, L_{\max}) = \begin{cases} (L, L + \max(1, \text{round}(0.3L))), \text{if } L \leq 3 \\ (\max(L - \text{round}(0.3L), 3), L + \text{round}(0.3L)), \text{if } L > 3 \end{cases} \tag{2}$$

where $\text{round}(\cdot)$ denotes rounding to the nearest integer.

For the total protein length, given the sum of fixed motif lengths $L_{\text{motifs}}$, we calculate:

$$L_{\min}^{\text{total}} = L_{\text{motifs}} + \sum_i L_{\min}^i \tag{3}$$

$$L_{\max}^{\text{total}} = \min(250, \max(L_{\text{total}}^{\text{GT}}, L_{\text{motifs}} + \sum_i L_{\max}^i)) \tag{4}$$

where $L_{\min}^i$ and $L_{\max}^i$ are the minimum and maximum lengths for the $i$-th variable region, and $L_{\text{total}}^{\text{GT}}$ is the ground truth total protein length.

GeomMotif benchmark employs a biologically informed approach to determining length ranges for variable regions in protein motif scaffolding tasks. We define ranges that allow approximately $\pm 30\%$ flexibility around the ground truth length, reflecting the natural structural variation observed in homologous proteins while maintaining consistent function. This range corresponds to the typical expansion and compression possible when a protein segment adopts different secondary structures (e.g., $\alpha$-helices vs. $\beta$-strands).

For very short regions ($\leq 3$ residues), we preserve the ground truth length as the minimum to avoid creating structurally impossible constraints, as these short connections often have limited conformational flexibility. For prefix and suffix regions, we apply the same principles to ensure biologically reasonable terminal segments.

The total protein length range is derived from the sum of fixed motif lengths plus the ranges of the variable regions, with an upper bound of 250 residues to ensure compatibility with models of moderate context length. This approach ensures that our benchmark presents challenges that are both structurally reasonable and computationally tractable while maintaining sufficient flexibility to explore diverse design solutions.

# B    DETAILED PROPERTY CHARACTERIZATION

## B.1    PROPERTY CALCULATION METHODS

The characterization of each motif in GeomMotif involves eight properties calculated using established biophysical metrics. Below we detail the precise calculation methods:

## B.2    STRUCTURAL PROPERTIES

### B.2.1    SECONDARY STRUCTURE COMPOSITION

We employ DSSP (Define Secondary Structure of Proteins) to assign secondary structure elements to each residue. The assignment uses the following classification:

- Helical structures: $\alpha$-helices (H), $3_{10}$-helices (G), and $\pi$-helices (I)
- Extended conformations: $\beta$-strands (E) and $\beta$-bridges (B)

- Loop regions: turns (T), bends (S), and unstructured coil (C)

The proportion of each category is calculated as:

$$P_{SS} = \frac{N_{SS}}{N_{total}} \tag{5}$$

where $N_{SS}$ is the number of residues with a particular secondary structure type and $N_{total}$ is the total number of residues in the motif.

### B.2.2 MOTIF SIZE

We record both the total residue count in each motif and the number of fragments. Fragment count ranges from 1 (contiguous) to 7 (highly discontinuous) for single motifs, and 3-7 for paired motifs.

### B.2.3 STRUCTURAL CONTEXT RATIO

For each motif, we calculate:

$$R_{context} = \frac{N_{internal}}{N_{external}} \tag{6}$$

where $N_{internal}$ is the number of contacts between residues within the motif and $N_{external}$ is the number of contacts between motif residues and non-motif residues. A contact is defined when any heavy atoms from two residues are within 4.5Å of each other. This ratio indicates how self-contained versus context-dependent a motif is.

## B.3 PHYSICOCHEMICAL PROPERTIES

### B.3.1 HYDROPHOBICITY PROFILE

We use the Eisenberg hydrophobicity scale (Eisenberg et al., 1984) to calculate the mean hydrophobicity:

$$\bar{H} = \frac{1}{N} \sum_{i=1}^{N} H_i \tag{7}$$

where $H_i$ is the hydrophobicity value of residue $i$ and $N$ is the number of residues in the motif. The Eisenberg scale values range from -2.53 (most hydrophilic) to 1.38 (most hydrophobic).

### B.3.2 BURIAL RATIO

We calculate relative solvent accessibility (RSA) using DSSP, which computes the accessible surface area normalized by the maximum possible exposure for each residue type. The burial ratio is:

$$R_{burial} = \frac{N_{RSA<0.2}}{N_{total}} \tag{8}$$

where $N_{RSA<0.2}$ is the number of residues with RSA < 0.2, indicating buried positions.

### B.3.3 HYDROPHOBIC CORE CONTENT

We define hydrophobic core residues as those that are both buried (RSA < 0.2) and hydrophobic (Eisenberg score > 0.5):

$$R_{core} = \frac{N_{buried \cap hydrophobic}}{N_{total}} \tag{9}$$

### B.3.4 CHARGE CHARACTERISTICS

We calculate absolute charge density as:

$$\rho_{charge} = \frac{1}{N} \sum_{i=1}^{N} |q_i| \tag{10}$$

where $q_i$ is the charge of residue $i$ (Arg/Lys: +1, Asp/Glu: -1, others: 0).

## B.4 PROPERTY DISTRIBUTION ACROSS THE BENCHMARK

To ensure comprehensive coverage of protein structural space, we analyzed the distribution of properties across all benchmark tasks:

- Secondary structure composition spans the full range of natural proteins, with helical content ranging from 0% to 100%, $\beta$-sheet content from 0% to 93.3%, and loop content from 0% to 23.3%.

- Motif sizes range from 30 to 75 residues, with fragment counts from 1 to 7, creating a spectrum of geometric constraint complexity.

- Hydrophobicity values range from -0.57 to 0.68 on the Eisenberg scale, covering both highly hydrophilic and hydrophobic motifs.

- Burial ratios span from highly exposed (0.30) to deeply buried (0.82) motifs.

- Charge densities range from 0.03 to 0.57, representing varying degrees of electrostatic complexity.

- Structural context as internal-to-external ratios vary from 0.68 (highly context-dependent) to 2.90 (highly self-contained).

This diversity enables detailed analysis of how these properties correlate with model performance, revealing specific strengths and weaknesses of different architectural approaches to protein generation.

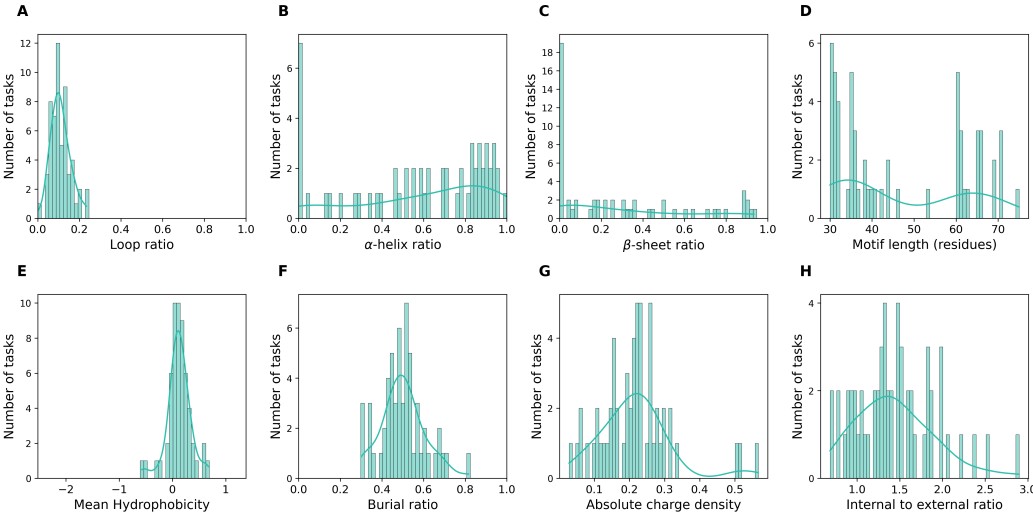

Figure 10: **The distribution of structural and physicochemical properties in GeomMotif tasks.** Histograms show the frequency distribution of (A) loop ratio, (B) $\alpha$-helix ratio, (C) $\beta$-sheet ratio, (D) motif size as number of residues, (E) mean hydrophobicity, (F) burial ratio, (G) absolute charge density, and (H) structural context as internal-to-external contact ratio. The diverse distribution of properties ensures comprehensive coverage of protein structural space and presents varied geometric preservation challenges.

Table 2: **The properties of protein fragments across different GeomMotif tasks**

| Number of Fragments | Motif | Task | Loop (%) | Helix (%) | Sheet (%) | Motif(s) Length | Mean Hydrophobicity | Burial (%) | Abs. Charge Density | Internal to External Contact ratio |
|---|---|---|---|---|---|---|---|---|---|---|
| 1 | single | 1_5OJ8 | 12.9 | 87.1 | 0.0 | 31 | 0.10 | 48.4 | 0.23 | 1.86 |
| 1 | single | 2_1TKY | 23.3 | 46.7 | 30.0 | 30 | 0.27 | 46.7 | 0.20 | 1.25 |
| 1 | single | 3_5XJ7 | 5.9 | 94.1 | 0.0 | 34 | 0.57 | 44.1 | 0.06 | 2.90 |
| 1 | single | 4_6KFQ | 9.7 | 90.3 | 0.0 | 31 | 0.35 | 51.6 | 0.16 | 1.42 |
| 1 | single | 5_5URP | 22.9 | 77.1 | 0.0 | 35 | -0.12 | 31.4 | 0.31 | 2.39 |
| 2 | single | 6_5XJ7 | 10.0 | 90.0 | 0.0 | 30 | 0.22 | 33.3 | 0.23 | 1.98 |
| 2 | single | 7_5OJ8 | 7.7 | 92.3 | 0.0 | 39 | 0.15 | 43.6 | 0.13 | 2.52 |
| 2 | single | 8_1M2G | 12.5 | 68.8 | 18.8 | 32 | -0.11 | 59.4 | 0.31 | 1.30 |
| 2 | single | 9_5CWP | 0.0 | 100.0 | 0.0 | 30 | -0.57 | 40.0 | 0.57 | 1.60 |
| 2 | single | 10_6FFV | 16.7 | 83.3 | 0.0 | 30 | 0.35 | 30.0 | 0.10 | 1.85 |
| 3 | single | 11_1Z6N | 13.9 | 86.1 | 0.0 | 36 | -0.04 | 44.4 | 0.22 | 1.64 |
| 3 | single | 12_3P2W | 6.2 | 78.1 | 15.6 | 32 | 0.22 | 43.8 | 0.16 | 1.38 |
| 3 | single | 13_6KFQ | 8.1 | 91.9 | 0.0 | 37 | 0.35 | 48.6 | 0.11 | 1.47 |
| 3 | single | 14_4BJI | 12.5 | 70.0 | 17.5 | 40 | 0.02 | 57.5 | 0.25 | 1.83 |
| 3 | single | 15_1A2J | 5.7 | 94.3 | 0.0 | 35 | 0.19 | 51.4 | 0.23 | 1.11 |
| 3 | paired | 1_5CWP | 6.1 | 93.9 | 0.0 | 66 | -0.50 | 36.4 | 0.52 | 2.06 |
| 3 | paired | 2_5CWN | 9.5 | 90.5 | 0.0 | 63 | -0.30 | 44.4 | 0.51 | 2.00 |
| 4 | single | 16_3PR9 | 10.0 | 13.3 | 76.7 | 30 | 0.30 | 30.0 | 0.20 | 1.79 |
| 4 | single | 17_4GVW | 13.6 | 61.4 | 25.0 | 44 | 0.25 | 52.3 | 0.16 | 1.83 |
| 4 | single | 18_4LQ4 | 16.1 | 51.6 | 32.3 | 31 | 0.16 | 54.8 | 0.26 | 1.01 |
| 4 | single | 19_1M2G | 15.6 | 34.4 | 50.0 | 32 | 0.06 | 65.6 | 0.22 | 0.95 |
| 4 | single | 20_3L86 | 19.5 | 58.5 | 22.0 | 41 | -0.00 | 53.7 | 0.15 | 1.69 |
| 4 | paired | 3_4K46 | 19.7 | 54.5 | 25.8 | 66 | 0.14 | 40.9 | 0.26 | 1.84 |
| 4 | paired | 4_5XJ7 | 4.2 | 95.8 | 0.0 | 71 | 0.60 | 43.7 | 0.06 | 2.20 |
| 4 | paired | 5_5OJ8 | 11.6 | 88.4 | 0.0 | 69 | 0.04 | 42.0 | 0.19 | 1.97 |
| 4 | paired | 6_2ZE5 | 8.3 | 86.7 | 5.0 | 60 | 0.06 | 48.3 | 0.23 | 1.46 |
| 4 | paired | 7_1DEX | 8.2 | 83.6 | 8.2 | 61 | 0.10 | 57.4 | 0.20 | 1.07 |
| 5 | single | 21_1TKY | 9.7 | 29.0 | 61.3 | 31 | -0.03 | 61.3 | 0.26 | 0.68 |
| 5 | single | 22_6TCS | 11.4 | 0.0 | 88.6 | 35 | 0.16 | 54.3 | 0.06 | 0.98 |
| 5 | single | 23_1SGW | 5.7 | 20.0 | 74.3 | 35 | 0.10 | 51.4 | 0.26 | 1.29 |
| 5 | single | 24_6OU0 | 6.7 | 0.0 | 93.3 | 30 | 0.24 | 53.3 | 0.23 | 1.04 |
| 5 | single | 25_4F3H | 5.6 | 38.9 | 55.6 | 36 | 0.10 | 66.7 | 0.19 | 0.95 |
| 5 | paired | 8_1IS1 | 10.0 | 71.7 | 18.3 | 60 | 0.08 | 35.0 | 0.30 | 1.49 |
| 5 | paired | 9_6KFQ | 4.2 | 95.8 | 0.0 | 71 | 0.68 | 45.1 | 0.03 | 1.53 |
| 5 | paired | 10_5DN1 | 16.7 | 47.0 | 36.4 | 66 | 0.06 | 53.0 | 0.26 | 1.35 |
| 5 | paired | 11_6KFQ | 10.7 | 89.3 | 0.0 | 75 | 0.37 | 49.3 | 0.13 | 1.59 |
| 5 | paired | 12_1HU3 | 18.5 | 81.5 | 0.0 | 65 | -0.04 | 33.8 | 0.28 | 1.63 |
| 6 | single | 26_1GIU | 10.5 | 26.3 | 63.2 | 38 | 0.18 | 71.1 | 0.16 | 0.91 |
| 6 | single | 27_4LQ4 | 4.3 | 52.2 | 43.5 | 46 | 0.30 | 56.5 | 0.20 | 1.25 |
| 6 | single | 28_6TCS | 9.7 | 0.0 | 90.3 | 31 | 0.02 | 67.7 | 0.16 | 0.68 |
| 6 | single | 29_2LAO | 12.5 | 37.5 | 50.0 | 32 | 0.11 | 68.8 | 0.22 | 0.88 |
| 6 | single | 30_6TCS | 11.4 | 0.0 | 88.6 | 35 | -0.03 | 54.3 | 0.14 | 0.80 |
| 6 | paired | 13_4BJI | 10.0 | 85.0 | 5.0 | 60 | 0.10 | 43.3 | 0.23 | 1.54 |
| 6 | paired | 14_5KZL | 16.7 | 83.3 | 0.0 | 60 | 0.09 | 50.0 | 0.22 | 1.21 |
| 6 | paired | 15_4LQ4 | 6.6 | 59.0 | 34.4 | 61 | 0.09 | 44.3 | 0.30 | 1.31 |
| 6 | paired | 16_1SGW | 8.2 | 47.5 | 44.3 | 61 | 0.05 | 52.5 | 0.28 | 1.21 |
| 6 | paired | 17_1BOL | 8.3 | 61.7 | 30.0 | 60 | 0.11 | 46.7 | 0.22 | 1.31 |
| 7 | single | 31_1GIU | 13.2 | 15.8 | 71.1 | 38 | 0.44 | 81.6 | 0.08 | 0.78 |
| 7 | single | 32_1GBG | 9.5 | 0.0 | 90.5 | 42 | 0.19 | 61.9 | 0.21 | 1.27 |
| 7 | single | 33_6TCS | 8.3 | 0.0 | 91.7 | 36 | 0.18 | 61.1 | 0.11 | 0.87 |
| 7 | single | 34_6TCS | 11.4 | 0.0 | 88.6 | 44 | 0.05 | 52.3 | 0.11 | 1.32 |
| 7 | single | 35_1A2J | 9.4 | 84.9 | 5.7 | 53 | 0.20 | 49.1 | 0.23 | 1.48 |
| 7 | paired | 18_6W5B | 14.5 | 4.8 | 80.6 | 62 | -0.21 | 50.0 | 0.34 | 0.96 |
| 7 | paired | 19_1Q0S | 13.0 | 78.3 | 8.7 | 69 | -0.02 | 47.8 | 0.29 | 1.31 |
| 7 | paired | 20_4GVW | 9.2 | 55.4 | 35.4 | 65 | 0.21 | 47.7 | 0.22 | 1.42 |
| 7 | paired | 21_3OSX | 15.4 | 63.1 | 21.5 | 65 | 0.22 | 52.3 | 0.15 | 1.12 |
| 7 | paired | 22_1Z6N | 12.7 | 70.4 | 16.9 | 71 | 0.01 | 46.5 | 0.27 | 1.53 |

## C  INTERPRETATION AND UTILITY OF THE SUN SCORE

The SUN (Successful, Unique, Novel) score is a holistic metric we adapt from generative modeling in materials science (Sriram et al., 2024). To calculate the score, we first identify all **S**uccessful designs—those that meet the primary geometric and structural constraints. Then, conditioned only on this successful set, we assess the **U**niqueness (structural diversity) and **N**ovelty of the generated solutions. This conditional approach ensures that diversity and novelty are measured only for viable designs, a methodological practice consistent with established evaluation protocols in protein generation (Yim et al., 2024).

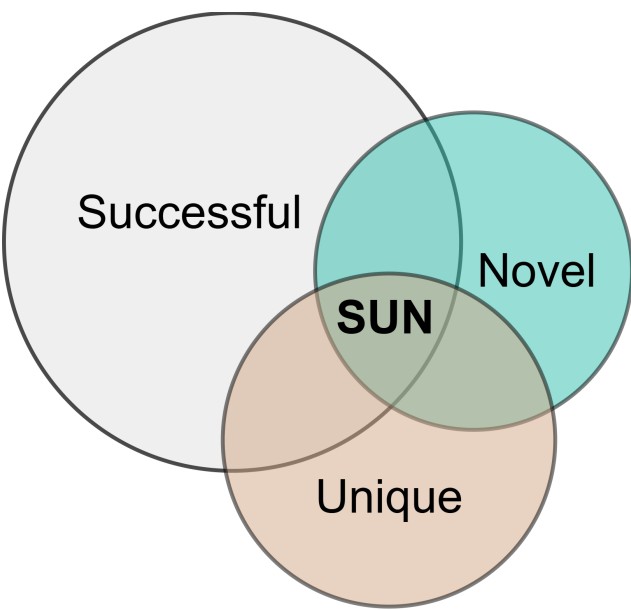

Figure 11: **Conceptual breakdown of the SUN score components.** The SUN score quantifies the proportion of generated designs that simultaneously satisfy the criteria for being **Successful** (meeting geometric and structural constraints), **Unique** (structurally distinct from other generated designs), and **Novel** (structurally distinct from known natural proteins). The final SUN score is represented by the intersection of all three sets.

The SUN score represents the aggregate success rate over many generation attempts and is best interpreted as a continuous value, analogous to success rates in high-throughput screening or yield in chemical synthesis. It quantifies the fraction of high-quality candidates a researcher can expect from a generative model.

Practically, the SUN score provides an **upper bound on the experimental success rate**, which helps guide resource allocation in a protein design campaign. For example, a model with a 44% SUN score may yield one promising candidate for every two or three designs generated, while a 4% SUN score suggests a researcher would need to screen roughly 25 candidates to find a single one of comparable quality. This metric is an upper bound because it assesses geometric plausibility—a necessary but insufficient condition for the ultimate functional success that must be confirmed by wet-lab validation.

The primary utility of the SUN score is providing a single, comprehensive value for high-level model comparison. For a more granular **diagnostic analysis**, the metric can be decomposed. By examining the individual **S**uccess, **U**niqueness, and **N**ovelty rates, and analyzing how they vary with the physicochemical properties of the GeomMotif tasks, a much richer picture of model behavior emerges. This detailed breakdown reveals specific architectural strengths and limitations, enabling informed model selection for specific design challenges and providing clear targets for future research.

# D  DETAILED STRUCTURAL CLASSIFICATION OF BENCHMARK TASKS

To further specify the structural diversity of the GeomMotif benchmark, this section provides a more granular breakdown of the included protein architectures. Figure 12 visualizes the distribution of tasks across the CATH hierarchy, extending to the Architecture and Topology Superfamily levels (CATH levels 2 and 3). This level of detail clarifies the specific types of protein folds represented within each of the major structural classes (all-alpha, all-beta, and alpha-beta) that are summarized in Fig. 4.

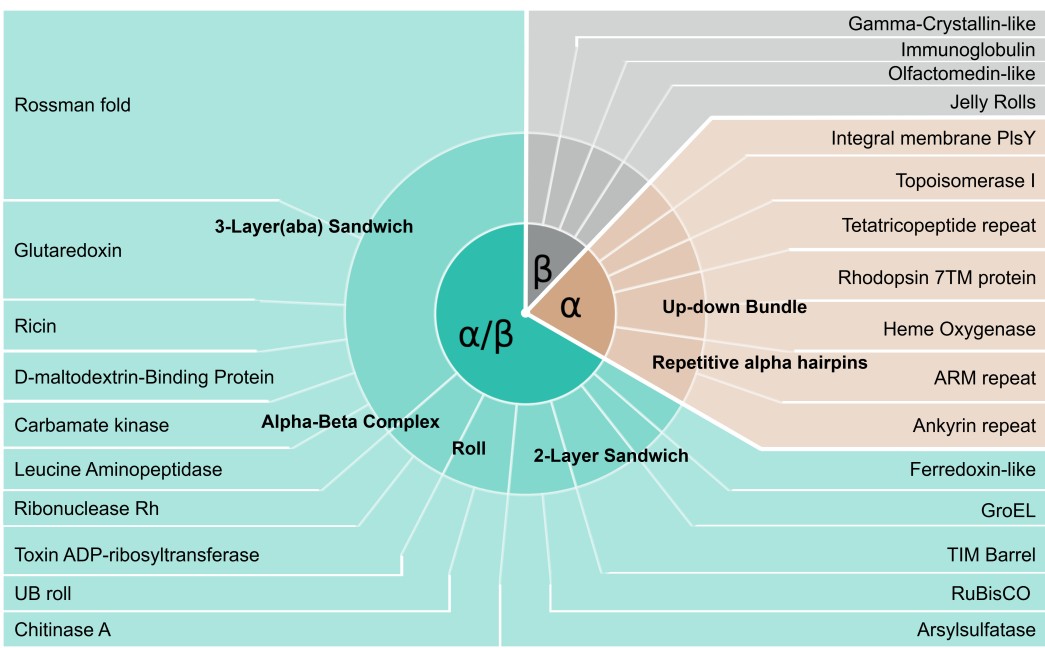

Figure 12: **Detailed CATH Classification of GeomMotif Benchmark Tasks.** The inner ring displays the three major CATH classes: all-alpha ($\alpha$), all-beta ($\beta$), and alpha-beta ($\alpha/\beta$). The outer rings detail the specific Architectures and Topologies within each class.

# E  MODEL CHOICES FOR THE EVALUATION PIPELINE

The construction of a robust and reproducible benchmark relies on a standardized evaluation pipeline. The choice of external software for inverse and forward folding is a critical decision, as any specific tool inevitably introduces systemic biases that can affect performance assessment. Acknowledging this, our selection of **ProteinMPNN** and **ESMFold** was a deliberate decision prioritizing methodological consistency with community standards, the computational feasibility of a large-scale benchmark, and reliance on tools with proven practical utility.

For structure-based models that generate a backbone, we use **ProteinMPNN** to design a corresponding amino acid sequence. This choice aligns our benchmark with established best practices in the field. ProteinMPNN is the canonical inverse folding tool used in the evaluation of leading models like RFdiffusion (Watson et al., 2023), Genie2 (Lin et al., 2024), and FrameFlow (Yim et al., 2024), as well as in the MotifBench benchmark Zheng et al. (2025). Using the same tool is crucial for methodological consistency, as it allows for fair comparisons and isolates the performance of the generative models themselves. Furthermore, the reliability of ProteinMPNN is not just computational as its ability to design high-fidelity sequences has been confirmed through extensive wet-lab experiments de Haas et al. (2024); Sumida et al. (2024); Wang et al. (2024a). Finally, from a practical standpoint, ProteinMPNN is well-suited for our benchmark as it can operate on the C$\alpha$-only backbone traces that some structure-based generators produce.

For the validation step, we predict the structure of designed sequences using **ESMFold**. This choice balances predictive accuracy with the significant computational requirements of a large-scale

benchmark. The GeomMotif evaluation involves folding tens of thousands of sequences, a task for which the speed of ESMFold is essential. Using a more computationally intensive predictor, such as AlphaFold2 (Jumper et al., 2021), would make the benchmark prohibitively expensive and inaccessible for many researchers, whereas ESMFold offers a balance of speed and precision (Hýskova et al., 2025). Our choice is also consistent with established validation pipelines, e.g. MotifBench (Zheng et al., 2025) benchmark. While all predictors have inherent biases, and discrepancies can arise between methods, adhering to a single, widely-adopted standard provides the most stable foundation for comparing models. Most importantly, a cornerstone of GeomMotif's design is ensuring every task is solvable *by the evaluation pipeline itself*. We guarantee this by pre-filtering our source PDBs, keeping only structures that ESMFold can accurately fold (RMSD $\leq 1.0$ Å). This critical step ensures our benchmark rigorously tests the generative model's capabilities, not the limitations of the validation tool.

## F    RATIONALE FOR THRESHOLD SELECTION

The construction and evaluation pipelines of GeomMotif rely on several key numerical thresholds. These values are grounded in established community standards, biophysical principles, and internal validation to ensure our benchmark is robust and interpretable. This section provides a concise justification for these choices.

### F.1    RMSD THRESHOLD FOR GEOMETRIC FIDELITY (1.0 Å)

A Root Mean Square Deviation (RMSD) threshold of 1.0 Å is used for two purposes: first, to select solvable tasks during benchmark construction, and second, to define success in geometric preservation for generated designs. The choice of this value is supported by several lines of evidence:

- **Community Standard.** RMSD threshold of 1.0 Å aligns with the precision demanded by other benchmarks in protein design. For instance, MotifBench employs a 1.0 Å threshold to demand "atomic precision" (Zheng et al., 2025), and the original RFdiffusion benchmark uses identical criteria as "stringent filters indicative of experimental success" validated against experimental outcomes (Watson et al., 2023).

- **Biophysical Plausibility.** This value reflects the upper limit of natural protein flexibility. Analyses of identical protein structures from different PDB entries show inherent RMSD variations up to 1.2 Å, while molecular dynamics simulations indicate that stable regions maintain an RMSD of approximately 1.0 Å (Kufareva & Abagyan, 2012; Maruyama et al., 2023). Our threshold thus ensures that benchmark tasks represent geometrically achievable conformations.

- **Guaranteed Solvability.** By pre-filtering our initial PDB set to include only structures that ESMFold predicts with an RMSD $\leq 1.0$ Å relative to the experimental ground truth, we guarantee that every task in GeomMotif has at least one known solution verifiable by our evaluation pipeline. This addresses a key limitation in prior benchmarks where task solvability was uncertain.

### F.2    TM-SCORE THRESHOLDS FOR STRUCTURAL SIMILARITY (0.5 AND 0.8)

Two distinct TM-score thresholds are employed to assess structural similarity, consistent with established conventions in structural biology.

- **TM-score $\geq$ 0.5 (for Task Construction).** During the construction of the GeomMotif dataset, a TM-score of 0.5 was used to cluster proteins. This is a widely accepted standard for determining if two proteins share the same fold, ensuring that our benchmark samples from a structurally diverse set of protein architectures (Zhang, 2005; Xu & Zhang, 2010).

- **TM-score $\leq$ 0.8 (for Novelty and Uniqueness).** In our evaluation pipeline, a more stringent threshold of 0.8 is used to define both novelty and uniqueness. A TM-score above 0.8 indicates that two proteins are highly likely to belong to the same folding family (Zhang & Zhang, 2025; Xu & Zhang, 2010; Zheng et al., 2024; He et al., 2023). This cutoff allows us to distinguish between genuinely new structural variations and minor deviations from

known or previously generated structures. The robustness of this choice was confirmed through a sensitivity analysis where varying the threshold between 0.7 and 0.9 did not alter the relative performance rankings of the evaluated models.

## F.3 STRUCTURE PREDICTION CONFIDENCE (pLDDT $\geq$ 70)

For a generated design to be deemed "Successful", it must not only preserve the motif geometry (RMSD $< 1.0$ Å) but also form a well-structured scaffold, as indicated by a high prediction confidence score (pLDDT $\geq$ 70) from ESMFold. This dual criterion ensures that the preserved motif is embedded within a physically plausible protein structure, filtering out designs that may be disordered or misfolded. The use of confidence scores as a filter for designability is a standard quality control step in the field that correlates with experimental success rates (Hermosilla et al., 2024; Hýskova et al., 2025).

## F.4 TASK CONSTRUCTION THRESHOLDS

The thresholds used during the construction of GeomMotif tasks were chosen to balance several competing design goals: ensuring tasks are sufficiently complex to be challenging, maintaining broad structural diversity, and keeping them computationally tractable within our framework.

- **Motif Definition (13 Å Radius).** A 13 Å radius was selected to define the spatial extent of a motif. This value was determined empirically to provide an optimal trade-off. Smaller radii (e.g., 10 Å) often captured insufficient structural context, resulting in motifs that were too simple. Conversely, larger radii tended to encompass a disproportionate fraction of the total protein (which is capped at 250 residues), limiting the scope for de novo scaffold generation. The 13 Å radius consistently produced motifs with meaningful tertiary structure while leaving a substantial portion of the protein to be generated.

- **Motif Size Filter ($>$30 Residues).** We excluded motifs containing fewer than 30 residues to establish a baseline of geometric complexity. Preliminary analysis showed that smaller motifs were frequently composed of simple surface features or single contiguous fragments. Setting a minimum size of 30 residues ensures that every task presents a non-trivial challenge, often involving multiple sequence fragments that must be correctly oriented.

- **Paired Motif Separation ($\geq$30 Å).** For paired-motif tasks, a minimum distance of 30 Å between motif centers was required. This ensures that the two motifs are spatially distinct and do not directly interact. This separation forces generative models to construct a substantial and structurally coherent scaffold to bridge the two fixed regions, directly testing their ability to handle long-range geometric constraints. The value was chosen to be safely greater than the sum of two motif radii (13 Å + 13 Å = 26 Å), guaranteeing a clear gap between them.

# G COMPLETE SPECIFICATION OF ALL BENCHMARK TASKS

Table 3: **Comprehensive listing of GeomMotif benchmark tasks.** Complete specification of all 57 benchmark tasks, including interval notation for each task. The notation follows the RFdiffusion contig format: $p_1, m_1, p_2, m_2, ..., m_n, p_{n+1}$, where $p_i$ represents ranges for generated residues and $m_i$ represents motif constraints. Total protein length must fall within the specified range. This rigorous task definition ensures reproducibility and standardization across model evaluations.

| Entry | Contigs | Length |
|---|---|---|
| **Single Motifs** | | |
| 1_5OJ8 | 16-30,A24-54,109-203 | 156-250 |
| 2_1TKY | 12-22,A18-47,124-230 | 166-250 |
| 3_5XJ7 | 83-153,A119-152,28-52 | 145-239 |
| 4_6KFQ | 55-101,A79-109,84-156 | 170-250 |
| 5_5URP | 22-42,A33-67,94-174 | 151-250 |
| 6_5XJ7 | 12-22,A18-26,8-16,A39-59,93-173 | 143-241 |
| 7_5OJ8 | 3-4,A4-32,11-19,A48-57,107-199 | 160-250 |
| 8_1M2G | 29-55,A43-68,57-107,A151-156,65-121 | 183-250 |
| 9_5CWP | 127-235,A182-194,4-8,A201-217,8-16 | 169-250 |
| 10_6FFV | 15-27,A22-31,21-39,A62-81,79-147 | 145-243 |
| 11_1Z6N | 22-40,A32-50,16-30,A74-83,46-86,A150-156,7-13 | 127-205 |
| 12_3P2W | 0-1,A1-13,107-199,A167-175,22-42,A208-217,2-3 | 163-250 |
| 13_6KFQ | 13-23,A19-27,3-4,A31-39,118-220,A209-227,2-3 | 173-250 |
| 14_4BJI | 18-34,A27-52,6-10,A61-70,8-14,A82-85,74-138 | 146-236 |
| 15_1A2J | 46-86,A67-79,5-9,A87-96,5-9,A104-115,51-95 | 142-234 |
| 16_3PR9 | 58-108,A84-93,9-17,A107-111,2-3,A114-122,2-3,A125-130,14-26 | 115-187 |
| 17_4GVW | 14-26,A21-23,3-5,A28-31,8-14,A43-62,13-25,A82-98,66-122 | 148-236 |
| 18_4LQ4 | 41-75,A59-63,10-18,A78-80,7-13,A91-106,8-14,A118-124,61-113 | 158-250 |
| 19_1M2G | 43-81,A63-74,27-51,A114-123,4-8,A130-133,13-23,A152-157,64-120 | 183-250 |
| 20_3L86 | 9-17,A14-29,3-7,A35-37,30-56,A81-98,28-52,A139-142,72-134 | 183-250 |
| 21_1TKY | 55-101,A79-81,14-26,A102-108,31-57,A153-156,30-56,A200-208,3-5,A213-220,3-5 | 167-250 |
| 22_6TCS | 3-4,A4-11,5-9,A19-25,8-14,A37-39,19-35,A67-79,7-13,A90-93,97-181 | 174-250 |
| 23_1SGW | 2-3,A3-11,3-4,A15-24,12-22,A42-44,6-12,A54-58,94-174,A193-200,0-1 | 152-250 |
| 24_6OU0 | 3-5,A5-13,122-226,A188-191,11-19,A207-211,10-18,A226-230,4-8,A237-243,1-2 | 181-250 |
| 25_4F3H | 13-25,A20-26,8-16,A39-46,3-5,A51-54,11-19,A70-82,8-14,A94-97,105-195 | 184-250 |
| 26_1GIU | 0-1,A1-5,8-14,A17-24,15-29,A47-53,3-7,A59-66,3-4,A70-76,56-104,A157-159,62-114 | 185-250 |
| 27_4LQ4 | 1-2,A2-10,11-19,A26-30,83-153,A149-152,5-9,A160-166,3-4,A170-186,8-16,A199-202,6-12 | 163-250 |
| 28_6TCS | 90-168,A130-137,5-9,A145-149,8-16,A162-163,12-22,A181-185,2-3,A188-193,8-14,A205-209,16-30 | 172-250 |
| 29_2LAO | 63-117,A91-95,13-25,A115-118,1-2,A120-123,24-44,A158-170,6-12,A180-181,1-2,A183-186,36-68 | 176-250 |
| 30_6TCS | 12-22,A18-24,8-16,A37-41,16-30,A65-71,2-3,A74-80,6-10,A89-93,8-14,A105-108,87-161 | 174-250 |
| 31_1GIU | 0-1,A1-4,29-55,A47-53,3-7,A59-66,3-4,A70-77,1-2,A79-83,48-90,A153-155,1-2,A157-159,62-114 | 185-250 |
| 32_1GBG | 50-94,A73-77,9-17,A91-92,9-17,A106-109,6-10,A118-122,6-10,A131-134,8-14,A146-152,2-3,A155-169,31-59 | 163-250 |
| 33_6TCS | 89-165,A128-134,8-16,A147-150,6-12,A160-161,12-22,A179-184,3-5,A189-195,5-9,A203-207,12-22,A225-229,3-4 | 174-250 |
| 34_6TCS | 3-4,A4-13,3-5,A18-24,9-17,A38-41,17-31,A66-70,3-4,A74-80,5-9,A88-93,8-14,A105-109,86-160 | 178-250 |
| 35_1A2J | 40-74,A58-61,3-7,A67-85,1-2,A87-89,1-2,A91-93,7-13,A104-113,5-9,A121-127,2-3,A130-136,36-68 | 148-231 |
| **Paired Motifs** | | |
| 1_5CWP | 10-18,A15-45,20-38,A75-77,73-135,A182-213,11-21 | 180-250 |
| 2_5CWN | 5-9,A8-37,6-12,A47-57,21-39,A88-109,69-127 | 164-250 |
| 3_4K46 | 1-2,A2-7,16-30,A31-64,27-51,A104-110,47-87,A178-196,13-23 | 170-250 |
| 4_5XJ7 | 0-1,A1-14,33-61,A62-74,3-5,A79-88,21-39,A119-152,28-52 | 156-229 |
| 5_5OJ8 | 3-4,A4-32,11-19,A48-57,67-125,A154-180,12-22,A198-200,7-13 | 169-250 |
| 6_2ZE5 | 50-94,A73-82,8-16,A95-114,9-17,A128-144,39-71,A200-212,11-19 | 177-250 |
| 7_1DEX | 30-56,A44-58,32-60,A105-120,34-62,A169-188,7-13,A199-208,17-33 | 181-250 |
| 8_1IS1 | 13-23,A19-21,13-23,A40-51,11-19,A67-84,20-38,A114-130,25-47,A167-176,6-12 | 148-222 |
| 9_6KFQ | 3-7,A6-22,11-19,A38-51,41-75,A110-139,27-49,A178-184,17-31,A209-211,13-23 | 183-250 |
| 10_5DN1 | 5-9,A8-20,20-36,A49-71,40-74,A129-136,6-10,A145-157,16-30,A181-189,36-66 | 189-250 |
| 11_6KFQ | 1-2,A2-18,15-29,A41-54,19-35,A82-97,32-60,A144-168,22-42,A201-203,18-34 | 182-250 |
| 12_1HU3 | 7-13,A11-21,3-5,A26-34,20-36,A63-73,27-49,A112-124,14-26,A145-165,22-40 | 158-234 |
| 13_4BJI | 17-31,A25-39,11-19,A55-69,22-40,A101-103,22-40,A135-139,11-21,A156-170,7-13,A181-187,3-5 | 153-229 |
| 14_5KZL | 1-2,A2-11,15-27,A33-47,23-43,A81-86,7-13,A97-100,48-90,A170-189,8-14,A201-205,0-1 | 162-250 |
| 15_4LQ4 | 18-34,A27-33,31-59,A79-99,10-18,A114-123,24-44,A158-164,11-21,A181-186,4-8,A193-202,6-12 | 165-250 |
| 16_1SGW | 1-2,A2-10,4-8,A17-25,20-38,A55-66,13-25,A86-96,4-8,A103-114,13-23,A133-140,42-78 | 158-243 |
| 17_1BOL | 50-92,A72-91,27-49,A130-139,13-25,A159-171,4-8,A178-183,1-2,A185-190,17-31,A215-219,3-4 | 175-250 |
| 18_6W5B | 1-2,A2-19,9-17,A33-38,3-4,A42-45,21-39,A76-79,6-10,A88-104,10-18,A119-127,1-2,A129-132,27-49 | 140-203 |
| 19_1Q0S | 37-69,A54-71,19-35,A99-105,23-43,A139-148,13-25,A168-171,10-18,A186-202,3-5,A207-210,6-12,A220-228,9-17 | 189-250 |
| 20_4GVW | 0-1,A1-7,4-8,A14-19,17-31,A44-66,10-18,A81-88,4-8,A95-98,51-95,A172-176,1-2,A178-189,3-4 | 155-232 |
| 21_3OSX | 15-27,A22-26,21-39,A57-60,6-12,A70-87,7-13,A98-108,17-33,A134-137,8-14,A149-164,10-18,A179-185,3-7 | 152-228 |
| 22_1Z6N | 2-3,A3-12,13-25,A32-50,6-10,A59-63,7-13,A74-83,3-7,A89-104,5-9,A112-115,24-44,A150-156,7-13 | 138-195 |

# H   FIXED-LENGTH EVALUATION

To distinguish between genuine geometric understanding and potential memorization artifacts, we conduct our evaluation in two complementary settings. The primary variable-length evaluation presented in the main text Sec. 4 allows biologically reasonable length variations for scaffold regions while preserving fixed motif geometry. This reflects realistic protein design scenarios where the total protein length is not predetermined. However, this flexibility introduces a potential confound: models might fail not due to geometric constraints but due to difficulties in determining appropriate scaffold lengths.

Our secondary fixed-length evaluation addresses this concern by constraining generated proteins to match the ground truth length exactly. This controlled setting serves two critical purposes. First, it provides 100% guaranteed solvability since the target length is known to produce a valid fold. Second, it isolates geometric preservation capabilities from length prediction, revealing whether models can solve the scaffolding problem when this additional complexity is removed. Comparing performance across these two settings helps identify whether model limitations stem from fundamental geometric reasoning deficits or from the additional challenge of length optimization. Models that perform well in fixed-length but poorly in variable-length scenarios may benefit from improved length prediction mechanisms, while consistent poor performance across both settings indicates deeper architectural limitations in geometric understanding.

Table 4: **Fixed-length evaluation.** Detailed breakdown of the SUN metric into its constituent components (Success, Novelty, Uniqueness) for single and paired motif tasks.

| **Model** | Successful, % ↑ | | Novel, % ↑ | | Unique, % ↑ | | SUN Score ↑ | |
| | Single | Paired | Single | Paired | Single | Paired | Single | Paired |
| --- | --- | --- | --- | --- | --- | --- | --- | --- |
| Genie2 | 61.6 ± 0.5 | 36.0 ± 0.6 | 61.6 ± 0.5 | **27.8** ± 0.6 | 61.1 ± 0.5 | **18.9** ± 0.3 | 61.1 ± 0.5 | **17.2** ± 0.3 |
| RFdiffusion | **76.4** ± 0.6 | **54.0** ± 0.4 | **76.4** ± 0.6 | 27.2 ± 0.5 | **70.9** ± 0.5 | 11.7 ± 0.2 | **70.9** ± 0.5 | 7.1 ± 0.2 |
| FrameFlow | 33.0 ± 0.7 | 34.3 ± 0.7 | 33.0 ± 0.7 | 22.4 ± 0.8 | 33.0 ± 0.7 | 27.1 ± 0.6 | 33.0 ± 0.7 | 17.8 ± 0.7 |
| DPLM-3B | 46.3 ± 0.6 | 47.2 ± 0.9 | 18.9 ± 0.2 | 0.0 ± 0.0 | 9.3 ± 0.2 | 0.0 ± 0.0 | 6.8 ± 0.1 | 0.0 ± 0.0 |
| DPLM-650M | 40.2 ± 0.3 | 36.7 ± 0.3 | 15.3 ± 0.3 | 0.6 ± 0.1 | 10.2 ± 0.2 | 0.5 ± 0.0 | 5.7 ± 0.1 | 0.1 ± 0.0 |
| ESM3 (seq) | 42.2 ± 0.3 | 36.5 ± 0.4 | 16.5 ± 0.4 | 0.9 ± 0.2 | 21.9 ± 0.2 | 0.6 ± 0.1 | 8.2 ± 0.3 | 0.5 ± 0.1 |
| ESM3 (seq & struct) | 70.2 ± 0.2 | 68.4 ± 0.4 | 27.4 ± 0.4 | 2.7 ± 0.2 | 42.7 ± 0.1 | 5.2 ± 0.1 | 17.1 ± 0.3 | 1.4 ± 0.1 |

In the fixed-length experiment, providing the model with the exact ground-truth protein length simplifies the generative task. This controlled setting reveals distinct behaviors between model architectures.

The performance of **structure-based models** shows a commensurate increase in both raw Success rates and final SUN scores. This parallel improvement indicates that the performance gain is not simply an artifact of memorization but rather a result of isolating the geometric scaffolding challenge from the separate complexity of length prediction.

In contrast, **sequence-based models** exhibit a large increase in their Success rates but only a modest improvement in their SUN scores. This pattern suggests that while these models are proficient at completing a sequence within a known length constraint, they struggle to generate novel and diverse structures, highlighting a limitation in their generalization capabilities.

This setting also reveals the potential of multimodal models. For instance, the **ESM3 (seq & struct)** model, which performed poorly in the variable-length experiment, shows a substantial improvement in its SUN score under fixed-length conditions, increasing from 1.4% to 9.25%. This suggests that providing accurate structural constraints can significantly enhance the performance of models that leverage both sequence and structure information.

# I    BENCHMARKING TIME EVALUATION

To evaluate the computational requirements of running GeomMotif depending on model type, we run the evaluation pipeline 5 times to calculate standard deviations. The results of the evaluation using 8 NVIDIA A100 GPUs are reported in Tab. 5.

Table 5: **Computational requirements by model type.**

| Model Type | Total Time (hours) | ProteinMPNN | scRMSD calc. | pLDDT calc. | Parsing |
|---|---|---|---|---|---|
| Sequence-based | 0.77 $\pm$ 0.01 | - | - | 0.64 $\pm$ 0.01 | 0.12 $\pm$ 0.01 |
| Structure-based | 6.03 $\pm$ 0.01 | 0.64 $\pm$ 0.01 | 4.93 $\pm$ 0.01 | - | 0.44 $\pm$ 0.01 |

Structure-based model evaluation requires approximately 8× longer than sequence-based models due to the additional ProteinMPNN sequence design step, which necessitates 8 sequences per generated structure followed by ESMFold prediction for each. Novelty and diversity calculations are identical for both model types and require negligible additional time (under 10 minutes total). We will include comprehensive computational requirements and runtime analysis in the revised appendix to facilitate benchmark reproduction.

# J  FAIRNESS IN COMPARING STRUCTURE-BASED AND SEQUENCE-BASED MODELS

A potential concern in our evaluation is that structure-based models utilize 8 ProteinMPNN sequences per generated backbone, while sequence-based models generate only one sequence per sample. This difference in the number of ESMFold predictions could be perceived as providing an advantage to structure-based approaches. To address this concern and ensure a fair comparison, we conducted additional experiments examining performance under two alternative settings: (1) structure-based models using only 1 ProteinMPNN sequence per backbone, and (2) sequence-based models generating 8× the number of sequences.

Table 6: **Impact of sampling budget on model performance.** Comparison of SUN scores when varying the number of samples: structure-based models with 1 vs. 8 ProteinMPNN sequences, and sequence-based models with 1× vs. 8× sampling. Additional compute improves all models but does not change the fundamental performance hierarchy.

| Model | Successful, % ↑ | | Novel, % ↑ | | Unique, % ↑ | | SUN Score ↑ | | |
|---|---|---|---|---|---|---|---|---|---|
| | Single | Paired | Single | Paired | Single | Paired | Single | Paired | Overall |
| DPLM-650M x1 | 15.9 ± 0.4 | 6.5 ± 0.5 | 8.7 ± 0.2 | 0.4 ± 0.1 | 8.1 ± 0.3 | 0.3 ± 0.1 | 4.0 ± 0.2 | 0.2 ± 0.1 | 2.1 ± 0.1 |
| DPLM-650M x8 | 19.2 ± 0.2 | 8.1 ± 0.2 | 11.9 ± 0.3 | 1.6 ± 0.1 | 10.9 ± 0.2 | 4.2 ± 0.2 | 7.2 ± 0.3 | 1.4 ± 0.1 | 4.3 ± 0.2 |
| ESM3 (seq) x1 | 17.4 ± 0.3 | 6.5 ± 0.5 | 11.3 ± 0.5 | 0.1 ± 0.0 | 10.1 ± 0.2 | 0.1 ± 0.0 | 6.8 ± 0.3 | 0.1 ± 0.0 | 3.5 ± 0.2 |
| ESM3 (seq) x8 | 24.7 ± 0.3 | 9.7 ± 0.5 | 18.7 ± 0.4 | 3.0 ± 0.4 | 17.0 ± 0.3 | 8.3 ± 0.5 | 13.5 ± 0.4 | 2.8 ± 0.4 | 8.2 ± 0.4 |
| RFdiffusion x1 | 40.5 ± 0.7 | 26.5 ± 0.6 | 40.5 ± 0.7 | 13.1 ± 0.3 | 39.2 ± 0.6 | 13.1 ± 0.3 | 39.2 ± 0.6 | 7.4 ± 0.2 | 23.3 ± 0.4 |
| RFdiffusion x8 | 65.1 ± 0.4 | 43.7 ± 1.0 | 65.1 ± 0.4 | 25.0 ± 0.5 | 62.4 ± 0.4 | 20.5 ± 0.7 | 62.4 ± 0.4 | 13.2 ± 0.4 | 37.8 ± 0.4 |
| Genie2 x1 | 36.2 ± 0.7 | 20.3 ± 0.2 | 36.2 ± 0.7 | 15.9 ± 0.3 | 36.2 ± 0.7 | 14.2 ± 0.1 | 36.2 ± 0.7 | 11.1 ± 0.2 | 23.7 ± 0.5 |
| Genie2 x8 | 60.1 ± 1.0 | 32.9 ± 0.4 | 60.1 ± 1.0 | 26.6 ± 0.5 | 59.9 ± 1.0 | 22.5 ± 0.3 | 59.9 ± 1.0 | 18.8 ± 0.4 | 39.4 ± 0.7 |

Table 6 presents the results of this analysis. When structure-based models are restricted to a single ProteinMPNN sequence (×1), their SUN scores decrease substantially—RFdiffusion drops from 37.8% to 23.3%, and Genie2 from 39.4% to 23.7%. However, even under this constraint, structure-based models still dramatically outperform sequence-based models. Conversely, when sequence-based models are provided with 8× computational budget (×8), their performance improves—ESM3 (seq) increases from 3.5% to 8.2%, and DPLM-650M from 2.1% to 4.3%—but the fundamental performance gap remains substantial. These results demonstrate that additional sampling capacity benefits all model types, but does not eliminate the architectural advantages of structure-based approaches for geometric preservation tasks. Our primary evaluation using 8 ProteinMPNN sequences follows the established protocol in motif scaffolding benchmarks (Watson et al., 2023; Yim et al., 2024; Lin et al., 2024; Zheng et al., 2025), which reflects standard practice in the field and enables direct comparison with published results.

## K    DETAILED FRAGMENT COMPLEXITY PERFORMANCE

This section provides the complete performance breakdown underlying the aggregate analysis in Sec. 4.3. Figure 13 shows individual model trajectories across fragment complexity, and Table 7 provides the detailed numerical results for all models and tasks.

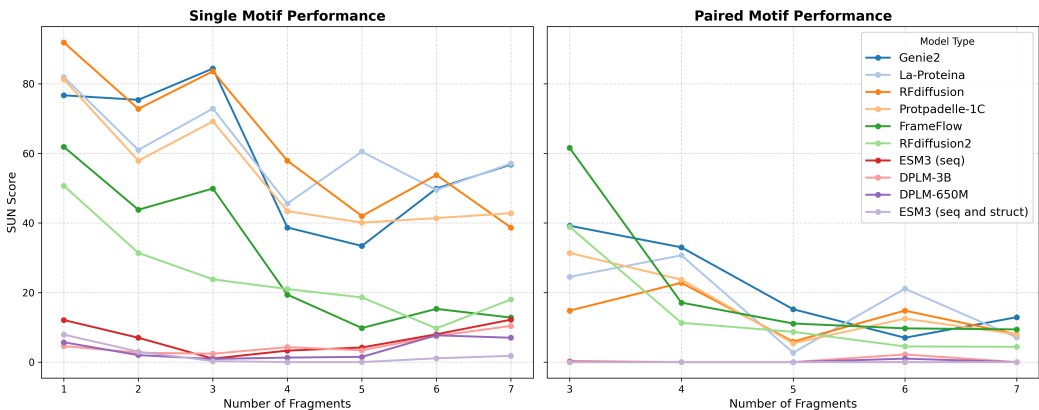

Figure 13: **Individual model performance by fragment complexity.** SUN scores for each model across single-motif (left) and paired-motif (right) tasks stratified by number of continuous fragments. Each bin contains 5 tasks.

Table 7: **SUN score performance stratified by the number of continuous fragments in motifs** (1-7 fragments) reveals non-monotonic relationships between fragment complexity and model performance. Structure-based models exhibit different performance patterns compared to sequence-based models, with the latter failing categorically on paired motifs. Bold values indicate best performance per fragment count category.

| Model | Number of fragments in **single** motif | | | | | | | Number of fragments in **paired** motifs | | | | |
|---|---|---|---|---|---|---|---|---|---|---|---|---|
| | 1 | 2 | 3 | 4 | 5 | 6 | 7 | 3 | 4 | 5 | 6 | 7 |
| Genie2 | 76.7 ± 0.5 | **75.4** ± 1.1 | **84.4** ± 1.6 | 38.7 ± 1.3 | 33.4 ± 2.8 | 49.9 ± 2.3 | 56.8 ± 0.7 | 39.2 ± 1.2 | **33.0** ± 0.7 | **15.2** ± 0.7 | 7.0 ± 0.8 | **12.9** ± 1.2 |
| La-Proteina | 82.0 ± 0.7 | 61.0 ± 1.8 | 72.9 ± 0.6 | 45.6 ± 2.3 | **60.5** ± 2.1 | 49.5 ± 1.4 | **57.1** ± 1.1 | 24.5 ± 1.0 | 30.7 ± 1.2 | 2.7 ± 0.1 | **21.1** ± 1.1 | 7.1 ± 0.3 |
| RFdiffusion | **91.9** ± 0.7 | 72.8 ± 1.9 | 83.6 ± 0.8 | **57.9** ± 2.2 | 42.0 ± 2.8 | **53.8** ± 1.1 | 38.7 ± 0.8 | 14.8 ± 1.6 | 22.8 ± 1.3 | 5.9 ± 0.2 | 14.8 ± 0.2 | 8.0 ± 0.4 |
| Protpadelle-1C | 81.3 ± 1.5 | 57.9 ± 0.7 | 69.2 ± 1.5 | 43.4 ± 1.7 | 40.1 ± 1.8 | 41.4 ± 2.3 | 42.8 ± 2.0 | 31.4 ± 0.8 | 23.7 ± 1.4 | 5.4 ± 0.5 | 12.5 ± 1.3 | 8.0 ± 0.6 |
| FrameFlow | 61.9 ± 1.8 | 43.8 ± 1.2 | 49.9 ± 2.4 | 19.4 ± 0.8 | 9.8 ± 0.9 | 15.3 ± 1.5 | 12.8 ± 1.0 | **61.6** ± 1.8 | 17.1 ± 1.7 | 11.1 ± 0.9 | 9.7 ± 1.4 | 9.4 ± 0.9 |
| RFdiffusion2 | 50.7 ± 1.4 | 31.4 ± 1.9 | 23.8 ± 1.4 | 21.0 ± 2.0 | 18.6 ± 1.8 | 9.7 ± 1.0 | 18.0 ± 1.2 | 38.9 ± 1.7 | 11.3 ± 3.0 | 8.7 ± 0.6 | 4.5 ± 0.2 | 4.4 ± 0.8 |
| ESM3 (seq) | 12.1 ± 1.0 | 7.0 ± 0.7 | 1.0 ± 0.1 | 3.3 ± 0.7 | 4.2 ± 0.8 | 8.0 ± 0.9 | 12.2 ± 1.0 | 0.2 ± 0.2 | 0.0 ± 0.0 | 0.0 ± 0.0 | 0.0 ± 0.0 | 0.0 ± 0.0 |
| DPLM-3B | 4.6 ± 0.4 | 2.7 ± 0.3 | 2.4 ± 0.4 | 4.3 ± 0.6 | 3.4 ± 0.8 | 7.5 ± 1.0 | 10.4 ± 0.6 | 0.0 ± 0.0 | 0.0 ± 0.0 | 0.0 ± 0.0 | 2.2 ± 0.4 | 0.0 ± 0.0 |
| DPLM-650M | 5.7 ± 0.7 | 2.0 ± 0.3 | 0.9 ± 0.3 | 1.3 ± 0.4 | 1.5 ± 0.3 | 7.7 ± 1.2 | 7.0 ± 0.6 | 0.0 ± 0.0 | 0.0 ± 0.0 | 0.0 ± 0.0 | 1.0 ± 0.3 | 0.0 ± 0.0 |
| ESM3 (seq&struct) | 7.9 ± 0.5 | 3.0 ± 0.6 | 0.3 ± 0.2 | 0.0 ± 0.0 | 0.0 ± 0.0 | 1.1 ± 0.7 | 1.8 ± 0.7 | 0.0 ± 0.0 | 0.0 ± 0.0 | 0.0 ± 0.0 | 0.0 ± 0.0 | 0.0 ± 0.0 |

These detailed results complement the aggregated analysis presented in Figure 6, where averaging across models by type reveals the systematic relationship between fragment complexity and performance.

# L    MODEL-SPECIFIC PERFORMANCE ACROSS ALL TASKS

Table 8: **Model-specific performance on single-motif tasks.** Detailed SUN scores for each model across all benchmark tasks, organized by fragment complexity (1-7 fragments). Values represent the percentage of successful, unique, and novel protein structures generated per 100 attempts. This granular performance breakdown reveals task-specific strengths and weaknesses of different architectural approaches.

| #Frags | Entry | Genie2 | La-Proteina | RFdiffusion | Protpardelle-1C | FrameFlow | RFdiffusion2 | ESM3 (seq) | DPLM3B | DPLM650M | ESM3 (seq & struct) |
|---|---|---|---|---|---|---|---|---|---|---|---|
| 1 | 1_5OJ8 | 100.0 ± 0.0 | 96.0 ± 0.0 | 98.0 ± 0.0 | 82.5 ± 1.5 | 81.4 ± 3.3 | 76.4 ± 1.0 | 18.1 ± 0.2 | 11.0 ± 0.0 | 11.0 ± 0.0 | 39.2 ± 2.5 |
|  | 2_1TKY | 5.4 ± 0.8 | 44.6 ± 3.9 | 81.6 ± 1.7 | 47.8 ± 2.6 | 19.8 ± 4.5 | 2.0 ± 1.1 | 24.1 ± 3.9 | 4.1 ± 0.4 | 11.7 ± 1.1 | 2.4 ± 1.9 |
|  | 3_5XJ7 | 100.0 ± 0.0 | 91.0 ± 0.8 | 99.0 ± 0.0 | 99.0 ± 0.9 | 89.6 ± 2.6 | 85.0 ± 2.0 | 5.8 ± 1.4 | 3.0 ± 0.9 | 5.3 ± 1.7 | 0.0 ± 0.0 |
|  | 4_6KFQ | 96.0 ± 0.9 | 82.7 ± 5.2 | 99.4 ± 0.8 | 93.2 ± 2.2 | 59.2 ± 5.6 | 51.2 ± 6.9 | 11.5 ± 2.0 | 3.7 ± 1.7 | 1.7 ± 1.7 | 0.0 ± 0.0 |
|  | 5_5URP | 85.0 ± 2.8 | 97.6 ± 0.5 | 83.0 ± 2.1 | 83.8 ± 3.4 | 58.6 ± 2.7 | 45.6 ± 6.4 | 0.0 ± 0.0 | 0.0 ± 0.0 | 0.0 ± 0.0 | 0.0 ± 0.0 |
| 2 | 6_5XJ7 | 86.2 ± 4.4 | 85.1 ± 2.4 | 89.0 ± 4.4 | 85.0 ± 4.2 | 36.8 ± 5.6 | 16.0 ± 2.4 | 32.2 ± 0.5 | 0.4 ± 0.5 | 0.0 ± 0.0 | 3.2 ± 2.2 |
|  | 7_5OJ8 | 99.4 ± 0.5 | 72.3 ± 0.9 | 93.4 ± 1.3 | 76.7 ± 2.7 | 91.6 ± 2.6 | 69.2 ± 3.2 | 4.3 ± 0.8 | 5.7 ± 0.6 | 6.9 ± 1.1 | 0.0 ± 0.0 |
|  | 8_1M2G | 56.0 ± 3.4 | 44.4 ± 4.2 | 46.0 ± 2.4 | 11.6 ± 2.9 | 3.2 ± 1.5 | 10.0 ± 2.5 | 3.9 ± 1.6 | 0.0 ± 0.0 | 0.0 ± 0.0 | 0.0 ± 0.0 |
|  | 9_5CWP | 98.0 ± 1.1 | 86.4 ± 3.1 | 96.6 ± 1.2 | 88.2 ± 1.5 | 69.4 ± 3.2 | 56.2 ± 4.9 | 0.0 ± 0.0 | 5.4 ± 2.0 | 2.0 ± 1.4 | 12.2 ± 2.7 |
|  | 10_6FFV | 45.8 ± 4.8 | 17.0 ± 3.5 | 36.8 ± 3.5 | 26.0 ± 3.7 | 1.6 ± 1.4 | 0.0 ± 0.0 | 0.0 ± 0.0 | 0.0 ± 0.0 | 0.0 ± 0.0 | 0.0 ± 0.0 |
| 3 | 11_1Z6N | 90.0 ± 1.8 | 71.5 ± 1.6 | 59.2 ± 3.0 | 73.9 ± 1.5 | 58.2 ± 4.3 | 35.6 ± 4.1 | 0.0 ± 0.0 | 0.0 ± 0.0 | 0.0 ± 0.0 | 0.0 ± 0.0 |
|  | 12_3P2W | 73.6 ± 3.6 | 88.6 ± 1.1 | 84.3 ± 1.9 | 80.7 ± 2.0 | 50.2 ± 5.7 | 23.4 ± 1.0 | 0.0 ± 0.0 | 0.0 ± 0.0 | 0.0 ± 0.0 | 0.0 ± 0.0 |
|  | 13_6KFQ | 85.0 ± 2.7 | 15.6 ± 1.6 | 85.4 ± 3.2 | 41.0 ± 6.6 | 18.6 ± 2.2 | 13.0 ± 2.3 | 0.0 ± 0.0 | 0.0 ± 0.0 | 0.0 ± 0.0 | 0.0 ± 0.0 |
|  | 14_4BJI | 87.8 ± 1.7 | 93.4 ± 2.3 | 92.6 ± 1.3 | 82.6 ± 1.8 | 59.6 ± 6.7 | 18.8 ± 2.5 | 4.0 ± 1.3 | 12.1 ± 2.2 | 4.2 ± 1.1 | 0.4 ± 0.5 |
|  | 15_1A2J | 93.2 ± 3.1 | 96.4 ± 1.7 | 95.6 ± 1.4 | 74.8 ± 2.0 | 63.8 ± 4.1 | 27.0 ± 6.5 | 0.0 ± 0.0 | 0.6 ± 0.5 | 0.0 ± 0.0 | 0.0 ± 0.0 |
| 4 | 16_3PR9 | 16.6 ± 2.6 | 20.0 ± 4.1 | 69.6 ± 5.2 | 51.6 ± 2.7 | 39.6 ± 8.0 | 34.2 ± 4.4 | 8.6 ± 1.9 | 16.6 ± 1.8 | 3.8 ± 1.2 | 0.0 ± 0.0 |
|  | 17_4GVW | 40.0 ± 2.7 | 47.4 ± 3.8 | 45.1 ± 3.4 | 30.4 ± 4.9 | 7.6 ± 1.7 | 5.6 ± 2.4 | 0.4 ± 0.5 | 0.0 ± 0.0 | 0.0 ± 0.0 | 0.0 ± 0.0 |
|  | 18_4LQ4 | 58.6 ± 3.4 | 59.7 ± 1.9 | 79.1 ± 2.1 | 57.5 ± 3.6 | 35.6 ± 3.7 | 26.2 ± 3.9 | 5.4 ± 0.9 | 3.7 ± 0.8 | 2.2 ± 1.5 | 0.0 ± 0.0 |
|  | 19_1M2G | 11.4 ± 2.4 | 24.8 ± 2.9 | 21.0 ± 3.0 | 4.6 ± 1.2 | 1.8 ± 1.0 | 2.0 ± 0.6 | 0.0 ± 0.0 | 0.0 ± 0.0 | 0.0 ± 0.0 | 0.0 ± 0.0 |
|  | 20_3L86 | 59.8 ± 1.8 | 72.3 ± 3.7 | 65.2 ± 4.7 | 68.0 ± 3.1 | 17.8 ± 3.5 | 26.4 ± 2.7 | 0.0 ± 0.0 | 0.0 ± 0.0 | 0.0 ± 0.0 | 0.0 ± 0.0 |
| 5 | 21_1TKY | 0.8 ± 0.4 | 51.6 ± 5.1 | 4.0 ± 1.7 | 4.4 ± 1.0 | 4.4 ± 1.0 | 0.0 ± 0.0 | 0.0 ± 0.0 | 0.0 ± 0.0 | 0.0 ± 0.0 | 0.0 ± 0.0 |
|  | 22_6TCS | 42.0 ± 2.5 | 57.0 ± 3.5 | 50.8 ± 3.4 | 47.4 ± 5.2 | 13.4 ± 3.8 | 29.4 ± 5.5 | 13.6 ± 2.6 | 12.2 ± 3.3 | 7.5 ± 1.4 | 0.0 ± 0.0 |
|  | 23_1SGW | 23.4 ± 3.9 | 75.1 ± 1.8 | 53.6 ± 1.7 | 62.4 ± 5.6 | 8.2 ± 1.8 | 13.8 ± 3.5 | 0.0 ± 0.0 | 0.0 ± 0.0 | 0.0 ± 0.0 | 0.0 ± 0.0 |
|  | 24_6OU0 | 39.8 ± 2.9 | 73.9 ± 2.9 | 43.6 ± 3.1 | 28.3 ± 3.2 | 4.8 ± 1.6 | 29.4 ± 4.9 | 0.0 ± 0.0 | 0.0 ± 0.0 | 0.0 ± 0.0 | 0.0 ± 0.0 |
|  | 25_4F3H | 58.8 ± 3.4 | 41.0 ± 2.3 | 60.0 ± 3.8 | 48.2 ± 5.1 | 14.6 ± 4.5 | 17.8 ± 2.6 | 7.9 ± 1.7 | 2.5 ± 1.0 | 1.3 ± 0.5 | 0.0 ± 0.0 |
| 6 | 26_1GIU | 41.0 ± 4.1 | 53.8 ± 4.2 | 82.6 ± 2.1 | 26.6 ± 4.0 | 13.8 ± 3.5 | 10.0 ± 4.0 | 0.0 ± 0.0 | 0.0 ± 0.0 | 0.0 ± 0.0 | 0.0 ± 0.0 |
|  | 27_4LQ4 | 84.6 ± 2.0 | 79.6 ± 2.7 | 77.0 ± 1.7 | 70.2 ± 3.8 | 46.4 ± 3.3 | 13.0 ± 1.4 | 0.0 ± 0.0 | 0.0 ± 0.0 | 0.0 ± 0.0 | 1.4 ± 1.0 |
|  | 28_6TCS | 60.2 ± 2.9 | 67.0 ± 4.3 | 51.4 ± 4.3 | 36.8 ± 3.3 | 8.8 ± 2.8 | 13.4 ± 3.1 | 5.7 ± 3.3 | 2.2 ± 1.8 | 1.2 ± 1.2 | 0.0 ± 0.0 |
|  | 29_2LAO | 6.8 ± 1.9 | 32.8 ± 4.0 | 8.4 ± 1.4 | 28.0 ± 5.1 | 0.0 ± 0.0 | 1.8 ± 1.0 | 0.0 ± 0.0 | 0.0 ± 0.0 | 0.0 ± 0.0 | 0.0 ± 0.0 |
|  | 30_6TCS | 56.0 ± 3.6 | 21.2 ± 4.4 | 52.0 ± 2.8 | 48.0 ± 4.1 | 14.6 ± 3.6 | 8.6 ± 3.4 | 36.7 ± 1.1 | 37.2 ± 1.6 | 43.0 ± 2.9 | 6.8 ± 2.0 |
| 7 | 31_1GIU | 27.0 ± 4.6 | 65.0 ± 2.4 | 37.4 ± 3.2 | 27.8 ± 4.4 | 10.0 ± 1.8 | 2.6 ± 1.9 | 0.0 ± 0.0 | 0.0 ± 0.0 | 0.0 ± 0.0 | 0.0 ± 0.0 |
|  | 32_1GBG | 68.2 ± 4.6 | 84.4 ± 3.8 | 31.2 ± 2.7 | 57.9 ± 4.1 | 20.4 ± 3.3 | 29.2 ± 3.1 | 5.1 ± 2.0 | 4.1 ± 2.5 | 0.4 ± 0.5 | 0.0 ± 0.0 |
|  | 33_6TCS | 69.4 ± 3.0 | 65.3 ± 6.1 | 53.6 ± 2.5 | 43.0 ± 2.2 | 20.0 ± 0.6 | 32.6 ± 5.3 | 17.1 ± 3.7 | 17.2 ± 5.4 | 7.2 ± 2.1 | 1.0 ± 1.3 |
|  | 34_6TCS | 69.8 ± 1.0 | 13.6 ± 1.9 | 18.2 ± 3.4 | 47.2 ± 6.1 | 7.6 ± 2.2 | 20.4 ± 3.4 | 28.9 ± 4.8 | 28.6 ± 2.0 | 21.1 ± 2.9 | 8.4 ± 2.9 |
|  | 35_1A2J | 55.8 ± 3.3 | 48.2 ± 5.6 | 44.1 ± 4.6 | 38.1 ± 2.8 | 11.4 ± 2.0 | 10.2 ± 1.5 | 4.6 ± 1.4 | 1.4 ± 0.8 | 2.9 ± 0.7 | 0.0 ± 0.0 |

Table 9: **Model-specific performance on paired-motif tasks.** Detailed SUN scores for each model across all benchmark tasks, organized by fragment complexity (3-7 fragments). Values represent the percentage of successful, unique, and novel protein structures generated per 100 attempts.

| #Frags | Entry | Genie2 | La-Proteina | RFdiffusion | Protpardelle-1C | FrameFlow | RFdiffusion2 | ESM3 (seq) | DPLM3B | DPLM650M | ESM3 (seq & struct) |
|---|---|---|---|---|---|---|---|---|---|---|---|
| 3 | 1_5CWP | 4.2 ± 2.4 | 16.2 ± 0.7 | 13.6 ± 1.8 | 25.6 ± 1.0 | 50.9 ± 3.2 | 19.6 ± 2.6 | 0.0 ± 0.0 | 0.0 ± 0.0 | 0.0 ± 0.0 | 0.0 ± 0.0 |
|  | 2_5CWN | 70.7 ± 2.4 | 31.6 ± 2.2 | 16.4 ± 1.6 | 33.3 ± 1.7 | 70.7 ± 3.9 | 62.0 ± 4.0 | 1.0 ± 1.1 | 0.0 ± 0.0 | 0.0 ± 0.0 | 0.0 ± 0.0 |
| 4 | 3_4K46 | 22.2 ± 3.9 | 50.1 ± 3.5 | 25.0 ± 2.6 | 37.0 ± 1.9 | 14.2 ± 1.7 | 3.4 ± 1.4 | 0.0 ± 0.0 | 0.0 ± 0.0 | 0.0 ± 0.0 | 0.0 ± 0.0 |
|  | 4_5XJ7 | 56.4 ± 0.3 | 28.0 ± 1.1 | 30.9 ± 1.4 | 45.6 ± 2.9 | 22.8 ± 2.2 | 15.8 ± 3.5 | 0.0 ± 0.0 | 0.0 ± 0.0 | 0.0 ± 0.0 | 0.0 ± 0.0 |
|  | 5_5OJ8 | 0.0 ± 0.0 | 0.0 ± 0.0 | 5.8 ± 1.5 | 5.8 ± 2.6 | 26.3 ± 1.9 | 14.8 ± 2.3 | 0.0 ± 0.0 | 0.0 ± 0.0 | 0.0 ± 0.0 | 0.0 ± 0.0 |
|  | 6_2ZE5 | 49.0 ± 2.8 | 29.1 ± 2.4 | 34.4 ± 3.6 | 22.3 ± 4.3 | 14.4 ± 4.3 | 21.0 ± 3.0 | 0.0 ± 0.0 | 0.0 ± 0.0 | 0.0 ± 0.0 | 0.0 ± 0.0 |
|  | 7_1DEX | 34.1 ± 3.4 | 44.1 ± 1.6 | 18.5 ± 3.1 | 8.6 ± 1.0 | 4.6 ± 2.0 | 1.2 ± 0.7 | 0.0 ± 0.0 | 0.0 ± 0.0 | 0.0 ± 0.0 | 0.0 ± 0.0 |
| 5 | 8_1IS1 | 0.0 ± 0.0 | 1.0 ± 1.1 | 0.6 ± 1.2 | 0.0 ± 0.0 | 11.8 ± 1.5 | 3.0 ± 1.3 | 0.0 ± 0.0 | 0.0 ± 0.0 | 0.0 ± 0.0 | 0.0 ± 0.0 |
|  | 9_6KFQ | 37.2 ± 0.4 | 3.8 ± 0.6 | 10.7 ± 1.3 | 12.2 ± 1.2 | 18.1 ± 2.1 | 21.0 ± 3.9 | 0.0 ± 0.0 | 0.0 ± 0.0 | 0.0 ± 0.0 | 0.0 ± 0.0 |
|  | 10_5DN1 | 17.3 ± 1.8 | 7.2 ± 0.8 | 18.2 ± 1.3 | 7.8 ± 1.8 | 19.6 ± 3.7 | 13.8 ± 2.6 | 0.0 ± 0.0 | 0.0 ± 0.0 | 0.0 ± 0.0 | 0.0 ± 0.0 |
|  | 11_6KFQ | 25.4 ± 2.8 | 0.0 ± 0.0 | 3.1 ± 1.2 | 5.5 ± 1.7 | 11.4 ± 2.1 | 10.0 ± 2.4 | 0.0 ± 0.0 | 0.0 ± 0.0 | 0.0 ± 0.0 | 0.0 ± 0.0 |
|  | 12_1HU3 | 0.0 ± 0.0 | 0.0 ± 0.0 | 0.0 ± 0.0 | 0.0 ± 0.0 | 0.0 ± 0.0 | 0.0 ± 0.0 | 0.0 ± 0.0 | 0.0 ± 0.0 | 0.0 ± 0.0 | 0.0 ± 0.0 |
| 6 | 13_4BJI | 0.0 ± 0.0 | 15.0 ± 4.0 | 3.8 ± 0.4 | 3.2 ± 2.1 | 1.8 ± 1.3 | 0.0 ± 0.0 | 0.0 ± 0.0 | 0.0 ± 0.0 | 0.0 ± 0.0 | 0.0 ± 0.0 |
|  | 14_5KZL | 11.4 ± 1.0 | 11.3 ± 0.7 | 18.2 ± 1.6 | 17.5 ± 2.7 | 19.6 ± 1.3 | 8.5 ± 1.2 | 0.0 ± 0.0 | 0.0 ± 0.0 | 0.0 ± 0.0 | 0.0 ± 0.0 |
|  | 15_4LQ4 | 0.0 ± 0.0 | 16.4 ± 2.4 | 8.9 ± 2.7 | 6.8 ± 2.8 | 1.4 ± 1.0 | 0.0 ± 0.0 | 0.0 ± 0.0 | 0.0 ± 0.0 | 0.0 ± 0.0 | 0.0 ± 0.0 |
|  | 16_1SGW | 3.6 ± 1.5 | 37.5 ± 3.1 | 29.5 ± 3.3 | 24.0 ± 2.8 | 14.8 ± 3.8 | 11.6 ± 3.2 | 0.0 ± 0.0 | 0.0 ± 0.0 | 0.0 ± 0.0 | 0.0 ± 0.0 |
|  | 17_1BOL | 25.4 ± 2.7 | 30.2 ± 5.5 | 17.1 ± 4.2 | 5.8 ± 1.3 | 9.4 ± 1.0 | 4.0 ± 1.3 | 0.0 ± 0.0 | 13.1 ± 3.3 | 3.6 ± 1.2 | 0.0 ± 0.0 |
| 7 | 18_6W5B | 9.2 ± 1.5 | 6.7 ± 0.9 | 10.1 ± 1.0 | 11.9 ± 1.8 | 11.3 ± 1.3 | 9.4 ± 1.8 | 0.0 ± 0.0 | 0.0 ± 0.0 | 0.0 ± 0.0 | 0.0 ± 0.0 |
|  | 19_1Q0S | 18.8 ± 1.9 | 9.5 ± 1.0 | 13.7 ± 2.6 | 8.0 ± 1.6 | 7.3 ± 2.1 | 1.8 ± 1.2 | 0.0 ± 0.0 | 0.0 ± 0.0 | 0.0 ± 0.0 | 0.0 ± 0.0 |
|  | 20_4GVW | 23.3 ± 2.9 | 14.8 ± 1.4 | 15.5 ± 1.8 | 17.0 ± 1.1 | 17.2 ± 2.6 | 6.5 ± 1.5 | 0.0 ± 0.0 | 0.0 ± 0.0 | 0.0 ± 0.0 | 0.0 ± 0.0 |
|  | 21_3OSX | 7.3 ± 2.1 | 4.4 ± 1.3 | 3.6 ± 0.6 | 2.4 ± 0.8 | 2.8 ± 1.3 | 0.0 ± 0.0 | 0.0 ± 0.0 | 0.0 ± 0.0 | 0.0 ± 0.0 | 0.0 ± 0.0 |
|  | 22_1Z6N | 4.6 ± 1.0 | 0.0 ± 0.0 | 0.0 ± 0.0 | 0.0 ± 0.0 | 6.6 ± 2.9 | 4.8 ± 2.9 | 0.0 ± 0.0 | 0.0 ± 0.0 | 0.0 ± 0.0 | 0.0 ± 0.0 |

Table 10: **Task-level metrics for Protpardelle-1c across all experiments.**

| #Frags | Entry | Experiment | Success Rate | Novel Success (%) | Unique_Clusters | SUN_Score |
|---|---|---|---|---|---|---|
| 3 | 1_5CWP | paired | 94.4 ± 2.2 | 88.2 ± 1.9 | 27.6 ± 0.6 | 25.8 ± 0.6 |
| | 2_5CWN | paired | 85.4 ± 4.1 | 59.2 ± 4.2 | 51.5 ± 2.5 | 35.7 ± 2.5 |
| 4 | 3_4K46 | paired | 52.8 ± 3.8 | 52.8 ± 3.8 | 37.4 ± 2.7 | 37.4 ± 2.7 |
| | 4_5XJ7 | paired | 70.8 ± 2.9 | 70.8 ± 2.9 | 44.2 ± 1.8 | 44.2 ± 1.8 |
| | 5_5OJ8 | paired | 11.6 ± 2.1 | 4.0 ± 1.7 | 11.6 ± 2.1 | 4.0 ± 1.7 |
| | 6_2ZE5 | paired | 24.2 ± 4.2 | 24.2 ± 4.2 | 19.5 ± 3.4 | 19.5 ± 3.4 |
| | 7_1DEX | paired | 12.8 ± 4.9 | 12.8 ± 4.9 | 9.8 ± 3.8 | 9.8 ± 3.8 |
| 5 | 10_5DN1 | paired | 89.4 ± 3.6 | 20.2 ± 3.4 | 40.2 ± 1.6 | 9.1 ± 1.5 |
| | 11_6KFQ | paired | 83.2 ± 3.6 | 7.6 ± 2.6 | 71.3 ± 3.1 | 6.5 ± 2.2 |
| | 12_1HU3 | paired | 0.0 ± 0.0 | 0.0 ± 0.0 | 0.0 ± 0.0 | 0.0 ± 0.0 |
| | 8_1IS1 | paired | 47.0 ± 3.3 | 0.0 ± 0.0 | 0.0 ± 0.0 | 0.0 ± 0.0 |
| | 9_6KFQ | paired | 71.2 ± 1.9 | 21.2 ± 5.3 | 42.1 ± 1.1 | 12.5 ± 3.1 |
| 6 | 13_4BJI | paired | 3.6 ± 2.6 | 3.6 ± 2.6 | 3.6 ± 2.6 | 3.6 ± 2.6 |
| | 14_5KZL | paired | 81.8 ± 3.9 | 43.6 ± 5.5 | 35.1 ± 1.7 | 18.7 ± 2.4 |
| | 15_4LQ4 | paired | 5.8 ± 2.6 | 5.8 ± 2.6 | 5.8 ± 2.6 | 5.8 ± 2.6 |
| | 16_1SGW | paired | 36.6 ± 3.3 | 36.2 ± 3.5 | 30.1 ± 2.7 | 29.8 ± 2.9 |
| | 17_1BOL | paired | 5.0 ± 1.9 | 5.0 ± 1.9 | 5.0 ± 1.9 | 5.0 ± 1.9 |
| 7 | 18_6W5B | paired | 53.2 ± 2.6 | 41.6 ± 2.2 | 15.6 ± 0.7 | 12.2 ± 0.7 |
| | 19_1Q0S | paired | 14.0 ± 2.7 | 14.0 ± 2.7 | 8.6 ± 1.7 | 8.6 ± 1.7 |
| | 20_4GVW | paired | 58.8 ± 7.1 | 47.2 ± 5.9 | 22.5 ± 2.7 | 18.1 ± 2.3 |
| | 21_3OSX | paired | 11.6 ± 3.0 | 2.0 ± 1.8 | 11.6 ± 3.0 | 2.0 ± 1.8 |
| | 22_1Z6N | paired | 66.8 ± 3.9 | 0.0 ± 0.0 | 0.0 ± 0.0 | 0.0 ± 0.0 |
| 1 | 1_5OJ8 | single | 96.6 ± 1.4 | 96.6 ± 1.4 | 82.7 ± 1.2 | 82.7 ± 1.2 |
| | 2_1TKY | single | 47.0 ± 3.8 | 47.0 ± 3.8 | 47.0 ± 3.8 | 47.0 ± 3.8 |
| | 3_5XJ7 | single | 99.0 ± 0.9 | 99.0 ± 0.9 | 99.0 ± 0.9 | 99.0 ± 0.9 |
| | 4_6KFQ | single | 95.2 ± 1.7 | 95.2 ± 1.7 | 93.2 ± 1.7 | 93.2 ± 1.7 |
| | 5_5URP | single | 84.8 ± 3.8 | 84.8 ± 3.8 | 84.8 ± 3.8 | 84.8 ± 3.8 |
| 2 | 10_6FFV | single | 25.4 ± 4.3 | 25.4 ± 4.3 | 25.4 ± 4.3 | 25.4 ± 4.3 |
| | 6_5XJ7 | single | 86.0 ± 3.0 | 86.0 ± 3.0 | 85.0 ± 3.0 | 85.0 ± 3.0 |
| | 7_5OJ8 | single | 83.6 ± 3.1 | 83.6 ± 3.1 | 76.5 ± 2.9 | 76.5 ± 2.9 |
| | 8_1M2G | single | 15.0 ± 3.6 | 15.0 ± 3.6 | 15.0 ± 3.6 | 15.0 ± 3.6 |
| | 9_5CWP | single | 97.4 ± 1.6 | 97.4 ± 1.6 | 88.4 ± 1.5 | 88.4 ± 1.5 |
| 3 | 11_1Z6N | single | 94.2 ± 2.3 | 94.2 ± 2.3 | 74.4 ± 1.8 | 74.4 ± 1.8 |
| | 12_3P2W | single | 87.6 ± 4.5 | 87.6 ± 4.5 | 79.6 ± 4.1 | 79.6 ± 4.1 |
| | 13_6KFQ | single | 43.0 ± 4.0 | 43.0 ± 4.0 | 42.0 ± 3.9 | 42.0 ± 3.9 |
| | 14_4BJI | single | 79.6 ± 3.9 | 79.6 ± 3.9 | 77.7 ± 3.8 | 77.7 ± 3.8 |
| | 15_1A2J | single | 79.4 ± 3.7 | 79.4 ± 3.7 | 78.4 ± 3.7 | 78.4 ± 3.7 |
| 4 | 16_3PR9 | single | 52.0 ± 7.9 | 52.0 ± 7.9 | 52.0 ± 7.9 | 52.0 ± 7.9 |
| | 17_4GVW | single | 30.4 ± 7.0 | 30.4 ± 7.0 | 29.4 ± 6.8 | 29.4 ± 6.8 |
| | 18_4LQ4 | single | 63.8 ± 1.3 | 63.8 ± 1.3 | 61.8 ± 1.3 | 61.8 ± 1.3 |
| | 19_1M2G | single | 4.8 ± 2.2 | 4.8 ± 2.2 | 4.8 ± 2.2 | 4.8 ± 2.2 |
| | 20_3L86 | single | 76.6 ± 3.5 | 76.6 ± 3.5 | 66.7 ± 3.0 | 66.7 ± 3.0 |
| 5 | 21_1TKY | single | 5.2 ± 2.0 | 5.2 ± 2.0 | 5.2 ± 2.0 | 5.2 ± 2.0 |
| | 22_6TCS | single | 49.2 ± 4.8 | 49.2 ± 4.8 | 49.2 ± 4.8 | 49.2 ± 4.8 |
| | 23_1SGW | single | 64.0 ± 6.6 | 64.0 ± 6.6 | 61.0 ± 6.3 | 61.0 ± 6.3 |
| | 24_6OU0 | single | 37.8 ± 2.2 | 37.8 ± 2.2 | 33.5 ± 2.0 | 33.5 ± 2.0 |
| | 25_4F3H | single | 45.0 ± 3.0 | 45.0 ± 3.0 | 45.0 ± 3.0 | 45.0 ± 3.0 |
| 6 | 26_1GIU | single | 27.6 ± 3.8 | 27.6 ± 3.8 | 27.6 ± 3.8 | 27.6 ± 3.8 |
| | 27_4LQ4 | single | 66.6 ± 5.0 | 66.6 ± 5.0 | 66.6 ± 5.0 | 66.6 ± 5.0 |
| | 28_6TCS | single | 37.2 ± 3.3 | 37.2 ± 3.3 | 37.2 ± 3.3 | 37.2 ± 3.3 |
| | 29_2LAO | single | 31.4 ± 3.9 | 31.4 ± 3.9 | 31.4 ± 3.9 | 31.4 ± 3.9 |
| | 30_6TCS | single | 51.6 ± 6.3 | 51.6 ± 6.3 | 51.6 ± 6.3 | 51.6 ± 6.3 |
| 7 | 31_1GIU | single | 24.8 ± 2.9 | 24.8 ± 2.9 | 24.8 ± 2.9 | 24.8 ± 2.9 |
| | 32_1GBG | single | 66.4 ± 4.5 | 66.4 ± 4.5 | 60.5 ± 4.1 | 60.5 ± 4.1 |
| | 33_6TCS | single | 43.6 ± 2.9 | 43.6 ± 2.9 | 43.6 ± 2.9 | 43.6 ± 2.9 |
| | 34_6TCS | single | 44.8 ± 2.7 | 44.8 ± 2.7 | 44.8 ± 2.7 | 44.8 ± 2.7 |
| | 35_1A2J | single | 43.8 ± 4.0 | 43.8 ± 4.0 | 39.7 ± 3.6 | 39.7 ± 3.6 |

Table 11: **Task-level metrics for La-Proteina across all experiments.**

| #Frags | Entry | Experiment | Success Rate | Novel Success (%) | Unique_Clusters | SUN_Score |
|---|---|---|---|---|---|---|
| 3 | 1_5CWP | paired | 97.0 ± 1.4 | 82.4 ± 2.7 | 19.4 ± 0.3 | 16.5 ± 0.5 |
| | 2_5CWN | paired | 98.2 ± 1.2 | 65.8 ± 3.4 | 47.6 ± 0.6 | 31.9 ± 1.7 |
| 4 | 3_4K46 | paired | 59.0 ± 4.0 | 58.2 ± 3.6 | 50.1 ± 3.4 | 49.5 ± 3.1 |
| | 4_5XJ7 | paired | 63.6 ± 4.7 | 63.6 ± 4.7 | 27.4 ± 2.0 | 27.4 ± 2.0 |
| | 5_5OJ8 | paired | 68.6 ± 3.0 | 0.0 ± 0.0 | 0.0 ± 0.0 | 0.0 ± 0.0 |
| | 6_2ZE5 | paired | 47.2 ± 2.4 | 47.2 ± 2.4 | 28.9 ± 1.5 | 28.9 ± 1.5 |
| | 7_1DEX | paired | 67.8 ± 6.3 | 67.8 ± 6.3 | 44.9 ± 4.2 | 44.9 ± 4.2 |
| 5 | 10_5DN1 | paired | 93.4 ± 0.8 | 60.6 ± 2.4 | 11.1 ± 0.1 | 7.2 ± 0.3 |
| | 11_6KFQ | paired | 98.0 ± 1.7 | 0.0 ± 0.0 | 0.0 ± 0.0 | 0.0 ± 0.0 |
| | 12_1HU3 | paired | 0.8 ± 1.2 | 0.0 ± 0.0 | 0.0 ± 0.0 | 0.0 ± 0.0 |
| | 8_1IS1 | paired | 61.0 ± 3.3 | 1.2 ± 0.7 | 61.0 ± 3.3 | 1.2 ± 0.7 |
| | 9_6KFQ | paired | 99.0 ± 0.6 | 18.6 ± 4.5 | 20.8 ± 0.1 | 3.9 ± 1.0 |
| 6 | 13_4BJI | paired | 24.2 ± 3.5 | 24.2 ± 3.5 | 17.3 ± 2.5 | 17.3 ± 2.5 |
| | 14_5KZL | paired | 98.6 ± 1.0 | 64.0 ± 2.3 | 17.2 ± 0.2 | 11.2 ± 0.4 |
| | 15_4LQ4 | paired | 27.0 ± 3.7 | 27.0 ± 3.7 | 14.5 ± 2.0 | 14.5 ± 2.0 |
| | 16_1SGW | paired | 60.6 ± 3.9 | 60.6 ± 3.9 | 40.1 ± 2.6 | 40.1 ± 2.6 |
| | 17_1BOL | paired | 25.6 ± 2.2 | 25.6 ± 2.2 | 25.6 ± 2.2 | 25.6 ± 2.2 |
| 7 | 18_6W5B | paired | 80.0 ± 4.3 | 16.6 ± 5.6 | 28.2 ± 1.5 | 5.9 ± 2.0 |
| | 19_1Q0S | paired | 24.2 ± 2.4 | 24.2 ± 2.4 | 9.5 ± 0.9 | 9.5 ± 0.9 |
| | 20_4GVW | paired | 67.0 ± 3.3 | 53.0 ± 2.1 | 17.7 ± 0.9 | 14.0 ± 0.6 |
| | 21_3OSX | paired | 33.8 ± 1.8 | 9.8 ± 2.5 | 15.0 ± 0.8 | 4.4 ± 1.1 |
| | 22_1Z6N | paired | 91.8 ± 1.5 | 0.0 ± 0.0 | 0.0 ± 0.0 | 0.0 ± 0.0 |
| 1 | 1_5OJ8 | single | 100.0 ± 0.0 | 100.0 ± 0.0 | 96.0 ± 0.0 | 96.0 ± 0.0 |
| | 2_1TKY | single | 46.6 ± 3.4 | 46.6 ± 3.4 | 46.6 ± 3.4 | 46.6 ± 3.4 |
| | 3_5XJ7 | single | 99.2 ± 0.7 | 99.2 ± 0.7 | 91.2 ± 0.7 | 91.2 ± 0.7 |
| | 4_6KFQ | single | 83.2 ± 4.8 | 83.2 ± 4.8 | 80.2 ± 4.7 | 80.2 ± 4.7 |
| | 5_5URP | single | 96.4 ± 2.2 | 96.4 ± 2.2 | 96.4 ± 2.2 | 96.4 ± 2.2 |
| 2 | 10_6FFV | single | 24.0 ± 4.5 | 24.0 ± 4.5 | 15.4 ± 2.9 | 15.4 ± 2.9 |
| | 6_5XJ7 | single | 86.6 ± 3.3 | 86.6 ± 3.3 | 82.6 ± 3.1 | 82.6 ± 3.1 |
| | 7_5OJ8 | single | 97.4 ± 1.0 | 97.4 ± 1.0 | 71.6 ± 0.7 | 71.6 ± 0.7 |
| | 8_1M2G | single | 42.6 ± 2.9 | 42.6 ± 2.9 | 42.6 ± 2.9 | 42.6 ± 2.9 |
| | 9_5CWP | single | 85.6 ± 3.4 | 85.6 ± 3.4 | 85.6 ± 3.4 | 85.6 ± 3.4 |
| 3 | 11_1Z6N | single | 92.6 ± 1.9 | 92.6 ± 1.9 | 72.5 ± 1.5 | 72.5 ± 1.5 |
| | 12_3P2W | single | 93.4 ± 2.3 | 93.4 ± 2.3 | 88.4 ± 2.2 | 88.4 ± 2.2 |
| | 13_6KFQ | single | 39.2 ± 6.8 | 34.4 ± 6.7 | 17.4 ± 3.0 | 15.3 ± 3.0 |
| | 14_4BJI | single | 90.8 ± 2.2 | 90.8 ± 2.2 | 90.8 ± 2.2 | 90.8 ± 2.2 |
| | 15_1A2J | single | 96.4 ± 1.9 | 96.4 ± 1.9 | 96.4 ± 1.9 | 96.4 ± 1.9 |
| 4 | 16_3PR9 | single | 18.0 ± 5.1 | 18.0 ± 5.1 | 18.0 ± 5.1 | 18.0 ± 5.1 |
| | 17_4GVW | single | 54.2 ± 1.7 | 54.2 ± 1.7 | 48.1 ± 1.5 | 48.1 ± 1.5 |
| | 18_4LQ4 | single | 92.4 ± 2.4 | 92.4 ± 2.4 | 59.6 ± 1.6 | 59.6 ± 1.6 |
| | 19_1M2G | single | 29.2 ± 6.6 | 29.2 ± 6.6 | 29.2 ± 6.6 | 29.2 ± 6.6 |
| | 20_3L86 | single | 84.6 ± 3.4 | 84.6 ± 3.4 | 72.7 ± 3.0 | 72.7 ± 3.0 |
| 5 | 21_1TKY | single | 57.4 ± 5.2 | 57.4 ± 5.2 | 52.4 ± 4.7 | 52.4 ± 4.7 |
| | 22_6TCS | single | 53.6 ± 3.0 | 53.6 ± 3.0 | 53.6 ± 3.0 | 53.6 ± 3.0 |
| | 23_1SGW | single | 96.8 ± 1.6 | 96.8 ± 1.6 | 75.8 ± 1.3 | 75.8 ± 1.3 |
| | 24_6OU0 | single | 77.8 ± 3.2 | 77.8 ± 3.2 | 72.8 ± 3.0 | 72.8 ± 3.0 |
| | 25_4F3H | single | 40.8 ± 2.1 | 40.8 ± 2.1 | 40.8 ± 2.1 | 40.8 ± 2.1 |
| 6 | 26_1GIU | single | 53.6 ± 3.1 | 53.6 ± 3.1 | 53.6 ± 3.1 | 53.6 ± 3.1 |
| | 27_4LQ4 | single | 86.8 ± 4.4 | 86.8 ± 4.4 | 76.9 ± 3.9 | 76.9 ± 3.9 |
| | 28_6TCS | single | 63.6 ± 3.4 | 63.6 ± 3.4 | 63.6 ± 3.4 | 63.6 ± 3.4 |
| | 29_2LAO | single | 37.2 ± 1.7 | 37.2 ± 1.7 | 37.2 ± 1.7 | 37.2 ± 1.7 |
| | 30_6TCS | single | 17.6 ± 4.3 | 17.6 ± 4.3 | 17.6 ± 4.3 | 17.6 ± 4.3 |
| 7 | 31_1GIU | single | 60.4 ± 4.4 | 60.4 ± 4.4 | 60.4 ± 4.4 | 60.4 ± 4.4 |
| | 32_1GBG | single | 88.6 ± 1.4 | 88.6 ± 1.4 | 84.6 ± 1.3 | 84.6 ± 1.3 |
| | 33_6TCS | single | 73.0 ± 3.3 | 73.0 ± 3.3 | 67.9 ± 3.1 | 67.9 ± 3.1 |
| | 34_6TCS | single | 17.2 ± 3.5 | 17.2 ± 3.5 | 17.2 ± 3.5 | 17.2 ± 3.5 |
| | 35_1A2J | single | 50.4 ± 1.5 | 50.4 ± 1.5 | 50.4 ± 1.5 | 50.4 ± 1.5 |

Table 12: **Task-level metrics for RFdiffusion2 across all experiments.**

| #Frags | Entry | Experiment | Success Rate | Novel Success (%) | Unique_Clusters | SUN_Score |
|---|---|---|---|---|---|---|
| 3 | 1_5CWP | paired | 18.4 ± 2.2 | 18.4 ± 2.2 | 18.4 ± 2.2 | 18.4 ± 2.2 |
| | 2_5CWN | paired | 64.2 ± 5.2 | 64.2 ± 5.2 | 62.2 ± 5.0 | 62.2 ± 5.0 |
| 4 | 3_4K46 | paired | 4.4 ± 1.0 | 4.4 ± 1.0 | 4.4 ± 1.0 | 4.4 ± 1.0 |
| | 4_5XJ7 | paired | 15.0 ± 1.8 | 15.0 ± 1.8 | 15.0 ± 1.8 | 15.0 ± 1.8 |
| | 5_5OJ8 | paired | 22.4 ± 5.3 | 22.4 ± 5.3 | 22.4 ± 5.3 | 22.4 ± 5.3 |
| | 6_2ZE5 | paired | 23.6 ± 3.3 | 23.6 ± 3.3 | 23.6 ± 3.3 | 23.6 ± 3.3 |
| | 7_1DEX | paired | 1.4 ± 1.5 | 1.4 ± 1.5 | 1.4 ± 1.5 | 1.4 ± 1.5 |
| 5 | 10_5DN1 | paired | 17.8 ± 3.0 | 15.8 ± 1.9 | 14.5 ± 2.4 | 12.8 ± 1.6 |
| | 11_6KFQ | paired | 21.8 ± 4.2 | 14.0 ± 3.4 | 19.8 ± 3.8 | 12.7 ± 3.1 |
| | 12_1HU3 | paired | 0.0 ± 0.0 | 0.0 ± 0.0 | 0.0 ± 0.0 | 0.0 ± 0.0 |
| | 8_1IS1 | paired | 3.6 ± 1.7 | 3.6 ± 1.7 | 3.6 ± 1.7 | 3.6 ± 1.7 |
| | 9_6KFQ | paired | 22.6 ± 5.5 | 20.8 ± 5.8 | 20.6 ± 5.1 | 19.0 ± 5.3 |
| 6 | 13_4BJI | paired | 0.0 ± 0.0 | 0.0 ± 0.0 | 0.0 ± 0.0 | 0.0 ± 0.0 |
| | 14_5KZL | paired | 9.4 ± 2.1 | 9.2 ± 2.0 | 8.4 ± 1.8 | 8.2 ± 1.8 |
| | 15_4LQ4 | paired | 0.0 ± 0.0 | 0.0 ± 0.0 | 0.0 ± 0.0 | 0.0 ± 0.0 |
| | 16_1SGW | paired | 10.0 ± 1.7 | 10.0 ± 1.7 | 10.0 ± 1.7 | 10.0 ± 1.7 |
| | 17_1BOL | paired | 1.8 ± 1.3 | 1.8 ± 1.3 | 1.8 ± 1.3 | 1.8 ± 1.3 |
| 7 | 18_6W5B | paired | 16.8 ± 4.0 | 16.0 ± 3.2 | 10.3 ± 2.5 | 9.8 ± 2.0 |
| | 19_1Q0S | paired | 1.6 ± 1.0 | 1.6 ± 1.0 | 1.6 ± 1.0 | 1.6 ± 1.0 |
| | 20_4GVW | paired | 7.8 ± 3.9 | 7.8 ± 3.9 | 6.8 ± 3.4 | 6.8 ± 3.4 |
| | 21_3OSX | paired | 1.0 ± 1.3 | 0.0 ± 0.0 | 0.0 ± 0.0 | 0.0 ± 0.0 |
| | 22_1Z6N | paired | 30.8 ± 2.6 | 5.6 ± 1.9 | 30.8 ± 2.6 | 5.6 ± 1.9 |
| 1 | 1_5OJ8 | single | 74.0 ± 3.0 | 74.0 ± 3.0 | 74.0 ± 3.0 | 74.0 ± 3.0 |
| | 2_1TKY | single | 1.6 ± 1.4 | 1.6 ± 1.4 | 1.6 ± 1.4 | 1.6 ± 1.4 |
| | 3_5XJ7 | single | 83.0 ± 3.6 | 83.0 ± 3.6 | 83.0 ± 3.6 | 83.0 ± 3.6 |
| | 4_6KFQ | single | 52.0 ± 5.3 | 52.0 ± 5.3 | 52.0 ± 5.3 | 52.0 ± 5.3 |
| | 5_5URP | single | 40.0 ± 4.7 | 40.0 ± 4.7 | 40.0 ± 4.7 | 40.0 ± 4.7 |
| 2 | 10_6FFV | single | 1.2 ± 1.9 | 1.2 ± 1.9 | 1.2 ± 1.9 | 1.2 ± 1.9 |
| | 6_5XJ7 | single | 17.4 ± 3.8 | 17.4 ± 3.8 | 17.4 ± 3.8 | 17.4 ± 3.8 |
| | 7_5OJ8 | single | 68.0 ± 2.1 | 68.0 ± 2.1 | 68.0 ± 2.1 | 68.0 ± 2.1 |
| | 8_1M2G | single | 11.0 ± 1.7 | 11.0 ± 1.7 | 11.0 ± 1.7 | 11.0 ± 1.7 |
| | 9_5CWP | single | 59.0 ± 2.8 | 59.0 ± 2.8 | 59.0 ± 2.8 | 59.0 ± 2.8 |
| 3 | 11_1Z6N | single | 34.0 ± 2.5 | 34.0 ± 2.5 | 34.0 ± 2.5 | 34.0 ± 2.5 |
| | 12_3P2W | single | 26.6 ± 3.6 | 26.6 ± 3.6 | 26.6 ± 3.6 | 26.6 ± 3.6 |
| | 13_6KFQ | single | 13.0 ± 2.8 | 13.0 ± 2.8 | 13.0 ± 2.8 | 13.0 ± 2.8 |
| | 14_4BJI | single | 17.4 ± 4.9 | 17.4 ± 4.9 | 17.4 ± 4.9 | 17.4 ± 4.9 |
| | 15_1A2J | single | 27.2 ± 4.4 | 27.2 ± 4.4 | 27.2 ± 4.4 | 27.2 ± 4.4 |
| 4 | 16_3PR9 | single | 38.2 ± 5.6 | 38.2 ± 5.6 | 38.2 ± 5.6 | 38.2 ± 5.6 |
| | 17_4GVW | single | 5.4 ± 1.9 | 5.4 ± 1.9 | 5.4 ± 1.9 | 5.4 ± 1.9 |
| | 18_4LQ4 | single | 29.2 ± 2.8 | 29.2 ± 2.8 | 29.2 ± 2.8 | 29.2 ± 2.8 |
| | 19_1M2G | single | 2.6 ± 1.7 | 2.6 ± 1.7 | 2.6 ± 1.7 | 2.6 ± 1.7 |
| | 20_3L86 | single | 28.8 ± 2.8 | 28.8 ± 2.8 | 28.8 ± 2.8 | 28.8 ± 2.8 |
| 5 | 21_1TKY | single | 0.0 ± 0.0 | 0.0 ± 0.0 | 0.0 ± 0.0 | 0.0 ± 0.0 |
| | 22_6TCS | single | 26.8 ± 3.1 | 26.8 ± 3.1 | 26.8 ± 3.1 | 26.8 ± 3.1 |
| | 23_1SGW | single | 13.4 ± 3.8 | 13.4 ± 3.8 | 13.4 ± 3.8 | 13.4 ± 3.8 |
| | 24_6OU0 | single | 25.6 ± 2.7 | 25.6 ± 2.7 | 25.6 ± 2.7 | 25.6 ± 2.7 |
| | 25_4F3H | single | 19.4 ± 3.4 | 19.4 ± 3.4 | 19.4 ± 3.4 | 19.4 ± 3.4 |
| 6 | 26_1GIU | single | 10.0 ± 2.6 | 10.0 ± 2.6 | 10.0 ± 2.6 | 10.0 ± 2.6 |
| | 27_4LQ4 | single | 19.2 ± 4.7 | 19.2 ± 4.7 | 19.2 ± 4.7 | 19.2 ± 4.7 |
| | 28_6TCS | single | 15.2 ± 4.0 | 15.2 ± 4.0 | 15.2 ± 4.0 | 15.2 ± 4.0 |
| | 29_2LAO | single | 1.6 ± 0.8 | 1.6 ± 0.8 | 1.6 ± 0.8 | 1.6 ± 0.8 |
| | 30_6TCS | single | 9.8 ± 2.8 | 9.8 ± 2.8 | 9.8 ± 2.8 | 9.8 ± 2.8 |
| 7 | 31_1GIU | single | 1.6 ± 1.0 | 1.6 ± 1.0 | 1.6 ± 1.0 | 1.6 ± 1.0 |
| | 32_1GBG | single | 28.8 ± 4.6 | 28.8 ± 4.6 | 28.8 ± 4.6 | 28.8 ± 4.6 |
| | 33_6TCS | single | 32.2 ± 6.0 | 32.2 ± 6.0 | 32.2 ± 6.0 | 32.2 ± 6.0 |
| | 34_6TCS | single | 25.6 ± 3.4 | 25.6 ± 3.4 | 25.6 ± 3.4 | 25.6 ± 3.4 |
| | 35_1A2J | single | 10.2 ± 2.4 | 10.2 ± 2.4 | 10.2 ± 2.4 | 10.2 ± 2.4 |

Table 13: **Task-level metrics for RFdiffusion across all experiments.**

| #Frags | Entry | Experiment | Successful, % ↑ | Novel, % ↑ | Unique, % ↑ | SUN Score ↑ |
|---|---|---|---|---|---|---|
| 3 | 1_5CWP | paired | 84.8 ± 2.79 | 27.0 ± 6.03 | 48.9 ± 1.61 | 15.6 ± 3.5 |
| | 2_5CWN | paired | 83.0 ± 3.58 | 40.6 ± 0.80 | 38.3 ± 1.65 | 18.7 ± 0.4 |
| 4 | 3_4K46 | paired | 25.8 ± 2.14 | 25.8 ± 2.14 | 22.6 ± 1.87 | 22.6 ± 1.9 |
| | 4_5XJ7 | paired | 79.0 ± 2.45 | 79.0 ± 2.45 | 31.4 ± 0.97 | 31.4 ± 1.0 |
| | 5_5OJ8 | paired | 20.8 ± 3.12 | 6.8 ± 2.32 | 13.0 ± 1.95 | 4.2 ± 1.4 |
| | 6_2ZE5 | paired | 41.8 ± 2.48 | 41.8 ± 2.48 | 38.5 ± 2.29 | 38.5 ± 2.3 |
| | 7_1DEX | paired | 21.2 ± 4.26 | 21.2 ± 4.26 | 15.7 ± 3.15 | 15.7 ± 3.1 |
| 5 | 10_5DN1 | paired | 57.8 ± 4.26 | 44.6 ± 4.54 | 22.0 ± 1.62 | 17.0 ± 1.7 |
| | 11_6KFQ | paired | 84.4 ± 3.44 | 8.8 ± 0.98 | 42.2 ± 1.72 | 4.4 ± 0.5 |
| | 12_1HU3 | paired | 0.0 ± 0.00 | 0.0 ± 0.00 | 0.0 ± 0.00 | 0.0 ± 0.0 |
| | 8_1IS1 | paired | 13.6 ± 3.72 | 0.6 ± 0.49 | 13.6 ± 3.72 | 0.6 ± 0.5 |
| | 9_6KFQ | paired | 94.0 ± 2.00 | 28.8 ± 2.79 | 32.5 ± 0.69 | 10.0 ± 1.0 |
| 6 | 13_4BJI | paired | 2.6 ± 1.20 | 2.6 ± 1.20 | 2.6 ± 1.20 | 2.6 ± 1.2 |
| | 14_5KZL | paired | 54.2 ± 5.27 | 45.4 ± 5.61 | 19.5 ± 1.90 | 16.3 ± 2.0 |
| | 15_4LQ4 | paired | 8.2 ± 3.49 | 8.2 ± 3.49 | 7.3 ± 3.10 | 7.3 ± 3.1 |
| | 16_1SGW | paired | 38.2 ± 4.31 | 38.2 ± 4.31 | 29.4 ± 3.31 | 29.4 ± 3.3 |
| | 17_1BOL | paired | 14.0 ± 3.69 | 14.0 ± 3.69 | 13.2 ± 3.47 | 13.2 ± 3.5 |
| 7 | 18_6W5B | paired | 30.4 ± 5.54 | 29.2 ± 5.91 | 9.8 ± 1.79 | 9.4 ± 1.9 |
| | 19_1Q0S | paired | 39.0 ± 7.40 | 39.0 ± 7.40 | 14.5 ± 2.75 | 14.5 ± 2.7 |
| | 20_4GVW | paired | 42.6 ± 3.83 | 42.6 ± 3.83 | 14.2 ± 1.28 | 14.2 ± 1.3 |
| | 21_3OSX | paired | 22.4 ± 5.16 | 10.0 ± 3.90 | 10.2 ± 2.35 | 4.5 ± 1.8 |
| | 22_1Z6N | paired | 87.8 ± 2.48 | 0.0 ± 0.00 | 0.0 ± 0.00 | 0.0 ± 0.0 |
| 1 | 1_5OJ8 | single | 100.0 ± 0.00 | 100.0 ± 0.00 | 98.0 ± 0.00 | 98.0 ± 0.0 |
| | 2_1TKY | single | 79.4 ± 3.38 | 79.4 ± 3.38 | 79.4 ± 3.38 | 79.4 ± 3.4 |
| | 3_5XJ7 | single | 100.0 ± 0.00 | 100.0 ± 0.00 | 99.0 ± 0.00 | 99.0 ± 0.0 |
| | 4_6KFQ | single | 97.8 ± 0.75 | 97.8 ± 0.75 | 97.8 ± 0.75 | 97.8 ± 0.7 |
| | 5_5URP | single | 82.2 ± 3.54 | 82.2 ± 3.54 | 82.2 ± 3.54 | 82.2 ± 3.5 |
| 2 | 10_6FFV | single | 34.8 ± 4.17 | 34.8 ± 4.17 | 34.8 ± 4.17 | 34.8 ± 4.2 |
| | 6_5XJ7 | single | 93.0 ± 2.10 | 93.0 ± 2.10 | 93.0 ± 2.10 | 93.0 ± 2.1 |
| | 7_5OJ8 | single | 98.0 ± 0.89 | 98.0 ± 0.89 | 93.1 ± 0.85 | 93.1 ± 0.8 |
| | 8_1M2G | single | 41.6 ± 5.50 | 41.6 ± 5.50 | 41.6 ± 5.50 | 41.6 ± 5.5 |
| | 9_5CWP | single | 99.0 ± 0.63 | 99.0 ± 0.63 | 97.0 ± 0.62 | 97.0 ± 0.6 |
| 3 | 11_1Z6N | single | 77.8 ± 4.92 | 77.8 ± 4.92 | 63.5 ± 4.01 | 63.5 ± 4.0 |
| | 12_3P2W | single | 87.0 ± 1.67 | 87.0 ± 1.67 | 83.0 ± 1.60 | 83.0 ± 1.6 |
| | 13_6KFQ | single | 93.8 ± 2.04 | 93.8 ± 2.04 | 86.7 ± 1.89 | 86.7 ± 1.9 |
| | 14_4BJI | single | 96.0 ± 2.19 | 96.0 ± 2.19 | 95.0 ± 2.17 | 95.0 ± 2.2 |
| | 15_1A2J | single | 96.4 ± 1.85 | 96.4 ± 1.85 | 96.4 ± 1.85 | 96.4 ± 1.9 |
| 4 | 16_3PR9 | single | 70.8 ± 3.97 | 70.8 ± 3.97 | 70.8 ± 3.97 | 70.8 ± 4.0 |
| | 17_4GVW | single | 62.8 ± 2.93 | 62.8 ± 2.93 | 42.8 ± 1.99 | 42.8 ± 2.0 |
| | 18_4LQ4 | single | 91.4 ± 0.80 | 91.4 ± 0.80 | 79.5 ± 0.70 | 79.5 ± 0.7 |
| | 19_1M2G | single | 24.8 ± 3.06 | 24.8 ± 3.06 | 24.8 ± 3.06 | 24.8 ± 3.1 |
| | 20_3L86 | single | 64.0 ± 2.37 | 64.0 ± 2.37 | 62.1 ± 2.30 | 62.1 ± 2.3 |
| 5 | 21_1TKY | single | 1.6 ± 0.80 | 1.6 ± 0.80 | 1.6 ± 0.80 | 1.6 ± 0.8 |
| | 22_6TCS | single | 47.8 ± 4.87 | 47.8 ± 4.87 | 47.8 ± 4.87 | 47.8 ± 4.9 |
| | 23_1SGW | single | 52.2 ± 1.94 | 52.2 ± 1.94 | 52.2 ± 1.94 | 52.2 ± 1.9 |
| | 24_6OU0 | single | 41.4 ± 4.27 | 41.4 ± 4.27 | 41.4 ± 4.27 | 41.4 ± 4.3 |
| | 25_4F3H | single | 71.0 ± 1.10 | 71.0 ± 1.10 | 65.7 ± 1.01 | 65.7 ± 1.0 |
| 6 | 26_1GIU | single | 82.4 ± 2.33 | 82.4 ± 2.33 | 82.4 ± 2.33 | 82.4 ± 2.3 |
| | 27_4LQ4 | single | 78.8 ± 3.71 | 78.8 ± 3.71 | 77.8 ± 3.66 | 77.8 ± 3.7 |
| | 28_6TCS | single | 50.4 ± 3.88 | 50.4 ± 3.88 | 50.4 ± 3.88 | 50.4 ± 3.9 |
| | 29_2LAO | single | 7.4 ± 3.01 | 7.4 ± 3.01 | 7.4 ± 3.01 | 7.4 ± 3.0 |
| | 30_6TCS | single | 54.4 ± 4.22 | 54.4 ± 4.22 | 54.4 ± 4.22 | 54.4 ± 4.2 |
| 7 | 31_1GIU | single | 38.2 ± 8.28 | 38.2 ± 8.28 | 38.2 ± 8.28 | 38.2 ± 8.3 |
| | 32_1GBG | single | 37.8 ± 4.35 | 37.8 ± 4.35 | 32.5 ± 3.75 | 32.5 ± 3.7 |
| | 33_6TCS | single | 54.2 ± 4.66 | 54.2 ± 4.66 | 53.2 ± 4.58 | 53.2 ± 4.6 |
| | 34_6TCS | single | 14.4 ± 3.38 | 14.4 ± 3.38 | 14.4 ± 3.38 | 14.4 ± 3.4 |
| | 35_1A2J | single | 60.4 ± 2.65 | 60.4 ± 2.65 | 50.7 ± 2.23 | 50.7 ± 2.2 |

Table 14: **Entry-level metrics for Genie2 across all experiments.**

| #Frags | Entry | Experiment | Success Rate | Novel Success (%) | Unique_Clusters | SUN_Score |
|---|---|---|---|---|---|---|
| 3 | 1_5CWP | paired | 4.2 ± 2.1 | 4.2 ± 2.1 | 4.2 ± 2.1 | 4.2 ± 2.1 |
| | 2_5CWN | paired | 91.6 ± 1.6 | 91.6 ± 1.6 | 74.3 ± 1.3 | 74.3 ± 1.3 |
| 4 | 3_4K46 | paired | 23.8 ± 2.1 | 23.8 ± 2.1 | 23.8 ± 2.1 | 23.8 ± 2.1 |
| | 4_5XJ7 | paired | 90.8 ± 2.5 | 90.8 ± 2.5 | 56.5 ± 1.5 | 56.5 ± 1.5 |
| | 5_5OJ8 | paired | 0.0 ± 0.0 | 0.0 ± 0.0 | 0.0 ± 0.0 | 0.0 ± 0.0 |
| | 6_2ZE5 | paired | 53.0 ± 5.4 | 53.0 ± 5.4 | 49.1 ± 5.0 | 49.1 ± 5.0 |
| | 7_1DEX | paired | 30.0 ± 5.2 | 30.0 ± 5.2 | 28.3 ± 4.9 | 28.3 ± 4.9 |
| 5 | 10_5DN1 | paired | 19.0 ± 2.8 | 19.0 ± 2.8 | 16.1 ± 2.3 | 16.1 ± 2.3 |
| | 11_6KFQ | paired | 90.2 ± 1.7 | 42.6 ± 2.2 | 55.5 ± 1.1 | 26.2 ± 1.3 |
| | 12_1HU3 | paired | 0.0 ± 0.0 | 0.0 ± 0.0 | 0.0 ± 0.0 | 0.0 ± 0.0 |
| | 8_1IS1 | paired | 0.0 ± 0.0 | 0.0 ± 0.0 | 0.0 ± 0.0 | 0.0 ± 0.0 |
| | 9_6KFQ | paired | 97.0 ± 0.9 | 97.0 ± 0.9 | 37.4 ± 0.3 | 37.4 ± 0.3 |
| 6 | 13_4BJI | paired | 0.0 ± 0.0 | 0.0 ± 0.0 | 0.0 ± 0.0 | 0.0 ± 0.0 |
| | 14_5KZL | paired | 14.8 ± 5.5 | 14.8 ± 5.5 | 10.9 ± 4.0 | 10.9 ± 4.0 |
| | 15_4LQ4 | paired | 0.0 ± 0.0 | 0.0 ± 0.0 | 0.0 ± 0.0 | 0.0 ± 0.0 |
| | 16_1SGW | paired | 2.4 ± 1.0 | 2.4 ± 1.0 | 2.4 ± 1.0 | 2.4 ± 1.0 |
| | 17_1BOL | paired | 22.2 ± 4.0 | 22.2 ± 4.0 | 22.2 ± 4.0 | 22.2 ± 4.0 |
| 7 | 18_6W5B | paired | 12.2 ± 3.9 | 12.2 ± 3.9 | 8.1 ± 2.6 | 8.1 ± 2.6 |
| | 19_1Q0S | paired | 21.8 ± 6.1 | 21.8 ± 6.1 | 18.2 ± 5.0 | 18.2 ± 5.0 |
| | 20_4GVW | paired | 40.0 ± 6.4 | 39.6 ± 6.6 | 22.4 ± 3.6 | 22.2 ± 3.7 |
| | 21_3OSX | paired | 13.2 ± 1.7 | 13.2 ± 1.7 | 7.2 ± 0.9 | 7.2 ± 0.9 |
| | 22_1Z6N | paired | 88.6 ± 3.3 | 7.2 ± 1.6 | 49.2 ± 1.8 | 4.0 ± 0.9 |
| 1 | 1_5OJ8 | single | 100.0 ± 0.0 | 100.0 ± 0.0 | 100.0 ± 0.0 | 100.0 ± 0.0 |
| | 2_1TKY | single | 4.8 ± 1.9 | 4.8 ± 1.9 | 4.8 ± 1.9 | 4.8 ± 1.9 |
| | 3_5XJ7 | single | 100.0 ± 0.0 | 100.0 ± 0.0 | 100.0 ± 0.0 | 100.0 ± 0.0 |
| | 4_6KFQ | single | 92.8 ± 2.0 | 92.8 ± 2.0 | 92.8 ± 2.0 | 92.8 ± 2.0 |
| | 5_5URP | single | 81.6 ± 3.5 | 81.6 ± 3.5 | 81.6 ± 3.5 | 81.6 ± 3.5 |
| 2 | 10_6FFV | single | 41.8 ± 2.9 | 41.8 ± 2.9 | 41.8 ± 2.9 | 41.8 ± 2.9 |
| | 6_5XJ7 | single | 87.6 ± 1.6 | 87.6 ± 1.6 | 87.6 ± 1.6 | 87.6 ± 1.6 |
| | 7_5OJ8 | single | 99.0 ± 0.6 | 99.0 ± 0.6 | 99.0 ± 0.6 | 99.0 ± 0.6 |
| | 8_1M2G | single | 54.8 ± 2.5 | 54.8 ± 2.5 | 54.8 ± 2.5 | 54.8 ± 2.5 |
| | 9_5CWP | single | 97.0 ± 1.1 | 97.0 ± 1.1 | 97.0 ± 1.1 | 97.0 ± 1.1 |
| 3 | 11_1Z6N | single | 89.4 ± 0.5 | 89.4 ± 0.5 | 89.4 ± 0.5 | 89.4 ± 0.5 |
| | 12_3P2W | single | 73.6 ± 0.5 | 73.6 ± 0.5 | 73.6 ± 0.5 | 73.6 ± 0.5 |
| | 13_6KFQ | single | 83.8 ± 2.0 | 83.8 ± 2.0 | 83.8 ± 2.0 | 83.8 ± 2.0 |
| | 14_4BJI | single | 91.2 ± 3.5 | 91.2 ± 3.5 | 90.2 ± 3.5 | 90.2 ± 3.5 |
| | 15_1A2J | single | 94.0 ± 2.4 | 94.0 ± 2.4 | 94.0 ± 2.4 | 94.0 ± 2.4 |
| 4 | 16_3PR9 | single | 12.6 ± 3.3 | 12.6 ± 3.3 | 12.6 ± 3.3 | 12.6 ± 3.3 |
| | 17_4GVW | single | 35.2 ± 5.4 | 35.2 ± 5.4 | 35.2 ± 5.4 | 35.2 ± 5.4 |
| | 18_4LQ4 | single | 63.4 ± 5.3 | 63.4 ± 5.3 | 63.4 ± 5.3 | 63.4 ± 5.3 |
| | 19_1M2G | single | 7.2 ± 2.4 | 7.2 ± 2.4 | 7.2 ± 2.4 | 7.2 ± 2.4 |
| | 20_3L86 | single | 59.8 ± 2.6 | 59.8 ± 2.6 | 59.8 ± 2.6 | 59.8 ± 2.6 |
| 5 | 21_1TKY | single | 1.0 ± 0.9 | 1.0 ± 0.9 | 1.0 ± 0.9 | 1.0 ± 0.9 |
| | 22_6TCS | single | 45.6 ± 3.4 | 45.6 ± 3.4 | 45.6 ± 3.4 | 45.6 ± 3.4 |
| | 23_1SGW | single | 21.4 ± 3.1 | 21.4 ± 3.1 | 21.4 ± 3.1 | 21.4 ± 3.1 |
| | 24_6OU0 | single | 37.8 ± 4.4 | 37.8 ± 4.4 | 37.8 ± 4.4 | 37.8 ± 4.4 |
| | 25_4F3H | single | 56.8 ± 3.4 | 56.8 ± 3.4 | 56.8 ± 3.4 | 56.8 ± 3.4 |
| 6 | 26_1GIU | single | 40.6 ± 5.0 | 40.6 ± 5.0 | 40.6 ± 5.0 | 40.6 ± 5.0 |
| | 27_4LQ4 | single | 86.2 ± 3.1 | 86.2 ± 3.1 | 85.2 ± 3.0 | 85.2 ± 3.0 |
| | 28_6TCS | single | 57.4 ± 2.1 | 57.4 ± 2.1 | 57.4 ± 2.1 | 57.4 ± 2.1 |
| | 29_2LAO | single | 7.4 ± 1.5 | 7.4 ± 1.5 | 7.4 ± 1.5 | 7.4 ± 1.5 |
| | 30_6TCS | single | 56.2 ± 2.7 | 56.2 ± 2.7 | 56.2 ± 2.7 | 56.2 ± 2.7 |
| 7 | 31_1GIU | single | 24.8 ± 5.1 | 24.8 ± 5.1 | 24.8 ± 5.1 | 24.8 ± 5.1 |
| | 32_1GBG | single | 66.6 ± 3.8 | 66.6 ± 3.8 | 66.6 ± 3.8 | 66.6 ± 3.8 |
| | 33_6TCS | single | 66.0 ± 2.8 | 66.0 ± 2.8 | 66.0 ± 2.8 | 66.0 ± 2.8 |
| | 34_6TCS | single | 74.4 ± 4.6 | 74.4 ± 4.6 | 74.4 ± 4.6 | 74.4 ± 4.6 |
| | 35_1A2J | single | 56.6 ± 3.4 | 56.6 ± 3.4 | 55.6 ± 3.3 | 55.6 ± 3.3 |

Table 15: **Entry-level metrics for FrameFlow across all experiments.**

| #Frags | Entry | Experiment | Success Rate | Novel Success (%) | Unique_Clusters | SUN_Score |
|---|---|---|---|---|---|---|
| 3 | 1_5CWP | paired | 55.4 ± 1.9 | 55.4 ± 1.9 | 51.4 ± 1.7 | 51.4 ± 1.7 |
| | 2_5CWN | paired | 82.6 ± 3.0 | 80.6 ± 3.2 | 77.2 ± 2.8 | 75.4 ± 3.0 |
| 4 | 3_4K46 | paired | 13.8 ± 3.9 | 13.8 ± 3.9 | 13.8 ± 3.9 | 13.8 ± 3.9 |
| | 4_5XJ7 | paired | 24.2 ± 3.4 | 24.2 ± 3.4 | 24.2 ± 3.4 | 24.2 ± 3.4 |
| | 5_5OJ8 | paired | 49.0 ± 4.3 | 35.0 ± 5.8 | 36.4 ± 3.2 | 26.0 ± 4.3 |
| | 6_2ZE5 | paired | 14.8 ± 3.1 | 14.8 ± 3.1 | 14.8 ± 3.1 | 14.8 ± 3.1 |
| | 7_1DEX | paired | 5.2 ± 2.7 | 5.2 ± 2.7 | 5.2 ± 2.7 | 5.2 ± 2.7 |
| 5 | 10_5DN1 | paired | 38.6 ± 3.1 | 24.0 ± 2.2 | 27.2 ± 2.2 | 16.9 ± 1.5 |
| | 11_6KFQ | paired | 40.6 ± 4.8 | 20.2 ± 2.7 | 25.6 ± 3.1 | 12.8 ± 1.7 |
| | 12_1HU3 | paired | 0.0 ± 0.0 | 0.0 ± 0.0 | 0.0 ± 0.0 | 0.0 ± 0.0 |
| | 8_1IS1 | paired | 13.2 ± 1.5 | 11.2 ± 1.2 | 13.2 ± 1.5 | 11.2 ± 1.2 |
| | 9_6KFQ | paired | 55.8 ± 2.8 | 34.6 ± 5.3 | 33.1 ± 1.7 | 20.5 ± 3.2 |
| 6 | 13_4BJI | paired | 0.6 ± 0.8 | 0.6 ± 0.8 | 0.6 ± 0.8 | 0.6 ± 0.8 |
| | 14_5KZL | paired | 44.0 ± 3.8 | 32.6 ± 4.3 | 30.8 ± 2.7 | 22.8 ± 3.0 |
| | 15_4LQ4 | paired | 0.6 ± 0.8 | 0.6 ± 0.8 | 0.6 ± 0.8 | 0.6 ± 0.8 |
| | 16_1SGW | paired | 13.4 ± 4.9 | 13.4 ± 4.9 | 13.4 ± 4.9 | 13.4 ± 4.9 |
| | 17_1BOL | paired | 10.8 ± 2.5 | 10.8 ± 2.5 | 10.8 ± 2.5 | 10.8 ± 2.5 |
| 7 | 18_6W5B | paired | 27.4 ± 3.6 | 25.6 ± 3.9 | 11.6 ± 1.5 | 10.8 ± 1.7 |
| | 19_1Q0S | paired | 11.2 ± 2.7 | 11.2 ± 2.7 | 10.0 ± 2.4 | 10.0 ± 2.4 |
| | 20_4GVW | paired | 28.2 ± 4.0 | 27.0 ± 4.3 | 18.8 ± 2.6 | 18.0 ± 2.9 |
| | 21_3OSX | paired | 2.2 ± 0.7 | 2.2 ± 0.7 | 2.2 ± 0.7 | 2.2 ± 0.7 |
| | 22_1Z6N | paired | 38.4 ± 1.0 | 4.8 ± 2.6 | 38.4 ± 1.0 | 4.8 ± 2.6 |
| 1 | 1_5OJ8 | single | 85.2 ± 2.2 | 85.2 ± 2.2 | 85.2 ± 2.2 | 85.2 ± 2.2 |
| | 2_1TKY | single | 19.4 ± 4.1 | 19.4 ± 4.1 | 19.4 ± 4.1 | 19.4 ± 4.1 |
| | 3_5XJ7 | single | 85.0 ± 2.6 | 85.0 ± 2.6 | 85.0 ± 2.6 | 85.0 ± 2.6 |
| | 4_6KFQ | single | 60.8 ± 4.1 | 60.8 ± 4.1 | 60.8 ± 4.1 | 60.8 ± 4.1 |
| | 5_5URP | single | 56.2 ± 5.0 | 56.2 ± 5.0 | 56.2 ± 5.0 | 56.2 ± 5.0 |
| 2 | 10_6FFV | single | 10.0 ± 2.2 | 10.0 ± 2.2 | 10.0 ± 2.2 | 10.0 ± 2.2 |
| | 6_5XJ7 | single | 41.8 ± 5.9 | 41.8 ± 5.9 | 41.8 ± 5.9 | 41.8 ± 5.9 |
| | 7_5OJ8 | single | 91.8 ± 2.4 | 91.8 ± 2.4 | 91.8 ± 2.4 | 91.8 ± 2.4 |
| | 8_1M2G | single | 4.2 ± 2.1 | 4.2 ± 2.1 | 4.2 ± 2.1 | 4.2 ± 2.1 |
| | 9_5CWP | single | 67.6 ± 1.7 | 67.6 ± 1.7 | 67.6 ± 1.7 | 67.6 ± 1.7 |
| 3 | 11_1Z6N | single | 58.4 ± 6.4 | 58.4 ± 6.4 | 58.4 ± 6.4 | 58.4 ± 6.4 |
| | 12_3P2W | single | 55.4 ± 3.8 | 55.4 ± 3.8 | 55.4 ± 3.8 | 55.4 ± 3.8 |
| | 13_6KFQ | single | 19.8 ± 2.5 | 19.8 ± 2.5 | 19.8 ± 2.5 | 19.8 ± 2.5 |
| | 14_4BJI | single | 57.6 ± 2.3 | 57.6 ± 2.3 | 57.6 ± 2.3 | 57.6 ± 2.3 |
| | 15_1A2J | single | 66.6 ± 2.6 | 66.6 ± 2.6 | 66.6 ± 2.6 | 66.6 ± 2.6 |
| 4 | 16_3PR9 | single | 43.0 ± 3.2 | 43.0 ± 3.2 | 43.0 ± 3.2 | 43.0 ± 3.2 |
| | 17_4GVW | single | 8.2 ± 2.5 | 8.2 ± 2.5 | 8.2 ± 2.5 | 8.2 ± 2.5 |
| | 18_4LQ4 | single | 31.6 ± 2.0 | 31.6 ± 2.0 | 31.6 ± 2.0 | 31.6 ± 2.0 |
| | 19_1M2G | single | 1.6 ± 0.5 | 1.6 ± 0.5 | 1.6 ± 0.5 | 1.6 ± 0.5 |
| | 20_3L86 | single | 20.8 ± 4.7 | 20.8 ± 4.7 | 20.8 ± 4.7 | 20.8 ± 4.7 |
| 5 | 21_1TKY | single | 0.0 ± 0.0 | 0.0 ± 0.0 | 0.0 ± 0.0 | 0.0 ± 0.0 |
| | 22_6TCS | single | 13.8 ± 5.8 | 13.8 ± 5.8 | 13.8 ± 5.8 | 13.8 ± 5.8 |
| | 23_1SGW | single | 10.2 ± 3.4 | 10.2 ± 3.4 | 10.2 ± 3.4 | 10.2 ± 3.4 |
| | 24_6OU0 | single | 6.0 ± 1.8 | 6.0 ± 1.8 | 6.0 ± 1.8 | 6.0 ± 1.8 |
| | 25_4F3H | single | 14.2 ± 3.8 | 14.2 ± 3.8 | 14.2 ± 3.8 | 14.2 ± 3.8 |
| 6 | 26_1GIU | single | 12.6 ± 2.2 | 12.6 ± 2.2 | 12.6 ± 2.2 | 12.6 ± 2.2 |
| | 27_4LQ4 | single | 44.4 ± 2.3 | 44.4 ± 2.3 | 44.4 ± 2.3 | 44.4 ± 2.3 |
| | 28_6TCS | single | 9.6 ± 1.9 | 9.6 ± 1.9 | 9.6 ± 1.9 | 9.6 ± 1.9 |
| | 29_2LAO | single | 0.0 ± 0.0 | 0.0 ± 0.0 | 0.0 ± 0.0 | 0.0 ± 0.0 |
| | 30_6TCS | single | 13.8 ± 2.7 | 13.8 ± 2.7 | 13.8 ± 2.7 | 13.8 ± 2.7 |
| 7 | 31_1GIU | single | 12.2 ± 2.1 | 12.2 ± 2.1 | 12.2 ± 2.1 | 12.2 ± 2.1 |
| | 32_1GBG | single | 17.0 ± 4.7 | 17.0 ± 4.7 | 17.0 ± 4.7 | 17.0 ± 4.7 |
| | 33_6TCS | single | 22.2 ± 2.8 | 22.2 ± 2.8 | 22.2 ± 2.8 | 22.2 ± 2.8 |
| | 34_6TCS | single | 5.2 ± 2.3 | 5.2 ± 2.3 | 5.2 ± 2.3 | 5.2 ± 2.3 |
| | 35_1A2J | single | 9.0 ± 2.8 | 9.0 ± 2.8 | 9.0 ± 2.8 | 9.0 ± 2.8 |

Table 16: **Entry-level metrics for DPLM-3B across all experiments.**

| #Frags | Entry | Experiment | Success Rate | Novel Success (%) | Unique_Clusters | SUN_Score |
|---|---|---|---|---|---|---|
| 3 | 1_5CWP | paired | 0.0 ± 0.0 | 0.0 ± 0.0 | 0.0 ± 0.0 | 0.0 ± 0.0 |
| | 2_5CWN | paired | 0.0 ± 0.0 | 0.0 ± 0.0 | 0.0 ± 0.0 | 0.0 ± 0.0 |
| 4 | 3_4K46 | paired | 38.6 ± 2.6 | 0.0 ± 0.0 | 0.0 ± 0.0 | 0.0 ± 0.0 |
| | 4_5XJ7 | paired | 23.6 ± 2.8 | 0.0 ± 0.0 | 0.0 ± 0.0 | 0.0 ± 0.0 |
| | 5_5OJ8 | paired | 0.0 ± 0.0 | 0.0 ± 0.0 | 0.0 ± 0.0 | 0.0 ± 0.0 |
| | 6_2ZE5 | paired | 0.0 ± 0.0 | 0.0 ± 0.0 | 0.0 ± 0.0 | 0.0 ± 0.0 |
| | 7_1DEX | paired | 1.0 ± 0.9 | 0.0 ± 0.0 | 0.0 ± 0.0 | 0.0 ± 0.0 |
| 5 | 10_5DN1 | paired | 29.0 ± 4.9 | 0.0 ± 0.0 | 0.0 ± 0.0 | 0.0 ± 0.0 |
| | 11_6KFQ | paired | 0.6 ± 0.5 | 0.0 ± 0.0 | 0.0 ± 0.0 | 0.0 ± 0.0 |
| | 12_1HU3 | paired | 0.0 ± 0.0 | 0.0 ± 0.0 | 0.0 ± 0.0 | 0.0 ± 0.0 |
| | 8_1IS1 | paired | 1.6 ± 0.8 | 0.0 ± 0.0 | 0.0 ± 0.0 | 0.0 ± 0.0 |
| | 9_6KFQ | paired | 66.0 ± 3.9 | 0.0 ± 0.0 | 0.0 ± 0.0 | 0.0 ± 0.0 |
| 6 | 13_4BJI | paired | 0.0 ± 0.0 | 0.0 ± 0.0 | 0.0 ± 0.0 | 0.0 ± 0.0 |
| | 14_5KZL | paired | 4.4 ± 1.5 | 0.0 ± 0.0 | 0.0 ± 0.0 | 0.0 ± 0.0 |
| | 15_4LQ4 | paired | 0.0 ± 0.0 | 0.0 ± 0.0 | 0.0 ± 0.0 | 0.0 ± 0.0 |
| | 16_1SGW | paired | 0.0 ± 0.0 | 0.0 ± 0.0 | 0.0 ± 0.0 | 0.0 ± 0.0 |
| | 17_1BOL | paired | 22.4 ± 1.9 | 19.8 ± 1.5 | 12.3 ± 1.0 | 10.9 ± 0.8 |
| 7 | 18_6W5B | paired | 53.6 ± 4.2 | 0.0 ± 0.0 | 0.0 ± 0.0 | 0.0 ± 0.0 |
| | 19_1Q0S | paired | 0.0 ± 0.0 | 0.0 ± 0.0 | 0.0 ± 0.0 | 0.0 ± 0.0 |
| | 20_4GVW | paired | 0.0 ± 0.0 | 0.0 ± 0.0 | 0.0 ± 0.0 | 0.0 ± 0.0 |
| | 21_3OSX | paired | 2.6 ± 2.2 | 0.0 ± 0.0 | 0.0 ± 0.0 | 0.0 ± 0.0 |
| | 22_1Z6N | paired | 0.0 ± 0.0 | 0.0 ± 0.0 | 0.0 ± 0.0 | 0.0 ± 0.0 |
| 1 | 1_5OJ8 | single | 100.0 ± 0.0 | 100.0 ± 0.0 | 11.0 ± 0.0 | 11.0 ± 0.0 |
| | 2_1TKY | single | 100.0 ± 0.0 | 29.0 ± 6.9 | 14.3 ± 0.0 | 4.1 ± 1.0 |
| | 3_5XJ7 | single | 70.8 ± 3.2 | 1.8 ± 1.7 | 70.8 ± 3.2 | 1.8 ± 1.7 |
| | 4_6KFQ | single | 37.6 ± 6.6 | 6.0 ± 0.6 | 18.8 ± 3.3 | 3.0 ± 0.3 |
| | 5_5URP | single | 0.0 ± 0.0 | 0.0 ± 0.0 | 0.0 ± 0.0 | 0.0 ± 0.0 |
| 2 | 10_6FFV | single | 0.0 ± 0.0 | 0.0 ± 0.0 | 0.0 ± 0.0 | 0.0 ± 0.0 |
| | 6_5XJ7 | single | 2.4 ± 1.4 | 0.4 ± 0.5 | 2.4 ± 1.4 | 0.4 ± 0.5 |
| | 7_5OJ8 | single | 45.0 ± 4.0 | 43.2 ± 4.0 | 6.3 ± 0.6 | 6.0 ± 0.6 |
| | 8_1M2G | single | 0.0 ± 0.0 | 0.0 ± 0.0 | 0.0 ± 0.0 | 0.0 ± 0.0 |
| | 9_5CWP | single | 3.8 ± 2.7 | 3.8 ± 2.7 | 3.8 ± 2.7 | 3.8 ± 2.7 |
| 3 | 11_1Z6N | single | 0.0 ± 0.0 | 0.0 ± 0.0 | 0.0 ± 0.0 | 0.0 ± 0.0 |
| | 12_3P2W | single | 0.0 ± 0.0 | 0.0 ± 0.0 | 0.0 ± 0.0 | 0.0 ± 0.0 |
| | 13_6KFQ | single | 17.2 ± 4.7 | 0.0 ± 0.0 | 0.0 ± 0.0 | 0.0 ± 0.0 |
| | 14_4BJI | single | 29.2 ± 2.8 | 14.8 ± 1.5 | 20.1 ± 1.9 | 10.2 ± 1.0 |
| | 15_1A2J | single | 1.6 ± 0.8 | 1.0 ± 0.6 | 1.6 ± 0.8 | 1.0 ± 0.6 |
| 4 | 16_3PR9 | single | 77.2 ± 3.5 | 27.6 ± 2.3 | 42.6 ± 1.9 | 15.2 ± 1.3 |
| | 17_4GVW | single | 3.8 ± 1.9 | 0.0 ± 0.0 | 0.0 ± 0.0 | 0.0 ± 0.0 |
| | 18_4LQ4 | single | 29.8 ± 3.5 | 7.0 ± 1.7 | 24.8 ± 2.9 | 5.8 ± 1.4 |
| | 19_1M2G | single | 2.2 ± 1.2 | 0.0 ± 0.0 | 0.0 ± 0.0 | 0.0 ± 0.0 |
| | 20_3L86 | single | 3.0 ± 0.6 | 0.0 ± 0.0 | 0.0 ± 0.0 | 0.0 ± 0.0 |
| 5 | 21_1TKY | single | 0.6 ± 0.8 | 0.0 ± 0.0 | 0.0 ± 0.0 | 0.0 ± 0.0 |
| | 22_6TCS | single | 11.0 ± 2.0 | 11.0 ± 2.0 | 11.0 ± 2.0 | 11.0 ± 2.0 |
| | 23_1SGW | single | 0.8 ± 0.7 | 0.0 ± 0.0 | 0.0 ± 0.0 | 0.0 ± 0.0 |
| | 24_6OU0 | single | 0.0 ± 0.0 | 0.0 ± 0.0 | 0.0 ± 0.0 | 0.0 ± 0.0 |
| | 25_4F3H | single | 7.0 ± 0.9 | 3.6 ± 1.6 | 4.7 ± 0.6 | 2.4 ± 1.1 |
| 6 | 26_1GIU | single | 0.0 ± 0.0 | 0.0 ± 0.0 | 0.0 ± 0.0 | 0.0 ± 0.0 |
| | 27_4LQ4 | single | 4.6 ± 2.4 | 0.0 ± 0.0 | 0.0 ± 0.0 | 0.0 ± 0.0 |
| | 28_6TCS | single | 3.4 ± 2.4 | 3.4 ± 2.4 | 3.4 ± 2.4 | 3.4 ± 2.4 |
| | 29_2LAO | single | 0.0 ± 0.0 | 0.0 ± 0.0 | 0.0 ± 0.0 | 0.0 ± 0.0 |
| | 30_6TCS | single | 47.2 ± 2.4 | 47.2 ± 2.4 | 36.7 ± 1.9 | 36.7 ± 1.9 |
| 7 | 31_1GIU | single | 0.0 ± 0.0 | 0.0 ± 0.0 | 0.0 ± 0.0 | 0.0 ± 0.0 |
| | 32_1GBG | single | 22.4 ± 3.0 | 6.0 ± 2.3 | 14.9 ± 2.0 | 4.0 ± 1.5 |
| | 33_6TCS | single | 18.4 ± 4.1 | 18.4 ± 4.1 | 18.4 ± 4.1 | 18.4 ± 4.1 |
| | 34_6TCS | single | 33.6 ± 5.1 | 32.6 ± 4.4 | 29.6 ± 4.5 | 28.8 ± 3.9 |
| | 35_1A2J | single | 1.4 ± 1.0 | 1.4 ± 1.0 | 1.4 ± 1.0 | 1.4 ± 1.0 |

Table 17: **Entry-level metrics for DPLM-650M across all experiments.**

| #Frags | Entry | Experiment | Success Rate | Novel Success (%) | Unique_Clusters | SUN_Score |
|---|---|---|---|---|---|---|
| 3 | 1_5CWP | paired | 0.0 ± 0.0 | 0.0 ± 0.0 | 0.0 ± 0.0 | 0.0 ± 0.0 |
| | 2_5CWN | paired | 0.0 ± 0.0 | 0.0 ± 0.0 | 0.0 ± 0.0 | 0.0 ± 0.0 |
| 4 | 3_4K46 | paired | 27.6 ± 3.6 | 0.0 ± 0.0 | 0.0 ± 0.0 | 0.0 ± 0.0 |
| | 4_5XJ7 | paired | 7.4 ± 1.9 | 0.0 ± 0.0 | 0.0 ± 0.0 | 0.0 ± 0.0 |
| | 5_5OJ8 | paired | 0.0 ± 0.0 | 0.0 ± 0.0 | 0.0 ± 0.0 | 0.0 ± 0.0 |
| | 6_2ZE5 | paired | 0.0 ± 0.0 | 0.0 ± 0.0 | 0.0 ± 0.0 | 0.0 ± 0.0 |
| | 7_1DEX | paired | 2.0 ± 1.4 | 0.0 ± 0.0 | 0.0 ± 0.0 | 0.0 ± 0.0 |
| 5 | 10_5DN1 | paired | 7.4 ± 2.7 | 0.0 ± 0.0 | 0.0 ± 0.0 | 0.0 ± 0.0 |
| | 11_6KFQ | paired | 8.6 ± 1.6 | 0.0 ± 0.0 | 0.0 ± 0.0 | 0.0 ± 0.0 |
| | 12_1HU3 | paired | 0.0 ± 0.0 | 0.0 ± 0.0 | 0.0 ± 0.0 | 0.0 ± 0.0 |
| | 8_1IS1 | paired | 1.2 ± 1.0 | 0.0 ± 0.0 | 0.0 ± 0.0 | 0.0 ± 0.0 |
| | 9_6KFQ | paired | 26.6 ± 3.8 | 0.0 ± 0.0 | 0.0 ± 0.0 | 0.0 ± 0.0 |
| 6 | 13_4BJI | paired | 0.0 ± 0.0 | 0.0 ± 0.0 | 0.0 ± 0.0 | 0.0 ± 0.0 |
| | 14_5KZL | paired | 8.0 ± 3.7 | 0.0 ± 0.0 | 0.0 ± 0.0 | 0.0 ± 0.0 |
| | 15_4LQ4 | paired | 0.0 ± 0.0 | 0.0 ± 0.0 | 0.0 ± 0.0 | 0.0 ± 0.0 |
| | 16_1SGW | paired | 0.0 ± 0.0 | 0.0 ± 0.0 | 0.0 ± 0.0 | 0.0 ± 0.0 |
| | 17_1BOL | paired | 11.8 ± 3.1 | 8.0 ± 2.1 | 7.4 ± 2.0 | 5.0 ± 1.3 |
| 7 | 18_6W5B | paired | 45.0 ± 5.7 | 0.0 ± 0.0 | 0.0 ± 0.0 | 0.0 ± 0.0 |
| | 19_1Q0S | paired | 0.0 ± 0.0 | 0.0 ± 0.0 | 0.0 ± 0.0 | 0.0 ± 0.0 |
| | 20_4GVW | paired | 0.0 ± 0.0 | 0.0 ± 0.0 | 0.0 ± 0.0 | 0.0 ± 0.0 |
| | 21_3OSX | paired | 0.0 ± 0.0 | 0.0 ± 0.0 | 0.0 ± 0.0 | 0.0 ± 0.0 |
| | 22_1Z6N | paired | 0.0 ± 0.0 | 0.0 ± 0.0 | 0.0 ± 0.0 | 0.0 ± 0.0 |
| 1 | 1_5OJ8 | single | 100.0 ± 0.0 | 100.0 ± 0.0 | 11.0 ± 0.0 | 11.0 ± 0.0 |
| | 2_1TKY | single | 100.0 ± 0.0 | 37.4 ± 1.5 | 32.4 ± 0.0 | 12.1 ± 0.5 |
| | 3_5XJ7 | single | 85.0 ± 5.3 | 7.2 ± 2.8 | 56.7 ± 3.6 | 4.8 ± 1.9 |
| | 4_6KFQ | single | 34.2 ± 5.0 | 3.4 ± 0.5 | 22.8 ± 3.3 | 2.3 ± 0.3 |
| | 5_5URP | single | 0.0 ± 0.0 | 0.0 ± 0.0 | 0.0 ± 0.0 | 0.0 ± 0.0 |
| 2 | 10_6FFV | single | 0.0 ± 0.0 | 0.0 ± 0.0 | 0.0 ± 0.0 | 0.0 ± 0.0 |
| | 6_5XJ7 | single | 0.0 ± 0.0 | 0.0 ± 0.0 | 0.0 ± 0.0 | 0.0 ± 0.0 |
| | 7_5OJ8 | single | 30.4 ± 6.3 | 30.4 ± 6.3 | 7.9 ± 1.6 | 7.9 ± 1.6 |
| | 8_1M2G | single | 6.0 ± 3.0 | 0.0 ± 0.0 | 0.0 ± 0.0 | 0.0 ± 0.0 |
| | 9_5CWP | single | 1.2 ± 0.7 | 1.2 ± 0.7 | 1.2 ± 0.7 | 1.2 ± 0.7 |
| 3 | 11_1Z6N | single | 0.0 ± 0.0 | 0.0 ± 0.0 | 0.0 ± 0.0 | 0.0 ± 0.0 |
| | 12_3P2W | single | 0.0 ± 0.0 | 0.0 ± 0.0 | 0.0 ± 0.0 | 0.0 ± 0.0 |
| | 13_6KFQ | single | 0.0 ± 0.0 | 0.0 ± 0.0 | 0.0 ± 0.0 | 0.0 ± 0.0 |
| | 14_4BJI | single | 21.8 ± 4.3 | 6.6 ± 2.2 | 13.6 ± 2.7 | 4.1 ± 1.4 |
| | 15_1A2J | single | 0.0 ± 0.0 | 0.0 ± 0.0 | 0.0 ± 0.0 | 0.0 ± 0.0 |
| 4 | 16_3PR9 | single | 11.4 ± 3.1 | 3.6 ± 1.9 | 9.1 ± 2.5 | 2.9 ± 1.5 |
| | 17_4GVW | single | 4.6 ± 2.1 | 0.0 ± 0.0 | 0.0 ± 0.0 | 0.0 ± 0.0 |
| | 18_4LQ4 | single | 11.6 ± 2.8 | 2.8 ± 1.2 | 11.6 ± 2.8 | 2.8 ± 1.2 |
| | 19_1M2G | single | 0.0 ± 0.0 | 0.0 ± 0.0 | 0.0 ± 0.0 | 0.0 ± 0.0 |
| | 20_3L86 | single | 0.0 ± 0.0 | 0.0 ± 0.0 | 0.0 ± 0.0 | 0.0 ± 0.0 |
| 5 | 21_1TKY | single | 0.4 ± 0.8 | 0.0 ± 0.0 | 0.0 ± 0.0 | 0.0 ± 0.0 |
| | 22_6TCS | single | 8.2 ± 1.2 | 8.2 ± 1.2 | 7.2 ± 1.0 | 7.2 ± 1.0 |
| | 23_1SGW | single | 0.0 ± 0.0 | 0.0 ± 0.0 | 0.0 ± 0.0 | 0.0 ± 0.0 |
| | 24_6OU0 | single | 0.0 ± 0.0 | 0.0 ± 0.0 | 0.0 ± 0.0 | 0.0 ± 0.0 |
| | 25_4F3H | single | 7.2 ± 2.8 | 2.4 ± 1.4 | 3.6 ± 1.4 | 1.2 ± 0.7 |
| 6 | 26_1GIU | single | 0.0 ± 0.0 | 0.0 ± 0.0 | 0.0 ± 0.0 | 0.0 ± 0.0 |
| | 27_4LQ4 | single | 3.0 ± 2.3 | 0.0 ± 0.0 | 0.0 ± 0.0 | 0.0 ± 0.0 |
| | 28_6TCS | single | 0.8 ± 0.7 | 0.8 ± 0.7 | 0.8 ± 0.7 | 0.8 ± 0.7 |
| | 29_2LAO | single | 0.0 ± 0.0 | 0.0 ± 0.0 | 0.0 ± 0.0 | 0.0 ± 0.0 |
| | 30_6TCS | single | 51.4 ± 2.9 | 51.4 ± 2.9 | 40.3 ± 2.3 | 40.3 ± 2.3 |
| 7 | 31_1GIU | single | 0.0 ± 0.0 | 0.0 ± 0.0 | 0.0 ± 0.0 | 0.0 ± 0.0 |
| | 32_1GBG | single | 10.4 ± 4.3 | 0.2 ± 0.4 | 10.4 ± 4.3 | 0.2 ± 0.4 |
| | 33_6TCS | single | 7.8 ± 2.9 | 7.8 ± 2.9 | 5.2 ± 2.0 | 5.2 ± 2.0 |
| | 34_6TCS | single | 30.0 ± 3.3 | 27.4 ± 3.7 | 25.7 ± 2.8 | 23.5 ± 3.1 |
| | 35_1A2J | single | 6.8 ± 1.2 | 5.4 ± 1.6 | 4.5 ± 0.8 | 3.6 ± 1.1 |

Table 18: **Entry-level metrics for ESM3 (seq & struct) across all experiments.**

| #Frags | Entry | Experiment | Success Rate | Novel Success (%) | Unique_Clusters | SUN_Score |
|---|---|---|---|---|---|---|
| 3 | 1_5CWP | paired | 0.0 ± 0.0 | 0.0 ± 0.0 | 0.0 ± 0.0 | 0.0 ± 0.0 |
| | 2_5CWN | paired | 0.0 ± 0.0 | 0.0 ± 0.0 | 0.0 ± 0.0 | 0.0 ± 0.0 |
| 4 | 3_4K46 | paired | 0.0 ± 0.0 | 0.0 ± 0.0 | 0.0 ± 0.0 | 0.0 ± 0.0 |
| | 4_5XJ7 | paired | 5.8 ± 1.7 | 0.0 ± 0.0 | 0.0 ± 0.0 | 0.0 ± 0.0 |
| | 5_5OJ8 | paired | 0.0 ± 0.0 | 0.0 ± 0.0 | 0.0 ± 0.0 | 0.0 ± 0.0 |
| | 6_2ZE5 | paired | 0.0 ± 0.0 | 0.0 ± 0.0 | 0.0 ± 0.0 | 0.0 ± 0.0 |
| | 7_1DEX | paired | 0.0 ± 0.0 | 0.0 ± 0.0 | 0.0 ± 0.0 | 0.0 ± 0.0 |
| 5 | 10_5DN1 | paired | 10.0 ± 2.1 | 0.0 ± 0.0 | 0.0 ± 0.0 | 0.0 ± 0.0 |
| | 11_6KFQ | paired | 0.0 ± 0.0 | 0.0 ± 0.0 | 0.0 ± 0.0 | 0.0 ± 0.0 |
| | 12_1HU3 | paired | 0.0 ± 0.0 | 0.0 ± 0.0 | 0.0 ± 0.0 | 0.0 ± 0.0 |
| | 8_1IS1 | paired | 0.0 ± 0.0 | 0.0 ± 0.0 | 0.0 ± 0.0 | 0.0 ± 0.0 |
| | 9_6KFQ | paired | 0.6 ± 0.8 | 0.0 ± 0.0 | 0.0 ± 0.0 | 0.0 ± 0.0 |
| 6 | 13_4BJI | paired | 0.0 ± 0.0 | 0.0 ± 0.0 | 0.0 ± 0.0 | 0.0 ± 0.0 |
| | 14_5KZL | paired | 0.0 ± 0.0 | 0.0 ± 0.0 | 0.0 ± 0.0 | 0.0 ± 0.0 |
| | 15_4LQ4 | paired | 0.0 ± 0.0 | 0.0 ± 0.0 | 0.0 ± 0.0 | 0.0 ± 0.0 |
| | 16_1SGW | paired | 0.0 ± 0.0 | 0.0 ± 0.0 | 0.0 ± 0.0 | 0.0 ± 0.0 |
| | 17_1BOL | paired | 0.0 ± 0.0 | 0.0 ± 0.0 | 0.0 ± 0.0 | 0.0 ± 0.0 |
| 7 | 18_6W5B | paired | 0.0 ± 0.0 | 0.0 ± 0.0 | 0.0 ± 0.0 | 0.0 ± 0.0 |
| | 19_1Q0S | paired | 0.0 ± 0.0 | 0.0 ± 0.0 | 0.0 ± 0.0 | 0.0 ± 0.0 |
| | 20_4GVW | paired | 0.0 ± 0.0 | 0.0 ± 0.0 | 0.0 ± 0.0 | 0.0 ± 0.0 |
| | 21_3OSX | paired | 0.0 ± 0.0 | 0.0 ± 0.0 | 0.0 ± 0.0 | 0.0 ± 0.0 |
| | 22_1Z6N | paired | 0.0 ± 0.0 | 0.0 ± 0.0 | 0.0 ± 0.0 | 0.0 ± 0.0 |
| 1 | 1_5OJ8 | single | 35.8 ± 3.3 | 35.8 ± 3.3 | 35.8 ± 3.3 | 35.8 ± 3.3 |
| | 2_1TKY | single | 2.0 ± 1.3 | 2.0 ± 1.3 | 2.0 ± 1.3 | 2.0 ± 1.3 |
| | 3_5XJ7 | single | 0.0 ± 0.0 | 0.0 ± 0.0 | 0.0 ± 0.0 | 0.0 ± 0.0 |
| | 4_6KFQ | single | 0.0 ± 0.0 | 0.0 ± 0.0 | 0.0 ± 0.0 | 0.0 ± 0.0 |
| | 5_5URP | single | 0.0 ± 0.0 | 0.0 ± 0.0 | 0.0 ± 0.0 | 0.0 ± 0.0 |
| 2 | 10_6FFV | single | 0.0 ± 0.0 | 0.0 ± 0.0 | 0.0 ± 0.0 | 0.0 ± 0.0 |
| | 6_5XJ7 | single | 3.4 ± 2.2 | 3.4 ± 2.2 | 3.4 ± 2.2 | 3.4 ± 2.2 |
| | 7_5OJ8 | single | 0.0 ± 0.0 | 0.0 ± 0.0 | 0.0 ± 0.0 | 0.0 ± 0.0 |
| | 8_1M2G | single | 0.0 ± 0.0 | 0.0 ± 0.0 | 0.0 ± 0.0 | 0.0 ± 0.0 |
| | 9_5CWP | single | 12.8 ± 2.4 | 12.8 ± 2.4 | 12.8 ± 2.4 | 12.8 ± 2.4 |
| 3 | 11_1Z6N | single | 0.0 ± 0.0 | 0.0 ± 0.0 | 0.0 ± 0.0 | 0.0 ± 0.0 |
| | 12_3P2W | single | 0.0 ± 0.0 | 0.0 ± 0.0 | 0.0 ± 0.0 | 0.0 ± 0.0 |
| | 13_6KFQ | single | 0.0 ± 0.0 | 0.0 ± 0.0 | 0.0 ± 0.0 | 0.0 ± 0.0 |
| | 14_4BJI | single | 2.2 ± 1.2 | 2.2 ± 1.2 | 2.2 ± 1.2 | 2.2 ± 1.2 |
| | 15_1A2J | single | 0.0 ± 0.0 | 0.0 ± 0.0 | 0.0 ± 0.0 | 0.0 ± 0.0 |
| 4 | 16_3PR9 | single | 0.0 ± 0.0 | 0.0 ± 0.0 | 0.0 ± 0.0 | 0.0 ± 0.0 |
| | 17_4GVW | single | 0.0 ± 0.0 | 0.0 ± 0.0 | 0.0 ± 0.0 | 0.0 ± 0.0 |
| | 18_4LQ4 | single | 0.0 ± 0.0 | 0.0 ± 0.0 | 0.0 ± 0.0 | 0.0 ± 0.0 |
| | 19_1M2G | single | 0.0 ± 0.0 | 0.0 ± 0.0 | 0.0 ± 0.0 | 0.0 ± 0.0 |
| | 20_3L86 | single | 0.0 ± 0.0 | 0.0 ± 0.0 | 0.0 ± 0.0 | 0.0 ± 0.0 |
| 5 | 21_1TKY | single | 0.0 ± 0.0 | 0.0 ± 0.0 | 0.0 ± 0.0 | 0.0 ± 0.0 |
| | 22_6TCS | single | 0.0 ± 0.0 | 0.0 ± 0.0 | 0.0 ± 0.0 | 0.0 ± 0.0 |
| | 23_1SGW | single | 0.0 ± 0.0 | 0.0 ± 0.0 | 0.0 ± 0.0 | 0.0 ± 0.0 |
| | 24_6OU0 | single | 0.0 ± 0.0 | 0.0 ± 0.0 | 0.0 ± 0.0 | 0.0 ± 0.0 |
| | 25_4F3H | single | 0.0 ± 0.0 | 0.0 ± 0.0 | 0.0 ± 0.0 | 0.0 ± 0.0 |
| 6 | 26_1GIU | single | 0.0 ± 0.0 | 0.0 ± 0.0 | 0.0 ± 0.0 | 0.0 ± 0.0 |
| | 27_4LQ4 | single | 1.0 ± 1.1 | 1.0 ± 1.1 | 1.0 ± 1.1 | 1.0 ± 1.1 |
| | 28_6TCS | single | 0.0 ± 0.0 | 0.0 ± 0.0 | 0.0 ± 0.0 | 0.0 ± 0.0 |
| | 29_2LAO | single | 0.0 ± 0.0 | 0.0 ± 0.0 | 0.0 ± 0.0 | 0.0 ± 0.0 |
| | 30_6TCS | single | 9.4 ± 2.6 | 9.4 ± 2.6 | 9.4 ± 2.6 | 9.4 ± 2.6 |
| 7 | 31_1GIU | single | 0.0 ± 0.0 | 0.0 ± 0.0 | 0.0 ± 0.0 | 0.0 ± 0.0 |
| | 32_1GBG | single | 0.0 ± 0.0 | 0.0 ± 0.0 | 0.0 ± 0.0 | 0.0 ± 0.0 |
| | 33_6TCS | single | 0.6 ± 0.8 | 0.6 ± 0.8 | 0.6 ± 0.8 | 0.6 ± 0.8 |
| | 34_6TCS | single | 9.4 ± 3.4 | 9.4 ± 3.4 | 9.4 ± 3.4 | 9.4 ± 3.4 |
| | 35_1A2J | single | 0.0 ± 0.0 | 0.0 ± 0.0 | 0.0 ± 0.0 | 0.0 ± 0.0 |

Table 19: **Entry-level metrics for ESM3 (seq) across all experiments.**

| #Frags | Entry | Experiment | Success Rate | Novel Success (%) | Unique_Clusters | SUN_Score |
|---|---|---|---|---|---|---|
| 3 | 1_5CWP | paired | 0.0 ± 0.0 | 0.0 ± 0.0 | 0.0 ± 0.0 | 0.0 ± 0.0 |
| | 2_5CWN | paired | 1.4 ± 1.2 | 1.4 ± 1.2 | 1.4 ± 1.2 | 1.4 ± 1.2 |
| 4 | 3_4K46 | paired | 39.8 ± 5.6 | 0.0 ± 0.0 | 0.0 ± 0.0 | 0.0 ± 0.0 |
| | 4_5XJ7 | paired | 16.6 ± 3.7 | 0.0 ± 0.0 | 0.0 ± 0.0 | 0.0 ± 0.0 |
| | 5_5OJ8 | paired | 0.0 ± 0.0 | 0.0 ± 0.0 | 0.0 ± 0.0 | 0.0 ± 0.0 |
| | 6_2ZE5 | paired | 0.0 ± 0.0 | 0.0 ± 0.0 | 0.0 ± 0.0 | 0.0 ± 0.0 |
| | 7_1DEX | paired | 0.0 ± 0.0 | 0.0 ± 0.0 | 0.0 ± 0.0 | 0.0 ± 0.0 |
| 5 | 10_5DN1 | paired | 38.8 ± 3.2 | 0.0 ± 0.0 | 0.0 ± 0.0 | 0.0 ± 0.0 |
| | 11_6KFQ | paired | 4.0 ± 0.9 | 0.0 ± 0.0 | 0.0 ± 0.0 | 0.0 ± 0.0 |
| | 12_1HU3 | paired | 0.0 ± 0.0 | 0.0 ± 0.0 | 0.0 ± 0.0 | 0.0 ± 0.0 |
| | 8_1IS1 | paired | 2.2 ± 1.5 | 0.0 ± 0.0 | 0.0 ± 0.0 | 0.0 ± 0.0 |
| | 9_6KFQ | paired | 14.8 ± 3.7 | 0.0 ± 0.0 | 0.0 ± 0.0 | 0.0 ± 0.0 |
| 6 | 13_4BJI | paired | 0.0 ± 0.0 | 0.0 ± 0.0 | 0.0 ± 0.0 | 0.0 ± 0.0 |
| | 14_5KZL | paired | 2.8 ± 1.2 | 0.0 ± 0.0 | 0.0 ± 0.0 | 0.0 ± 0.0 |
| | 15_4LQ4 | paired | 0.0 ± 0.0 | 0.0 ± 0.0 | 0.0 ± 0.0 | 0.0 ± 0.0 |
| | 16_1SGW | paired | 0.0 ± 0.0 | 0.0 ± 0.0 | 0.0 ± 0.0 | 0.0 ± 0.0 |
| | 17_1BOL | paired | 0.0 ± 0.0 | 0.0 ± 0.0 | 0.0 ± 0.0 | 0.0 ± 0.0 |
| 7 | 18_6W5B | paired | 29.8 ± 3.3 | 0.0 ± 0.0 | 0.0 ± 0.0 | 0.0 ± 0.0 |
| | 19_1Q0S | paired | 1.6 ± 1.4 | 0.0 ± 0.0 | 0.0 ± 0.0 | 0.0 ± 0.0 |
| | 20_4GVW | paired | 0.0 ± 0.0 | 0.0 ± 0.0 | 0.0 ± 0.0 | 0.0 ± 0.0 |
| | 21_3OSX | paired | 7.0 ± 2.3 | 0.0 ± 0.0 | 0.0 ± 0.0 | 0.0 ± 0.0 |
| | 22_1Z6N | paired | 0.0 ± 0.0 | 0.0 ± 0.0 | 0.0 ± 0.0 | 0.0 ± 0.0 |
| 1 | 1_5OJ8 | single | 100.0 ± 0.0 | 95.2 ± 1.7 | 18.8 ± 0.0 | 17.9 ± 0.3 |
| | 2_1TKY | single | 30.6 ± 2.0 | 30.6 ± 2.0 | 26.5 ± 1.7 | 26.5 ± 1.7 |
| | 3_5XJ7 | single | 91.2 ± 4.8 | 14.8 ± 3.4 | 32.6 ± 1.7 | 5.3 ± 1.2 |
| | 4_6KFQ | single | 18.4 ± 3.9 | 16.6 ± 3.4 | 11.5 ± 2.5 | 10.4 ± 2.1 |
| | 5_5URP | single | 0.0 ± 0.0 | 0.0 ± 0.0 | 0.0 ± 0.0 | 0.0 ± 0.0 |
| 2 | 10_6FFV | single | 0.0 ± 0.0 | 0.0 ± 0.0 | 0.0 ± 0.0 | 0.0 ± 0.0 |
| | 6_5XJ7 | single | 64.4 ± 3.0 | 52.4 ± 3.0 | 37.7 ± 1.8 | 30.6 ± 1.8 |
| | 7_5OJ8 | single | 7.6 ± 0.8 | 7.6 ± 0.8 | 3.8 ± 0.4 | 3.8 ± 0.4 |
| | 8_1M2G | single | 10.0 ± 1.4 | 3.6 ± 1.4 | 7.5 ± 1.1 | 2.7 ± 1.0 |
| | 9_5CWP | single | 0.0 ± 0.0 | 0.0 ± 0.0 | 0.0 ± 0.0 | 0.0 ± 0.0 |
| 3 | 11_1Z6N | single | 0.0 ± 0.0 | 0.0 ± 0.0 | 0.0 ± 0.0 | 0.0 ± 0.0 |
| | 12_3P2W | single | 0.0 ± 0.0 | 0.0 ± 0.0 | 0.0 ± 0.0 | 0.0 ± 0.0 |
| | 13_6KFQ | single | 0.0 ± 0.0 | 0.0 ± 0.0 | 0.0 ± 0.0 | 0.0 ± 0.0 |
| | 14_4BJI | single | 4.0 ± 2.2 | 4.0 ± 2.2 | 4.0 ± 2.2 | 4.0 ± 2.2 |
| | 15_1A2J | single | 0.0 ± 0.0 | 0.0 ± 0.0 | 0.0 ± 0.0 | 0.0 ± 0.0 |
| 4 | 16_3PR9 | single | 69.6 ± 3.3 | 8.4 ± 1.9 | 55.7 ± 2.6 | 6.7 ± 1.5 |
| | 17_4GVW | single | 1.2 ± 1.5 | 1.2 ± 1.5 | 1.2 ± 1.5 | 1.2 ± 1.5 |
| | 18_4LQ4 | single | 18.6 ± 4.4 | 6.6 ± 1.2 | 13.9 ± 3.3 | 5.0 ± 0.9 |
| | 19_1M2G | single | 7.6 ± 2.7 | 0.0 ± 0.0 | 0.0 ± 0.0 | 0.0 ± 0.0 |
| | 20_3L86 | single | 0.2 ± 0.4 | 0.0 ± 0.0 | 0.0 ± 0.0 | 0.0 ± 0.0 |
| 5 | 21_1TKY | single | 2.6 ± 0.8 | 0.0 ± 0.0 | 0.0 ± 0.0 | 0.0 ± 0.0 |
| | 22_6TCS | single | 11.8 ± 2.9 | 11.8 ± 2.9 | 11.8 ± 2.9 | 11.8 ± 2.9 |
| | 23_1SGW | single | 0.0 ± 0.0 | 0.0 ± 0.0 | 0.0 ± 0.0 | 0.0 ± 0.0 |
| | 24_6OU0 | single | 0.0 ± 0.0 | 0.0 ± 0.0 | 0.0 ± 0.0 | 0.0 ± 0.0 |
| | 25_4F3H | single | 21.2 ± 1.8 | 13.6 ± 1.9 | 10.6 ± 0.9 | 6.8 ± 0.9 |
| 6 | 26_1GIU | single | 0.0 ± 0.0 | 0.0 ± 0.0 | 0.0 ± 0.0 | 0.0 ± 0.0 |
| | 27_4LQ4 | single | 2.4 ± 1.4 | 0.0 ± 0.0 | 0.0 ± 0.0 | 0.0 ± 0.0 |
| | 28_6TCS | single | 7.8 ± 3.0 | 7.8 ± 3.0 | 6.7 ± 2.6 | 6.7 ± 2.6 |
| | 29_2LAO | single | 0.0 ± 0.0 | 0.0 ± 0.0 | 0.0 ± 0.0 | 0.0 ± 0.0 |
| | 30_6TCS | single | 48.0 ± 5.1 | 48.0 ± 5.1 | 33.7 ± 3.6 | 33.7 ± 3.6 |
| 7 | 31_1GIU | single | 0.0 ± 0.0 | 0.0 ± 0.0 | 0.0 ± 0.0 | 0.0 ± 0.0 |
| | 32_1GBG | single | 34.4 ± 2.4 | 9.8 ± 1.2 | 21.5 ± 1.5 | 6.1 ± 0.7 |
| | 33_6TCS | single | 19.8 ± 5.4 | 19.8 ± 5.4 | 18.8 ± 5.1 | 18.8 ± 5.1 |
| | 34_6TCS | single | 32.6 ± 4.3 | 32.6 ± 4.3 | 28.8 ± 3.8 | 28.8 ± 3.8 |
| | 35_1A2J | single | 5.0 ± 1.9 | 4.2 ± 1.5 | 5.0 ± 1.9 | 4.2 ± 1.5 |

