# OpenReview forum: "GeomMotif: A Benchmark for Arbitrary Geometric Preservation in Protein Generation"
_ICLR.cc/2026/Conference — ICLR 2026 Poster_

### Official Review · Reviewer_oyxS · 2025-10-25

**Soundness:** 4
**Presentation:** 3
**Contribution:** 2
**Rating:** 6
**Confidence:** 4

**Summary:**

The authors introduce GeomMotif, a new benchmark to better evaluate the performance of current motif-scaffolding techniques in de-novo protein design. The benchmark consists of 57 modality-agnostic scaffolding tasks. To ensure systematic coverage of protein structural space, the authors uniformly sample proteins from the PDB containing varying numbers of motifs, ranging from single-motifs to multi-fragment ones with up to 7 fragments. Each motif and fragment is annotated with physicochemical metadata as well.
For each task, the authors generate scaffolds using multiple methods, demonstrating that the motifs or fragments are indeed "scaffoldable" and hence the tasks solvable. Interestingly, when comparing structure- and sequence-based models, the authors find that structure models outperform sequence-based ones. Sequence-based models struggle particularly with multi-fragment, non-local motifs. They also observe that the secondary structure and other parameters of motifs strongly influences success. Helices are easier to scaffold than β-sheets, and that burial and contact ratio are important determinants of performance. The GeomMotif benchmark is publicly available on github and hugging face for use by the broader research community.

**Strengths:**

The manuscript is clearly written, and the creation pipeline of GeomMotif is well explained. The benchmark evaluation incl. its feasibility and the comparison between structure-based and sequence-only models reveals several interesting trends, highlighting both challenges and opportunities for these approaches. As expected, structure-based models outperform sequence-only models on this structurally oriented task. Moreover, motifs containing helices are easier to scaffold, a finding consistent with previous observations.

**Weaknesses:**

No specific weaknesses in the manuscript.

**Questions:**

I have two questions regarding the benchmark creation:

1. Many motifs include loops or parts of loops. Why did the authors remove these when creating the benchmark instead of including and classifying them as well? This aspect would be important for multiple applications; antibodies being the most obvious example.
2. Does the benchmark include motifs derived from actual protein–protein interactions (PPIs) or enzymes? If so, for the PPI-derived motifs are the corresponding target proteins also included? It would be interesting to see scaffolding evaluated in the presence of the target protein, as this introduces additional constraints on how the scaffold must be generated.

---

> ### Author Response · Authors · 2025-11-26
> **Official Comment by Authors**
>
> Thank you for your thoughtful review and for recognizing the systematic approach and clear presentation of GeomMotif. Below we address your questions about the benchmark design.
>
> > **Q1.** Many motifs include loops or parts of loops. Why did the authors remove these when creating the benchmark instead of including and classifying them as well? This aspect would be important for multiple applications; antibodies being the most obvious example.
>
> We want to clarify that **loops were not removed** from GeomMotif. We applied a threshold permitting **motifs with up to 25% loop content** while filtering those exceeding this limit. Our tasks span loop content from 0% to 23.3% (Table 2 in revised manuscript), with many tasks containing substantial unstructured regions (e.g., 2_1TKY at 23.3%, 5_5URP at 22.9%).
>
> This threshold directly supports our **guaranteed solvability requirement**. Every GeomMotif task has a verified solution by our evaluation pipeline. We pre-filter all benchmark proteins to confirm ESMFold can accurately fold them (RMSD ≤ 1.0 Å). Motifs dominated by loops lack well-defined tertiary structure and are notoriously difficult for structure prediction models to fold accurately. Including high loop content would compromise this guarantee, making it impossible to distinguish between model limitations and inherently unsolvable tasks. This issue has plagued previous benchmarks (as noted by Zheng et al., 2025 in MotifBench).
>
> Moreover, **models still face substantial loop design challenges** in our benchmark. When motifs contain multiple non-contiguous fragments, models must generate connecting regions that naturally include loops, turns, and flexible linkers between the fixed structural elements.
>
> ---
>
> > **Q2.** Does the benchmark include motifs derived from actual protein–protein interactions (PPIs) or enzymes? If so, for the PPI-derived motifs are the corresponding target proteins also included?
>
> **We intentionally excluded PPIs and functional sites** to isolate geometric preservation from functional constraints. This design choice has two components.
>
> For PPIs, **we construct tasks exclusively from monomeric structures** (Section 3.2). Proteins in complexes often adopt different conformations in isolation, creating conformational ambiguity that would compromise evaluation consistency. When comparing generated structures against ground truth, we need the target geometry to be stable and well-defined independent of binding partners. Developing benchmarks for multimeric systems represents important future work that requires different methodological approaches to handle conformational flexibility and interface-dependent structural changes.
>
> For functional motifs more broadly, **GeomMotif takes a complementary approach** to existing benchmarks. The key difference in task design is that we systematically sample structures with diverse geometric contexts and characterize them across multiple dimensions (single/paired motifs, fragment counts, physicochemical properties) to maximize coverage of protein structural space. In contrast, RFdiffusion, MotifBench, and the recent Atomic Motif Enzyme Benchmark manually select known functional sites based on their biological importance. Both approaches provide valuable but different perspectives. Our geometric focus allows us to assess whether models can preserve arbitrary structural arrangements across varied contexts, revealing patterns like how secondary structure composition and burial affect scaffolding difficulty. This complements functional benchmarks that evaluate whether models can satisfy the specific physicochemical requirements of biologically validated sites.
>
> ---
>
> *We hope these responses address your concerns. If you find them satisfactory, we would be grateful if you would consider raising your rating. Please let us know if there is anything else we can clarify.*

---

### Official Review · Reviewer_aTcU · 2025-10-29

**Soundness:** 3
**Presentation:** 3
**Contribution:** 3
**Rating:** 6
**Confidence:** 3

**Summary:**

This paper introduces **GeomMotif**, a systematic benchmark designed to address a gap in the evaluation of protein motif scaffolding models: the lack of focus on general **geometric preservation**, as opposed to just functional motif preservation.

**Key Contributions:**

*   **A Novel Benchmark:** GeomMotif consists of 57 tasks derived from the PDB, each with a guaranteed ground-truth solution, designed to test the preservation of arbitrary structural fragments.
*   **Detailed Task Characterization:** Tasks are annotated with comprehensive structural and physicochemical properties, allowing for a fine-grained analysis of model strengths and weaknesses.
*   **Multi-faceted Evaluation:** The benchmark uses metrics for geometric fidelity (scRMSD, pLDDT) and structural diversity (clustering) to provide a holistic view of model performance.

The results show that different classes of models struggle with different types of geometric challenges, highlighting the utility of this benchmark for guiding future development in protein generative modeling.

**Strengths:**

1.  **Addresses a Critical Need in the Field:** The paper correctly identifies a significant gap in protein design: the lack of a systematic benchmark for the fundamental task of motif scaffolding. As most functional protein design now relies heavily on preserving specific motifs, the creation of a standardized benchmark to rigorously compare the performance of different scaffolding methods is both timely and important for the community.

2.  **Well-Designed and Comprehensive Evaluation Protocol:** The evaluation protocol is a key strength of this work. It is both reasonable and thorough. The introduction of the **SUN (Successful, Unique, Novel)** metric is particularly noteworthy, as it provides a single, well-motivated score that elegantly integrates the three most critical aspects of generative design performance, moving beyond simplistic success rates.

3.  **Thorough and Diverse Benchmark Construction:** The construction of the benchmark itself is exemplary. The selection and analysis of the motifs are highly detailed and thoughtfully curated to cover a wide and representative range of application scenarios. This comprehensiveness ensures that the benchmark can effectively probe the strengths and weaknesses of different models across various structural and physicochemical contexts.

**Weaknesses:**

### Weaknesses:

1.  **Exclusion of Small Motifs Limits Scope and Practical Relevance:** The benchmark's decision to exclude motifs with fewer than 30 residues is a notable limitation. In many real-world design scenarios, particularly for active sites or epitope engineering, the critical functional motifs are often very small, sometimes comprising just a few key residues. These small, and often non-contiguous, fragments are notoriously difficult for current models to constrain geometrically. Including a dedicated set of small-motif tasks would not only better reflect practical challenges but also provide a more stringent test of model capabilities, significantly enhancing the benchmark's overall utility.

2.  **Evaluation Could Be Extended to More Recent Methods and Tasks:** While the paper provides a solid comparison of established models, the field of protein generation is advancing rapidly. The benchmark would be more impactful and have greater immediate relevance if it included an evaluation of more recent state-of-the-art models, such as Protpardelle-1c, RFdiffusion2, or Proteina. Additionally, exploring performance on more nuanced motif-scaffolding tasks, like unindex motif-scaffolding, would provide a more comprehensive picture of model capabilities.

3.  **Lack of Focus on Side-Chain Preservation:** The current evaluation protocol is centered on backbone geometry (Cα atoms). However, for most functional applications of motif scaffolding (e.g., designing binders or enzymes), preserving the precise 3D conformation of the motif's *side-chains* is as crucial as preserving the backbone. Many of these functionally critical motifs are the very small ones mentioned in the first point. While the evaluated models are primarily backbone-focused, the benchmark itself would be more forward-looking and comprehensive if it included metrics for side-chain fidelity. Reporting on heavy-atom RMSD for the motif region, for instance, would better assess a model's ability to preserve function and encourage the development of all-atom scaffolding methods.

**Questions:**

1.  **On the Solvability Criterion and Benchmark Scale:** Regarding the construction of the benchmark, the paper employs a strict criterion for solvability (ESMFold prediction RMSD ≤ 1.0 Å to the PDB structure), which yields a final set of 107 structures. Could the authors comment on the rationale for this strictness? While it guarantees high-quality, predictable scaffolds, it also significantly constrains the diversity and scale of the benchmark. Have the authors considered an alternative definition of solvability more local to the motif itself—for instance, requiring only that the *motif region* is accurately predicted by a structure prediction model? This might allow for a much larger and more diverse set of benchmark tasks while still ensuring that the core scaffolding problem is well-posed.

2.  **Incorporating Functionally Relevant Motifs:** The paper makes a strong case for a benchmark focused purely on *geometric* preservation, distinct from function-focused benchmarks. As a complementary direction, have the authors considered creating a subset of tasks within GeomMotif where the geometric fragments are specifically chosen from known functional sites (e.g., catalytic residues, binding interfaces, or other hotspots)? This could help bridge the gap between geometric and functional challenges, providing valuable insights into whether current models are better or worse at preserving fragments that are known to be functionally important, even when evaluated on purely geometric metrics.

---

> ### Author Response · Authors · 2025-11-26
> **Official Comment by Authors (part 1)**
>
> Thank you for your thorough review and the time you invested in evaluating our work. We appreciate your recognition that GeomMotif addresses a critical need in the field by providing a benchmark for the fundamental task of motif scaffolding with a well-designed evaluation protocol and thorough benchmark construction.
>
> > **W1.** Exclusion of Small Motifs Limits Scope and Practical Relevance: The benchmark's decision to exclude motifs with fewer than 30 residues is a notable limitation. In many real-world design scenarios, particularly for active sites or epitope engineering, the critical functional motifs are often very small, sometimes comprising just a few key residues. These small, and often non-contiguous, fragments are notoriously difficult for current models to constrain geometrically. Including a dedicated set of small-motif tasks would not only better reflect practical challenges but also provide a more stringent test of model capabilities, significantly enhancing the benchmark's overall utility.
>
> Small functional motifs are indeed important in practical applications and are already addressed by the **Atomic Motif Enzyme Benchmark** from the recent RFdiffusion2 work, which specifically focuses on catalytic active sites. GeomMotif takes a **complementary approach** by systematically sampling structures with diverse geometric contexts rather than manually selecting known functional sites based on biological importance.
>
> We excluded motifs smaller than 30 residues to establish a **baseline of geometric complexity** that ensures each task presents a meaningful structural challenge. Preliminary analysis showed that smaller motifs frequently consisted of simple surface features or single contiguous fragments. Our design prioritizes **systematic coverage of protein structural space** with tasks stratified by fragment count and physicochemical properties. This approach reveals patterns in how models handle secondary structure composition, burial, and spatial complexity across varied contexts. Both the functional benchmarks like AME and our geometric approach provide valuable but different insights into model capabilities.
>
> > **W2.** Evaluation Could Be Extended to More Recent Methods and Tasks: While the paper provides a solid comparison of established models, the field of protein generation is advancing rapidly. The benchmark would be more impactful and have greater immediate relevance if it included an evaluation of more recent state-of-the-art models, such as Protpardelle-1c, RFdiffusion2, or Proteina. Additionally, exploring performance on more nuanced motif-scaffolding tasks, like unindex motif-scaffolding, would provide a more comprehensive picture of model capabilities.
>
> We thank the reviewer for these suggestions. We have evaluated all three models mentioned. The results reveal important insights about model specialization and design choices:
>
> | Model | Success (S) | Success (P) | Novel (S) | Novel (P) | Unique (S) | Unique (P) | SUN (S) | SUN (P) | Overall SUN |
> | :--- | :--- | :--- | :--- | :--- | :--- | :--- | :--- | :--- | :--- |
> | Genie2 | 60.1 | 32.9 | 60.1 | 26.6 | 59.9 | 22.5 | 59.9 | **18.8** | 39.4 |
> | La-Proteina | **67.1** | **62.7** | **67.1** | **35.2** | 61.3 | **22.7** | 61.3 | 16.2 | 38.8 |
> | RFdiffusion | 65.1 | 43.7 | 65.1 | 25.0 | **62.4** | 20.5 | **62.4** | 13.2 | 37.8 |
> | Protpardelle-1C | 56.2 | 44.6 | 56.2 | 25.2 | 53.5 | 22.6 | 53.5 | 14.1 | 33.8 |
> | FrameFlow | 30.6 | 25.1 | 30.6 | 19.7 | 30.6 | 20.2 | 30.6 | 16.0 | 23.3 |
> | RFdiffusion2 | 24.9 | 13.5 | 24.9 | 11.4 | 24.9 | 12.7 | 24.9 | 10.8 | 17.9 |
>
> *(S) = Single-motif, (P) = Paired-motif.*
>
> **La-Proteina demonstrates strong performance** with the highest success rates on both single and paired motifs. **RFdiffusion2 shows notably weaker performance**. **RFdiffusion2 was specifically tuned for enzyme design tasks** starting from small atomic motifs of catalytic active sites. This narrow focus appears to hinder performance on general geometric preservation tasks. In contrast, the original RFdiffusion was trained to be as broad as possible and performs much better on GeomMotif. This situation **shows the complementary nature** of the Atomic Motif Enzyme Benchmark and GeomMotif in evaluating different aspects of scaffolding capability.
>
> Regarding unindexed motif-scaffolding, we do not include such tasks because **there is no way to guarantee solvability** for problems where motif positions are not specified. This guaranteed solvability is a core design feature of GeomMotif that distinguishes it from other benchmarks. As noted by Zheng et al. (2025) in MotifBench, the absence of solvability guarantees has been a significant limitation in previous benchmarking efforts.
>
> *We have incorporated these results in our manuscript. Specifically, we have updated Fig. 5, Tab. 1, and App. L.*
>
> ---
>
> (continued below)

---

> ### Author Response · Authors · 2025-11-26
> **Official Comment by Authors (part 2)**
>
> > **W3.** Lack of Focus on Side-Chain Preservation: The current evaluation protocol is centered on backbone geometry (Cα atoms). However, for most functional applications of motif scaffolding (e.g., designing binders or enzymes), preserving the precise 3D conformation of the motif's side-chains is as crucial as preserving the backbone. Many of these functionally critical motifs are the very small ones mentioned in the first point. While the evaluated models are primarily backbone-focused, the benchmark itself would be more forward-looking and comprehensive if it included metrics for side-chain fidelity. Reporting on heavy-atom RMSD for the motif region, for instance, would better assess a model's ability to preserve function and encourage the development of all-atom scaffolding methods.
>
> We agree that side-chain preservation is important for functional applications. Our current focus on **backbone geometry reflects the primary output modality** of most current generative models and aligns with established evaluation practices in the field. The models we evaluate either generate backbone-only structures or generate sequences that are subsequently folded. In both cases, side-chain conformations emerge from the sequence design and folding steps rather than being directly controlled during generation. Adding all-atom evaluation metrics would conflate the geometric scaffolding capability with the performance of these downstream steps. As all-atom generative models become more prevalent, extending GeomMotif to include side-chain metrics would be a natural and valuable direction.
>
> > **Q1.** On the Solvability Criterion and Benchmark Scale: Regarding the construction of the benchmark, the paper employs a strict criterion for solvability (ESMFold prediction RMSD ≤ 1.0 Å to the PDB structure), which yields a final set of 107 structures. Could the authors comment on the rationale for this strictness? While it guarantees high-quality, predictable scaffolds, it also significantly constrains the diversity and scale of the benchmark. Have the authors considered an alternative definition of solvability more local to the motif itself—for instance, requiring only that the motif region is accurately predicted by a structure prediction model? This might allow for a much larger and more diverse set of benchmark tasks while still ensuring that the core scaffolding problem is well-posed.
>
> This threshold directly supports our **guaranteed solvability requirement**. Every GeomMotif task has a verified solution by our evaluation pipeline. We pre-filter all benchmark proteins to confirm ESMFold can accurately fold them (RMSD ≤ 1.0 Å).
>
> **ESMFold is a core component of our evaluation pipeline** and has inherent biases and limitations like any structure prediction model. Including proteins that ESMFold cannot accurately fold would make it **impossible to distinguish between model limitations and inadequacies of the evaluation pipeline itself**. Other benchmarks have suffered from including potentially unsolvable problems, compromising the interpretability of results.
>
> Regarding a more local solvability definition focused only on the motif region, such an approach would not adequately ensure that the complete scaffolding task is well-posed. The conformational stability and context of scaffold regions fundamentally affect whether motif geometry can be maintained in a physically plausible protein structure.
>
> Importantly, our 107-structure filtered dataset to sample motifs from not only guarantees that every task has a verified solution, but also provides **substantial diversity across CATH fold classes** (Fig. 4, App. D) **and physicochemical properties** (Fig. 10, Tab. 2 in revised version).
>
> ---
>
> (continued below)

---

> ### Author Response · Authors · 2025-11-26
> **Official Comment by Authors (part 3)**
>
> > **Q2.** Incorporating Functionally Relevant Motifs: The paper makes a strong case for a benchmark focused purely on geometric preservation, distinct from function-focused benchmarks. As a complementary direction, have the authors considered creating a subset of tasks within GeomMotif where the geometric fragments are specifically chosen from known functional sites (e.g., catalytic residues, binding interfaces, or other hotspots)? This could help bridge the gap between geometric and functional challenges, providing valuable insights into whether current models are better or worse at preserving fragments that are known to be functionally important, even when evaluated on purely geometric metrics.
>
> We thank the reviewer for acknowledging the strong case we make for focusing on pure geometric preservation. As discussed in our responses to W1 and W2, **functional motif scaffolding is already directly addressed** by existing benchmarks including RFdiffusion, MotifBench, and the recent Atomic Motif Enzyme Benchmark, which specifically evaluate performance on known functional sites such as catalytic residues and binding interfaces.
>
> GeomMotif provides a **complementary perspective** by systematically sampling structures with diverse geometric contexts to maximize coverage of protein structural space. This approach isolates the fundamental capability of arbitrary geometric preservation from the confounding effects of specific functional constraints. Importantly, **models can perform very differently** on geometric versus functional benchmarks, as evidenced by RFdiffusion2's strong performance on enzyme design tasks but weaker performance on GeomMotif's general geometric preservation challenges. Both approaches provide valuable insights into different aspects of model capabilities.
>
> ---
>
> *We hope we were able to answer your questions and address your concerns. If you find our responses satisfactory, we would be grateful if you would consider raising your rating. Please let us know if there is anything else we can clarify.*

---

### Official Review · Reviewer_TUmD · 2025-10-30

**Soundness:** 3
**Presentation:** 3
**Contribution:** 2
**Rating:** 6
**Confidence:** 3

**Summary:**

The authors construct a motif scaffolding benchmark through selecting diverse proteins from the PDB and computationally selecting motifs. The benchmark only measures whether the motif geometry is maintained after folding the designed sequence. They then compare a wide range of current state of the art protein design models on the benchmark, finding a clear difference between structure based models and sequence based models. The authors further analyze which motifs models find most challenging in terms of structural properties such as secondary structure composition and burial ratios.

**Strengths:**

Compared to MotifBench, the author's benchmark has a distinct advantage in that it doesn't require the generative model to design a structure because they do not compute self-consistency RMSD and only an RMSD on the motif after folding. This means they can compare structure-based and sequence-based models on the same benchmark. In doing so the authors reveal a significant difference between these two styles of model. Given that all models were run on the same benchmark by the same authors without bias as to a specific model, I believe these kinds of unbiased results are quite beneficial to the research community.

I also appreciated the analysis into which kinds of motifs are difficult for different models. For example the relative ease of scaffolding high helical content motifs and the difficulty with low helical content motifs is something that could be considered part of the folklore of protein design but not has evidence backing up this claim. The difficulty with buried motifs is also interesting and perhaps a new perspective for which motifs can be challenging to scaffold in practice.

**Weaknesses:**

I believe the motivation for the benchmark given in the introduction is unclear. It is claimed that MotifBench has a diagnostic blindspot in that it conflates geometric preservation with maintenance of function. I think what the authors mean here is that MotifBench requires both motif preservation and a low scRMSD to be a success whereas their benchmark only requires motif preservation and a high pLDDT of the folded structure. I don't see how low scRMSD equates to a test of 'function' for MotifBench. In my mind the main difference is that GeomMotif doesn't require a starting structure so can compare a wider array of models. I  think this should be made clearer.

With regards to comparing structure based and sequence based models, there seems to be quite an unfair advantage given to structure based models because they are allowed 8 MPNN sequences to attempt to find a good sequence that meets the success criteria. However, sequence based models only have 1 sequence to try to find success. It would therefore seem more appropriate to compare 100 structure samples with 8 MPNN sequences each with 800 sequence samples from the sequence generative model i.e. they have the same number of allowable calls to ESMFold to find a success.

I also believe this benchmark missed an opportunity to reduce the noise in the evaluation criteria. The authors use purely computational filters to finally select 57 motif scaffolding tasks from the entire PDB. This seems like quite a small number of tasks given that the hand selected MotifBench benchmark has 30 problems. Computationally selecting tasks opens the possibility of selecting a very large number of tasks. I believe a more comprehensive and lower noise evalauation of different methods could have been achieved with a much large number of motif scaffolding tasks but with fewer attempts at each task. For example, you currently have 100 attempts at each task for a total of 5700 designs. You could have instead picked 5700 motif scaffolding tasks with 1 attempt each which would have covered a much broader array of problems.

This problem is exemplified in Table 2 where the authors attempt to interpret drops in performance at certain numbers of fragments in the motif with some property of a model e.g. stating that Genie2 has a drop in performance at 4-5 fragments but good performance at 1-3 and 6-7 fragments. They then conclude that 'intermediate' complexity is difficult for the model. I personally am unconvinced by this explanation as I see no fundamental reason as to why intermediate numbers of segments should be harder than a large number of segments. I would offer an alternative explanation that simply the 5 tasks with 5 fragments happened to be harder than the other tasks because there is such a small number of tasks per number of fragments bin and the random sample of tasks that had 5 fragments happened to be hard. I think this is to be expected when the total number of tasks is small. If the authors wish to make general conclusions about task difficulty versus number of fragments I think many more tasks should be included in the benchmark.

In terms of writing, I think the clarity of section 4.4 could be improved with a figure since it is hard to make conclusions as a reader just reading stated numbers in the main text.

Overall, I believe the analysis of different models is of interest to the community more so than the benchmark itself due to the aforementioned issues. I am leaning towards a weak accept.

**Questions:**

Why not include some small motifs in the benchmark? Scaffolding the geometry of small functional sites is also an important capability for protein design models.

Would there be a way to include motifs that have more loop content? Sometimes loops can mediate important interactions and scaffolding their shape would be a practically relevant task.

---

> ### Author Response · Authors · 2025-11-26
> **Official Comment by Authors (part 1)**
>
> Thank you for your thorough review and the time you invested in evaluating our work. We appreciate the positive feedback and below we address each point in detail.
>
> > **W1.** I believe the motivation for the benchmark given in the introduction is unclear. It is claimed that MotifBench has a diagnostic blindspot in that it conflates geometric preservation with maintenance of function. I think what the authors mean here is that MotifBench requires both motif preservation and a low scRMSD to be a success whereas their benchmark only requires motif preservation and a high pLDDT of the folded structure. I don't see how low scRMSD equates to a test of 'function' for MotifBench. In my mind the main difference is that GeomMotif doesn't require a starting structure so can compare a wider array of models. I think this should be made clearer.
>
> Thank you for this excellent point. You are absolutely right that not requiring a starting structure enables **modality-agnostic comparison** between sequence-based and structure-based models on identical tasks. We appreciate you highlighting this key strength.
>
> Regarding the "diagnostic blind spot," let us clarify our motivation. **First, functional motif selection introduces bias.** MotifBench focuses on manually curated catalytic sites and binding interfaces, which inherently biases the benchmark toward specific structural and physicochemical contexts. GeomMotif systematically samples across the Protein Data Bank to ensure broad, unbiased coverage.
>
> **Second, geometric preservation and functional success should not be conflated.** Functional benchmarks implicitly assume that successful geometric preservation correlates with wet-lab functionality. However, beyond (backbone) geometry, functional success depends critically on physicochemical forces, i.e. residue identities, charge distributions, hydrophobic packing, side-chain conformations, etc. When a model fails on a functional motif, you cannot diagnose whether the failure stems from geometric inability or physicochemical incompatibility.
>
> GeomMotif isolates the geometric challenge by design. We guarantee every task is solvable (via ESMFold pre-filtering) and evaluate only 3D structure preservation. This provides a **pure diagnostic of geometric capability** that is complementary to functional benchmarks.
>
> We have revised the introduction to clarify these distinctions and emphasize modality-agnostic comparison as a central contribution.
>
> > **W2.** With regards to comparing structure based and sequence based models, there seems to be quite an unfair advantage given to structure based models because they are allowed 8 MPNN sequences to attempt to find a good sequence that meets the success criteria. However, sequence based models only have 1 sequence to try to find success. It would therefore seem more appropriate to compare 100 structure samples with 8 MPNN sequences each with 800 sequence samples from the sequence generative model i.e. they have the same number of allowable calls to ESMFold to find a success.
>
> Our evaluation follows the **standard protocol for motif scaffolding benchmarks** established by RFdiffusion, FrameFlow, Genie2, and MotifBench, which use 8 ProteinMPNN sequences per structure sample. This protocol reflects the fact that structure-based models generate backbone coordinates and require a separate inverse folding step to design compatible sequences, while sequence-based models directly generate sequences as their primary output. The 8 ProteinMPNN sequences represent the sampling needed to find a foldable sequence for a given backbone geometry. We are evaluating **end-to-end system performance** rather than isolated model components, which reflects practical protein design workflows.
>
> However, following your suggestion, we conducted additional experiments comparing structure-based models with 1 ProteinMPNN sequence versus sequence-based models with 8× compute:
>
> | Model | SUN Score |
> |-------|-----------|
> | DPLM-650M ×1 | 2.1 ± 0.1 |
> | DPLM-650M ×8 | 4.3 ± 0.2 |
> | ESM3 (seq) ×1 | 3.4 ± 0.1 |
> | ESM3 (seq) ×8 | 8.2 ± 0.3 |
> | RFdiffusion ×1 | 23.3 ± 0.3 |
> | RFdiffusion ×8 | 37.8 ± 0.3 |
> | Genie2 ×1 | 23.7 ± 0.4 |
> | Genie2 ×8 | 39.4 ± 0.5 |
>
> **Additional compute increases performance for all models but does not change the fundamental performance gap**. Structure-based models with single ProteinMPNN sequences still substantially outperform sequence-based models with 8× sampling. Our approach evaluates each generative model exactly 100 times per task, which provides a fair comparison while following established evaluation standards.
>
> *We have added these experiments as Appendix J with detailed single- and paired-motif breakdowns in Table 6.*
>
> ---
>
> (continued below)

---

> ### Author Response · Authors · 2025-11-26
> **Official Comment by Authors (part 2)**
>
> > **W3.** I also believe this benchmark missed an opportunity to reduce the noise in the evaluation criteria. The authors use purely computational filters to finally select 57 motif scaffolding tasks from the entire PDB. This seems like quite a small number of tasks given that the hand selected MotifBench benchmark has 30 problems. Computationally selecting tasks opens the possibility of selecting a very large number of tasks. I believe a more comprehensive and lower noise evaluation of different methods could have been achieved with a much large number of motif scaffolding tasks but with fewer attempts at each task. For example, you currently have 100 attempts at each task for a total of 5700 designs. You could have instead picked 5700 motif scaffolding tasks with 1 attempt each which would have covered a much broader array of problems.
>
> We acknowledge this alternative approach represents a valid design choice with different trade-offs. Our design prioritizes **per-task diversity analysis and robust metric computation**. The SUN metric requires sufficient samples per task to measure diversity through clustering and novelty through PDB comparison. With only one sample per task, we cannot assess whether a model generates structurally diverse solutions or merely repeats similar designs. The 100 samples per task enable bootstrap analysis that quantifies uncertainty and **provide reliable results even for weakly performing models**. For instance, we obtain meaningful 4% SUN scores for sequence-based models, whereas single-sample evaluation would yield predominantly zero measurements for these models. A 5700×1 approach would broaden structural context space but require fundamentally different evaluation metrics that sacrifice diversity assessment. Our 57 tasks span **diverse CATH fold classes and comprehensive physicochemical properties** (Fig. 4, App. D), providing systematic coverage of protein structural space with guaranteed solvability for every task.
>
> > **W4.** This problem is exemplified in Table 2 where the authors attempt to interpret drops in performance at certain numbers of fragments in the motif with some property of a model e.g. stating that Genie2 has a drop in performance at 4-5 fragments but good performance at 1-3 and 6-7 fragments. They then conclude that 'intermediate' complexity is difficult for the model. I personally am unconvinced by this explanation as I see no fundamental reason as to why intermediate numbers of segments should be harder than a large number of segments. I would offer an alternative explanation that simply the 5 tasks with 5 fragments happened to be harder than the other tasks because there is such a small number of tasks per number of fragments bin and the random sample of tasks that had 5 fragments happened to be hard. I think this is to be expected when the total number of tasks is small. If the authors wish to make general conclusions about task difficulty versus number of fragments I think many more tasks should be included in the benchmark.
>
> We thank you for this important observation. Upon aggregating performance across models by type and plotting with error bars (Fig.6), we observe that **structure-based models show the expected monotonic relationship between fragment complexity and performance**. The apparent non-monotonic patterns in Table 2 reflect task-specific difficulty variations rather than fundamental model properties. We have revised Section 4.3 to focus on the aggregate trends showing that increasing fragment complexity systematically reduces success rates for structure-based models. We also included Appendix K with the detailed breakdowns for each model.
>
>
> > **W5.** In terms of writing, I think the clarity of section 4.4 could be improved with a figure since it is hard to make conclusions as a reader just reading stated numbers in the main text.
>
> We appreciate this suggestion. We incorporated figures visualizing the property-performance relationships in Section 4.4 (Fig. 7-9).
>
> ---
>
> (continued below)

---

> ### Author Response · Authors · 2025-11-26
> **Official Comment by Authors (part 3)**
>
> > **Q1.** Why not include some small motifs in the benchmark? Scaffolding the geometry of small functional sites is also an important capability for protein design models.
>
> Following the suggestions of reviewers *v8Ac* and *aTcU*, we evaluated RFdiffusion2 along with La-Proteina and Protpardelle-1C. Small functional motifs are directly addressed by the **Atomic Motif Enzyme Benchmark** from the RFdiffusion2 work, which specifically focuses on catalytic active sites comprising just a few key residues. Interestingly, **RFdiffusion2 shows contrasting performance across benchmarks**. On the Atomic Motif Enzyme Benchmark focused on small functional motifs, RFdiffusion2 achieves 41/41 solved cases compared to 16/41 for the original RFdiffusion [1]. However, on GeomMotif this picture reverses: RFdiffusion2, which was specifically tuned for enzyme design tasks, achieves only 17.9% SUN score compared to 37.8% for the general-purpose RFdiffusion. This demonstrates that **models can excel at functional constraints while struggling with general geometric preservation**, highlighting the complementary nature of functional and geometric benchmarks. GeomMotif systematically samples structures with diverse geometric contexts to provide insights that complement functional benchmarks evaluating biologically validated sites.
>
> [1] Ahern, W. et al., Atom level enzyme active site scaffolding using RFdiffusion2. https://www.biorxiv.org/content/10.1101/2025.04.09.648075v2
>
> > **Q2.** Would there be a way to include motifs that have more loop content? Sometimes loops can mediate important interactions and scaffolding their shape would be a practically relevant task.
>
> Loops were not removed from GeomMotif. Our tasks include substantial loop content with many tasks containing significant unstructured regions. We applied filtering criteria to support our **guaranteed solvability requirement** where every task has a verified solution by our evaluation pipeline. Motifs dominated by loops lack well-defined tertiary structure and are difficult for structure prediction models to fold accurately. Including excessive loop content would compromise this guarantee, making it impossible to distinguish between model limitations and inherently unsolvable tasks. Importantly, **models still face substantial loop design challenges** in our benchmark when motifs contain multiple non-contiguous fragments requiring generated connecting regions that naturally include loops and flexible linkers between fixed structural elements.
>
> ---
>
> *We hope these responses address your concerns. If you find them satisfactory, we would be grateful if you would consider raising your rating. Please let us know if there is anything else we can clarify.*

---

### Official Review · Reviewer_v8Ac · 2025-10-31

**Soundness:** 3
**Presentation:** 3
**Contribution:** 3
**Rating:** 4
**Confidence:** 4

**Summary:**

This paper presents GeomMotif, a benchmark for evaluating geometric preservation in protein generation. Unlike existing benchmarks focusing on functional motifs, GeomMotif targets general structural fragments, assessing a model’s ability to maintain 3D geometry during motif scaffolding. The authors construct 57 benchmark tasks from the Protein Data Bank, covering diverse structural and physicochemical contexts. Results show that different model classes (sequence-based vs. structure-based) have distinct strengths, highlighting the challenge of arbitrary geometric preservation. Overall, GeomMotif provides a systematic and general framework for assessing geometry-aware protein generation.

**Strengths:**

- Clearly defines an important and underexplored evaluation problem in protein generation—geometric preservation.
- Proposes a well-constructed and diverse benchmark with 57 tasks covering various structural and physicochemical contexts.
- Provides systematic evaluation metrics (scRMSD, pLDDT) that go beyond success rates and enable detailed model comparison.
- Offers valuable insights into the complementary strengths of sequence-based and structure-based generative models.

**Weaknesses:**

- Lacks direct comparison with existing motif-scaffolding benchmarks (e.g., the RFDiffusion benchmark), which would better highlight the novelty and distinct contributions of GeomMotif.
- The evaluation focuses mainly on general geometric cases, without testing more complex or realistic scenarios such as symmetric motifs, small functional motifs, or protein target chains as motifs, which would demonstrate broader applicability.

**Questions:**

Q1 The paper constructs a diverse set of 57 benchmark tasks and systematically evaluates existing models. Could the authors further **categorize these tasks** (e.g., by difficulty, biochemical properties, or structural complexity) to form **tiered subsets** that would facilitate more detailed and interpretable future evaluations?

Q2 The selection of structure-based baselines seems limited to Genie2, RFdiffusion, and FrameFlow. Could the authors clarify the rationale behind this choice and consider including **more recent methods** such as Proteina or RFdiffusion2 to provide a more comprehensive comparison?

---

> ### Author Response · Authors · 2025-11-26
> **Official Comment by Authors**
>
> Thank you for your thoughtful review of our work. We appreciate your recognition that GeomMotif addresses an important problem with systematic evaluation metrics enabling detailed model comparison. Below we address your questions and concerns.
>
> > **Q2.** The selection of structure-based baselines seems limited to Genie2, RFdiffusion, and FrameFlow. Could the authors clarify the rationale behind this choice and consider including more recent methods such as Proteina or RFdiffusion2 to provide a more comprehensive comparison?
>
> Thank you for suggesting these baselines. We have evaluated both models you mentioned, along with Protpardelle-1C, another recently published model for motif scaffolding. The results reveal important insights about model specialization:
>
> | Model | Success (S) | Success (P) | Novel (S) | Novel (P) | Unique (S) | Unique (P) | SUN (S) | SUN (P) | Overall SUN |
> |-------|-------------|-------------|-----------|-----------|------------|------------|---------|---------|-------------|
> | Genie2 | 60.1 | 32.9 | 60.1 | 26.6 | 59.9 | 22.5 | 59.9 | **18.8** | 39.4 |
> | La-Proteina | **67.1** | **62.7** | **67.1** | **35.2** | 61.3 | **22.7** | 61.3 | 16.2 | 38.8 |
> | RFdiffusion | 65.1 | 43.7 | 65.1 | 25.0 | **62.4** | 20.5 | **62.4** | 13.2 | 37.8 |
> | Protpardelle-1C | 56.2 | 44.6 | 56.2 | 25.2 | 53.5 | 22.6 | 53.5 | 14.1 | 33.8 |
> | FrameFlow | 30.6 | 25.1 | 30.6 | 19.7 | 30.6 | 20.2 | 30.6 | 16.0 | 23.3 |
> | RFdiffusion2 | 24.9 | 13.5 | 24.9 | 11.4 | 24.9 | 12.7 | 24.9 | 10.8 | 17.9 |
>
> *Note: (S) = Single-motif, (P) = Paired-motif.*
>
> **La-Proteina demonstrates strong performance** with the highest success and novelty rates. This likely stems from La-Proteina being trained on the largest dataset among all evaluated models. **Genie2 achieves the highest SUN score** and remains marginally better overall. We hypothesize this advantage stems from Genie2 being the only model **trained from scratch purely on multi-fragment motif-scaffolding tasks** rather than being first pretrained and then fine-tuned for scaffolding.
>
> **RFdiffusion2 shows notably weaker performance than the original RFdiffusion** despite being the newest model. This likely stems from RFdiffusion2 being **specifically developed and tuned for enzyme design** starting from precise atomic motifs of catalytic active sites, while the original RFdiffusion was trained to be as broad as possible. We hypothesize that this narrow focus results in high scores on the Atomic Motif Enzyme Benchmark (41/41 solved cases [1]) but poor performance on GeomMotif (Success rate 19.2%, SUN 17.9%). In contrast, the original RFdiffusion was trained for a broad line of tasks and shows mediocre AME results (16/41 [1]) but much higher GeomMotif performance (Success rate 54.4%, SUN 37.8%). This pattern **highlights the complementary nature** of the Atomic Motif Enzyme Benchmark and GeomMotif in evaluating different aspects of scaffolding capability.
>
> *We have incorporated these results in our manuscript, specifically updating Fig. 5, Tab. 1, and App. L.*
>
> [1] Ahern, W. et al., Atom level enzyme active site scaffolding using RFdiffusion2. https://www.biorxiv.org/content/10.1101/2025.04.09.648075v2
>
> ---
>
> > **W1.** Lacks direct comparison with existing motif-scaffolding benchmarks (e.g., the RFDiffusion benchmark), which would better highlight the novelty and distinct contributions of GeomMotif.
>
> The RFdiffusion2 results help illustrate the complementarity between GeomMotif and functional benchmarks. GeomMotif is designed to be **complementary rather than directly comparable** to functional benchmarks. The key methodological difference is that functional benchmarks manually select known catalytic sites and binding interfaces based on biological importance, while we systematically sample structures to ensure broad representation across different fold classes and physicochemical properties.
>
> The RFdiffusion2 evaluation shows this complementarity in practice. RFdiffusion2 performs strongly on the Atomic Motif Enzyme Benchmark for enzyme design tasks, yet shows notably weaker performance on GeomMotif compared to the original RFdiffusion. This demonstrates that **models can excel at functional constraints while struggling with general geometric preservation**. Both benchmark types provide essential but different insights for understanding model capabilities.
>
> ---
>
> *(continues below)*

---

> ### Author Response · Authors · 2025-11-26
> **Official Comment by Authors (part 2)**
>
> > **W2.** The evaluation focuses mainly on general geometric cases, without testing more complex or realistic scenarios such as symmetric motifs, small functional motifs, or protein target chains as motifs, which would demonstrate broader applicability.
>
> We designed GeomMotif to isolate geometric preservation from other complexities, so we made deliberate decisions about our benchmark's scope.
>
> For **small functional motifs**, these are directly addressed by the Atomic Motif Enzyme Benchmark from recent RFdiffusion2 work, which specifically focuses on catalytic active sites. Including such motifs in GeomMotif would conflate geometric and functional challenges, undermining our core design principle of testing pure geometric preservation.
>
> For **protein target chains as motifs** (protein-protein interactions), we **intentionally construct tasks exclusively from monomeric structures**. Proteins in complexes often adopt different conformations in isolation, creating conformational ambiguity that would compromise evaluation consistency. When comparing generated structures against ground truth, we need the target geometry to be stable and well-defined independent of binding partners. Developing benchmarks for multimeric systems represents important future work that requires different methodological approaches to handle conformational flexibility and interface-dependent structural changes.
>
> We agree that **symmetric motifs** represent a valuable direction for future benchmark development. These introduce additional complexity beyond pure geometric preservation that warrants dedicated methodological treatment.
>
> > **Q1.** The paper constructs a diverse set of 57 benchmark tasks and systematically evaluates existing models. Could the authors further categorize these tasks (e.g., by difficulty, biochemical properties, or structural complexity) to form tiered subsets that would facilitate more detailed and interpretable future evaluations?
>
> Thank you for this suggestion. Our benchmark already provides extensive characterization enabling such analysis. Each task is defined by **eight structural and physicochemical properties** including secondary structure composition, burial ratio, hydrophobicity, charge density, and structural context ratio (Table 2 in revised manuscript). Tasks are additionally stratified by **fragment complexity** from 1-7 fragments and organized into single versus paired motif categories.
>
> These characterizations directly enable the detailed property-driven analysis presented in Section 4.4, where we demonstrate how helical content, burial patterns, and contact ratios systematically affect model performance. Researchers can readily construct tiered subsets based on any combination of these properties. For instance, tasks could be grouped by difficulty using success rates, or by structural similarity using secondary structure profiles. The **comprehensive property annotation** provides the foundation for flexible categorization schemes tailored to specific research questions without imposing a single prescribed hierarchy. *Following the recomendations of the reviewer *TUmD*, we included 3 figures to visualize the model sensitivity to structural properties (Fig. 7-9).*
>
> ---
>
> *We hope these responses address your concerns. If you find them satisfactory, we would be grateful if you would consider raising your rating. Please let us know if there is anything else we can clarify.*

---

### Author Response · Authors · 2025-12-03
**Official comment by Authors (part 1)**

We thank the Reviewers and Area Chair for their time and careful evaluation of our work. We appreciate the constructive feedback and specific suggestions that have strengthened our manuscript.

This response is organized as follows:

1. **TL;DR for the Area Chair**, a concise summary of GeomMotif's contributions and key findings
2. **Reviewer-Identified Strengths**, a synthesis of the positive aspects recognized across all reviews
3. **Reviewer-Identified Weaknesses and Our Resolutions**, a detailed accounting of each concern raised and the concrete steps we have taken to address them

---

**TL;DR for the Area Chair:**

GeomMotif benchmarks protein generation models on preserving 3D structural fragments during generation (analogous to image outpainting).

**Why needed?** Existing benchmarks focus only on functional sites (e.g. enzyme active sites), making it unclear whether model failures stem from inability to preserve geometry or from other complex functional requirements. Task solvability is often uncertain.

**Key improvements:**
- Systematically samples diverse structural contexts (not just functional sites)
- Guarantees all tasks are solvable
- Enables fair comparison between structure- and sequence-based models
- Characterizes tasks by complexity and 8 properties for detailed analysis

**Main findings (10 generative models evaluated):**
- Structure-based models achieve ~37-39% success; sequence-based models ~2-4%
- Sequence-based models fail categorically on spatially separated motifs; structure-based models show significant degradation (SUN score drops from ~60% to ~15%)
- Performance correlates with secondary structure (helices easier than β-sheets)
- RFdiffusion2's contrasting performance (41/41 enzyme tasks vs. 17.9% SUN here) shows that functional and geometric benchmarks test distinct capabilities.

**Summary of Revisions.** We added evaluation of 3 new baselines (La-Proteina, RFdiffusion2, Protpardelle-1C) with results in Table 1, Figure 5, and Appendix L. We conducted compute-fairness experiments (new Appendix J, Table 6). We added Figures 7-9 visualizing property-performance relationships per Reviewer ***TUmD***'s suggestion. We revised the Introduction to clarify modality-agnostic evaluation.

---

## Reviewer-Identified Strengths

- **Addresses critical gap.** Reviewers ***v8Ac*** and ***aTcU*** recognized that GeomMotif isolates geometric preservation from functional constraints.

- **Well-constructed benchmark.** All reviewers praised the systematic 57-task construction; Reviewer ***aTcU*** called it "exemplary".

- **Enables modality-agnostic comparison.** Reviewers ***TUmD*** and ***oyxS*** highlighted the "distinct advantage" that GeomMotif allows fair comparison of structure-based and sequence-based models on identical tasks.

- **Comprehensive evaluation protocol.** Reviewers ***v8Ac*** and ***aTcU*** commended the systematic metrics (scRMSD, pLDDT, SUN) enabling detailed model comparison. Reviewer ***aTcU*** called the SUN metric "particularly noteworthy."

- **Unbiased results beneficial to community.** Reviewer ***TUmD*** noted that having all models evaluated by the same authors without bias provides results "quite beneficial to the research community."

- **Reveals actionable insights.** Reviewers ***v8Ac***, ***TUmD***, and ***oyxS*** valued the property-based analysis showing which structural features systematically affect model performance.

- **Clear presentation.** Reviewer ***oyxS*** praised the manuscript as "clearly written" with benchmark construction "well explained," noting "No specific weaknesses in the manuscript."

---

*(continues below)*

---

> ### Author Response · Authors · 2025-12-03
> **Official comment by Authors (part 2)**
>
> ## Reviewer-Identified Weaknesses and Resolutions
>
> - **Limited evaluation of recent methods.** Reviewers ***v8Ac*** (Q2) and ***aTcU*** (W2) noted the absence of recent models (Protpardelle-1c, RFdiffusion2, La-Proteina).
>   - We evaluated all three models with comprehensive results added to **Table 1, Fig. 5, and App. L**. Notably, **RFdiffusion2's contrasting performance** (41/41 on enzyme benchmark vs. 17.9% SUN on GeomMotif) demonstrates the complementary nature of functional vs. geometric benchmarks.
>
> - **Benchmark scope excludes certain motif types.** Reviewers ***aTcU*** (W1), ***v8Ac*** (W2), and ***oyxS*** (Q1, Q2) questioned excluding small motifs, symmetric motifs, PPIs, and high-loop-content motifs.
>   - **Small functional motifs are comprehensively covered by the Atomic Motif Enzyme Benchmark**, making GeomMotif complementary rather than redundant.
>   - Our 30-residue threshold enables diversity of tasks across CATH fold classes (Fig. 4).
>   - **Loops were not removed**—tasks include up to 23.3% loop content (Table 2), with filtering only for excessive loop content to maintain guaranteed solvability.
>   - **PPIs excluded because proteins in complexes adopt different conformations in isolation**, creating evaluation ambiguity.
>
> - **Benchmark scale considerations.** Reviewer ***TUmD*** (W3, W4) suggested more tasks with fewer attempts per task could reduce noise.
>   - Our 100-sample design **enables the SUN metric's diversity assessment through clustering** (impossible with single samples), **bootstrap uncertainty quantification**, and **meaningful scores for weak models** (4% interpretable vs. predominantly zeros).
>   - Our 57 tasks provide comprehensive coverage of CATH folds and physicochemical properties (Fig. 4, Fig. 10).
>   - Task-specific variations exist but **aggregate trends by model type are consistent** (Fig. 6, App. K).
>
> - **Evaluation fairness.** Reviewer ***TUmD*** (W2) noted structure-based models receive 8 ProteinMPNN sequences while sequence-based models get 1 sequence.
>   - We conducted experiments comparing equal compute budgets (**Appendix J, Table 6**). Structure-based models with 1× MPNN (RFdiffusion: 23.3%, Genie2: 23.7%) substantially outperform sequence-based with 8× samples (ESM3: 8.2%, DPLM-650M: 4.3%). **The performance gap persists regardless of sampling budget.** Our protocol follows established standards (RFdiffusion, FrameFlow, Genie2, MotifBench).
>
> - **Comparison with functional benchmarks.** Reviewer ***v8Ac*** (W1) suggested direct comparison with existing benchmarks would highlight GeomMotif's contributions.
>   - The **RFdiffusion2 evaluation provides this comparison empirically**: models specialized for functional constraints (41/41 enzyme tasks) can struggle with general geometric preservation (17.9% vs. 37.8% SUN). GeomMotif systematically samples structural space while functional benchmarks curate biologically important sites, so both perspectives are essential.
>
> - **No side-chain evaluation.** Reviewer ***aTcU*** (W3) noted backbone-only focus misses side-chain preservation.
>   - Current models output backbones (structure-based) or sequences (sequence-based). **Including side-chain metrics would conflate scaffolding capability with downstream folding/packing performance**. Our backbone focus aligns with established practices and current model capabilities.
>
> - **Clarity improvements.** Reviewer ***TUmD*** (W1, W5) suggested improving introduction on modality-agnostic nature of GeomMotif and requested figures for Section 4.4.
>   - **Introduction revised** to clarify modality-agnostic comparison and how functional motif selection introduces bias.
>   - **Figures 7-9 added** visualizing property-performance relationships.
>
>   ---
>
> We believe we have substantively addressed each concern raised by the reviewers. We thank the reviewers and Area Chair for feedback that has strengthened this contribution to the protein design community.

---

### Meta-Review · Area_Chair_VEGr · 2026-01-06

**Summary:**

The main concern of the paper is limited evaluation of recent methods, the exclusion of certain motif types.

**Reviewer Concerns:**

All the reviewer concerns have been addressed well

**Reviewer Scores:**

The reviewers have participated in the discussion.

---

### Decision · Program_Chairs · 2026-01-26

Accept (Poster)